# Representation of phosphorus cycle in Joint UK Land Environment Simulator (vn5.5_JULES-CNP)

Mahdi (André) Nakhavali[1], Lina M. Mercado[1,2], Iain P. Hartley[1], Stephen Sitch[1], Fernanda V Cunha[3], Raffaello di Ponzio[3], Laynara F. Lugli[3], Carlos A. Quesada[3], Kelly M. Andersen[1,4,5], Sarah E. Chadburn[6], Andy J. Wiltshire[1,7], Douglas B. Clark[2], Gyovanni Ribeiro[3], Lara Siebert[3], Anna C. M. Moraes[3], Jéssica Schmeisk Rosa[3], Rafael Assis[3] and José L. Camargo[3]

[1]University of Exeter, College of Life and Environmental Sciences, Exeter, EX4 4QE, United Kingdom
[2]UK Centre for Ecology and Hydrology, Wallingford, OX10 8BB, United Kingdom
[3]Coordination of Environmental Dynamics, National Institute of Amazonian Research, Manaus, AM 69060-062, Brazil
[4] University of Edinburgh, School of Geosciences, Edinburgh, EH8 9AB, UK
[5] Nanyang Technological University, Asian School of the Environment, Singapore, 639798, Singapore
[6]College of Engineering, Mathematics, and Physical Sciences, University of Exeter, Exeter, EX4 4QE, United Kingdom
[7]Met Office Hadley Centre, Exeter, Devon, EX1 3PB, United Kingdom

*Correspondence to*: Mahdi (André) Nakhavali (m.nakhavali@exeter.ac.uk)

## Abstract

Most Land Surface Models (LSMs), the land components of Earth system models (ESMs), include representation of nitrogen (N) limitation on ecosystem productivity. However only few of these models have incorporated phosphorus (P) cycling. In tropical ecosystems, this is likely to be important as N tends to be abundant but the availability of rock-derived elements, such as P, can be very low. Thus, without a representation of P cycling, tropical forest response in areas such as Amazonia to rising atmospheric $CO_2$ conditions remains highly uncertain. In this study, we introduced P dynamics and its interactions with the N and carbon (C) cycles into the Joint UK Land Environment Simulator (JULES). The new model (JULES-CNP) includes the representation of P stocks in vegetation and soil pools, as well as key processes controlling fluxes between these pools. We develop and evaluate JULES-CNP using in situ data collected at a low fertility site in the Central Amazon, with a soil P content representative of 60% of soils across the Amazon basin, to parameterise, calibrate and evaluate JULES-CNP. Novel soil and plant P pool observations are used for parameterisation and calibration and the model is evaluated against C fluxes and stocks, and for those soil P pools not used for parameterisation/calibration. We then evaluate the model at additional P limited test sites across the Amazon, in Panama and Hawaii showing a significant improvement over the C and CN only versions of the model. The model is then applied under elevated $CO_2$ (600 ppm) at our study site Central Amazon to quantify the impact of P limitation on $CO_2$ fertilization. We compare our results against current state of the art CNP models using the same methodology that was used in the AmazonFACE model intercomparison study. The model is able to reproduce the observed plant and soil P pools and fluxes used for evaluation under ambient $CO_2$. We estimate P to limit net primary productivity (NPP) by 24% under current $CO_2$ and by 46% under elevated $CO_2$. Under elevated $CO_2$, biomass in simulations accounting for CNP increase by 10% relative to contemporary $CO_2$ conditions, although it is 5% lower compared with CN and C-only simulations. Our results highlight the potential for high P limitation and therefore lower $CO_2$ fertilization capacity in the Amazon forest with low fertility soils.

## 1. Introduction

Land ecosystems currently take up about 30% of anthropogenic $CO_2$ emissions (Friedlingstein *et al.*, 2020), thus buffering the anthropogenic increase in atmospheric $CO_2$. Tropical forests play a major role in the land C cycle, account for about half of global net primary production (NPP)(Schimel *et al.*, 2015), and store the highest above ground carbon among all biomes (Pan *et al.*, 2011; Mitchard, 2018).

The C sink capacity of tropical forests may be constrained by nutrient availability for plant photosynthesis and growth (Vitousek and Howarth, 1991; Elser *et al.*, 2007; LeBauer and Treseder, 2008) via P (Nordin, Högberg and Näsholm, 2001; Shen *et al.*, 2011) and/or N related processes (DeLuca, Keeney and McCarty, 1992; Perakis and Hedin, 2002). Global process-based models of vegetation dynamics and function suggest a continued land C sink in the tropical forests, largely attributed to the $CO_2$ fertilization effect (Sitch *et al.*, 2008; Schimel, Stephens and Fisher, 2015; Koch, Hubau and Lewis, 2021). However, many of these models typically do not consider P constraints on plant growth (Fleischer *et al.*, 2019), which is likely to be an important limiting nutrient in tropical ecosystems, characterised by old and heavily weathered soils. The importance of nutrient cycling representation in Earth System Models (ESMs), and the lack thereof, was highlighted by Hungate *et al.* (2003) and Zaehle and Dalmonech (2011), showing the significance of nitrogen inclusion in ESMs for generating more realistic estimations of the future evolution of the terrestrial C sink. However, in the Coupled Climate C Cycle Model Inter-comparison Project (C4MIP), none of the participating ESMs included N dynamics (Friedlingstein *et al.*, 2006). Seven years later, for the update in CMIP5 (Anav *et al.*, 2013), three models out of eighteen with N dynamics were included (Bentsen *et al.*, 2013; Long *et al.*, 2013; Ji *et al.*, 2014). Although much progress has been made in the inclusion of an N cycle in ESMs so far, none of the CMIP5 models included P cycling and in the most recent CMIP6, only one model includes P (ACCESSESM1.5 model) (Arora *et al.*, 2020).

The long history of soil development in tropical regions which involves the loss of rock-derived nutrients through weathering and leaching on geologic timescales (Vitousek et al., 1997, 2010) results in highly weathered soils. Soil P is hypothesized to be among the key limiting nutrients to plant growth in tropical forests (Vitousek *et al.*, 1997, 2010; Hou *et al.*, 2020), unlike temperate forest where N is hypothesised to be the main constraint (Aerts and Chapin, 1999; Luo *et al.*, 2004). Low P availability in tropical soils is related to the limited un-weathered parent material or organic compounds as source of P (Walker and Syers, 1976), active sorption (Sanchez, 1977) and high occlusion (Yang and Post, 2011) which further reduce plant available P. Although N limitation can impact the terrestrial C sink response to increasing atmospheric $CO_2$ by changing plant C fixation capacity (Luo et al., 2004), this can be partially ameliorated over time by input of N into the biosphere via the continuous inputs of N into ecosystems from atmospheric deposition and biological N fixation (Vitousek et al., 2010). P-limitation is pervasive in natural ecosystems (Hou *et al.*, 2020) and the lack of large P inputs into ecosystems, especially those growing on highly weathered soil, may make P limitation a stronger constraint on ecosystem response to elevated $CO_2$ ($eCO_2$) than N (Gentile et al., 2012; Sardans, Rivas-Ubach and Peñuelas, 2012). This causes considerable uncertainty in predicting the future of the Amazon forest C sink (Yang *et al.*, 2014).

There is evidence to suggest P limitation on plant productivity in the Amazon forest (Malhi, 2012) where it has been shown that the younger, more fertile west and south-west Amazon soils have higher tree turnover (Phillips et al., 2004; Stephenson and Van Mantgem, 2005) and stem growth rates (Malhi *et al.*, 2004) and lower above ground biomass (Baker *et al.*, 2004; Malhi *et al.*, 2006) compared to their central and eastern counterparts. Total soil P has been found as the best predictor of stem growth (Quesada *et al.*, 2010) and of total NPP (Aragão *et al.*, 2009) across this fertility gradient, and foliar P is positively related to plant photosynthetic capacity ($V_{cmax}$ and $J_{cmax}$) in these forests (Mercado *et al.*, 2011).

However, modelling studies are unable to reproduce observed spatial patterns of NPP and biomass in the Amazon , one possible reason being the lack of inclusion of soil P constraints on plant productivity and function (Wang, Law and Pak, 2010; Vicca *et al.*, 2012a; Yang *et al.*, 2014). Nevertheless, some modelling studies have focused on improving process and parameter representation using the observational data of spatial variation in woody biomass residence time (Johnson *et al.*, 2016), soil texture and soil P to parameterise the maximum carboxylation capacity ($V_{cmax}$) (Castanho *et al.*, 2013). Results from these studies successfully represent observed patterns of Amazon forest biomass growth increases with increasing soil fertility. However, the full representation of these interactions and the impact of the soil nutrient availability on biomass productivity is still missing in most of ESMs.

So far, several dynamic global vegetation models have been developed to represent P cycling within the soil (Yang *et al.*, 2013; Haverd *et al.*, 2018) and between plant and soils for tropical forests particularly (Yang *et al.*,

2014; Zhu *et al.*, 2016; Goll *et al.*, 2017). Furthermore, a comprehensive study included several models with C-
N-P cycling and their feedbacks on the atmospheric C fixation and biomass growth in Amazon forests under
ambient and elevated $CO_2$ conditions ($eCO_2$) (Fleischer *et al.*, 2019). Despite these developments, data to
underpin them and their projections, particularly for the tropics, is sparse and remains challenging particularly
for the Amazon forest (Reed *et al.*, 2015; Jiang *et al.*, 2019). Moreover, due to the lack of detailed
measurements, the P-related processes such as ad/desorption and uptake represented in these models are under-
constrained and likely oversimplified, thus the future predictions of Amazon forest responses to $eCO_2$ and
climate change are uncertain. To fill this gap, in this study, we use data collected as part of the Amazon
Fertilization Experiment (AFEX), the first project that focuses on experimental soil nutrient manipulation in the
Amazon, with a comprehensive data collection program covering plant ecophysiology, C stocks and fluxes, soil
processes including P stocks. Thus, our model parameterization compared to prior P modelling studies includes
detailed P processes representation using the site measurements.
Here, we describe the development and implementation of the terrestrial P cycle in the Joint UK Land
Environment Simulator (JULES) (Clark *et al.*, 2011), the land component of the UK Earth System Model
(UKESM), following the structure of the prior N cycle development (Wiltshire *et al.*, 2021) and utilising state of
the art already tested and implemented descriptions of P cycling in other land surface models (Wang, Houlton
and Field, 2007; Zhu *et al.*, 2016; Goll *et al.*, 2017).
The model (JULES-CNP) is parameterized and calibrated using novel in situ P soil and plant data from a well-
studied forest site in Central Amazon near to Manaus, Brazil with soil P content representative of 60% of soils
across the Amazon basin. The new developed P component estimates the sorption of the soil organic and
inorganic P based on the saturation status of the adsorbed P pools, which is unique compared to the other
existing P models and enable more realistic estimation of P ad/desorption processes. We first evaluate the model
at our study site but also at additional five test sites across the Amazon, in Panama and Hawaii. We then apply
the model under ambient and $eCO_2$ following the protocol of Fleischer *et al.*, (2019) to predict nutrient
limitations on land biogeochemistry under these conditions. Predictions of the $CO_2$ fertilization effect in JULES-
CNP are compared to those in current versions of the model with coupled C and N cycles (JULES-CN) and with
C cycle only (JULES-C).
**2.   Material and methods**
**2.1  JULES**
JULES is a process-based model that integrates water, energy, C cycling (JULES-C) (Clark *et al.*, 2011) and N
cycling (JULES-CN) (Wiltshire *et al.*, 2021) between the atmosphere, vegetation and soil (Best *et al.*, 2011;
Clark *et al.*, 2011). Vegetation dynamics are represented in JULES using the TRIFFID model, using nine
distinct plant functional types (PFTs) (tropical and temperate broadleaf evergreen trees, broadleaf deciduous
trees, needle-leaf evergreen and deciduous trees, C3 and C4 grasses, and evergreen and deciduous shrubs), as
well as height competition (Harper *et al.*, 2016).  Leaf-level photosynthesis (Collatz *et al.*, 1991; Collatz, Ribas-
Carbo and Berry, 1992) is scaled to estimate canopy level Gross Primary Productivity (GPP) using a multilayer
approach that accounts for vertical variation of radiation interception and partition of sunlit and shaded leaves
and associated vertical variation of leaf N and P -exponential decrease through the canopy (Clark *et al.*,
2011:Mercado *et al* 2007, Mercado et al 2009) - while the C:P and N:P ratios remain the same. NPP is estimated
as the difference between GPP and autotrophic respiration for each living tissue (leaf, wood, root). NPP is then
allocated to increase tissue C stocks and to spread, i.e., expand the fractional coverage of the PFT. The resultant
PFT fractional coverages also depend on competition across PFTs for resources, e.g., light. Tissue turnover and
vegetation mortality add C into the litter pools. Representation of soil organic C (SOC) follows the Rothamsted
Carbon model RothC equations (Jenkinson *et al.*, 1990; Jenkinson and Coleman, 2008) defining four C pools:
decomposable plant material (DPM) and resistant plant material (RPM), which receive direct input from
litterfall, and microbial biomass (BIO) and humified material (HUM) which receive a fraction of decomposed C
from DPM and RPM which is not released to the atmosphere. The limitation of N on SOC is applied to the
vegetation and soil components using a dynamic C:N ratio to modify the mineralization and immobilization
processes as described in Wiltshire *et al.*, (2021). Note that the soil component of JULES-CN can be run either
as a single box model or vertically resolved over soil depth (JULES-CN layered), and in this paper we build
upon the vertically resolved version described in Wiltshire et al. (2021).

## 2.2 JULES-CNP

JULES-CNP includes the representation of the P cycle in JULES version (vn5.5) and it is built on existing and well tested representations of P cycling in other global land surface models (Wang, Houlton and Field, 2007; Yang *et al.*, 2014; Goll *et al.*, 2017; Sun *et al.*, 2021). It includes P fluxes within the vegetation and soil components, and the specification of P pools and processes related to P cycling within the soil column (Fig. 1). A parent material pool is introduced to consider the input of weathered P. The adsorbed, desorbed and occluded fractions of P for both organic and inorganic P are also represented. However, except for parent material and occluded P pools, all other pools are estimated at each soil layer. The description of changes in pools and associated relative fluxes are explained in detail in the next sections. Although JULES-CN includes N leaching and deposition, P leaching and deposition are not included in the current version of JULES-CNP.

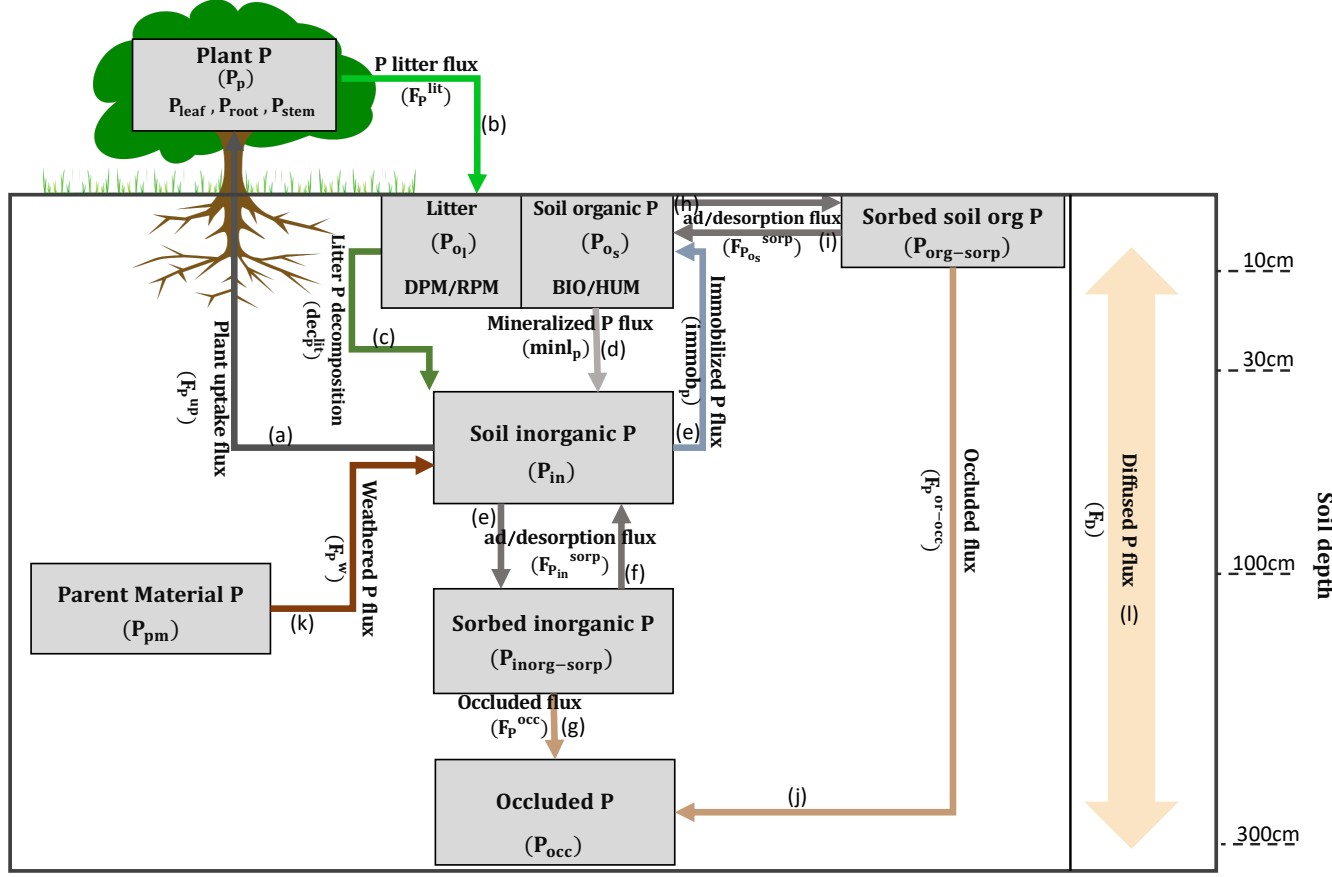

**Figure.1 – JULES-CNP model scheme including P pools (grey boxes) and fluxes (arrows)**

### 2.2.1    P pools

JULES represents eight P pools comprising organic and inorganic P: plant P ($P_p$) and soil P pools (in each soil layer (n)), litter P ($P_{O_l}$), soil organic P ($P_{O_s}$), soil inorganic P ($P_{in}$), organic sorbed ($P_{org-sorp}$), inorganic sorbed ($P_{inorg-sorp}$), parent material ($P_{pm}$) and occluded ($P_{occ}$) P comprised of both organic and inorganic P. All pools are in units of kg P m$^{-2}$ (Fig 1, Tables 1 and 2).

Plant P pool is composed of leaf ($P_{leaf}$), fine root ($P_{root}$) and stem together with coarse root ($P_{stem}$), which are related to their associated C pools ($C_{leaf}, C_{root}, C_{stem}$) in (kg C m$^{-2}$) and fixed C to P ratios ($C:P_{leaf}, C:P_{root} C:P_{stem}$) as follows:

$$P_{leaf} = \frac{C_{leaf}}{C:P_{leaf}}$$ (eq.1)

$P_{root} = \frac{C_{root}}{C:P_{root}}$          (eq.2)

$P_{stem} = \frac{C_{stem}}{C:P_{stem}}$          (eq.3)

Therefore, the plant P pool ($P_p$) is the sum of all vegetation P pools as follows:

$P_p = P_{leaf} + P_{root} + P_{stem}$          (eq.4)

Description of the plant P pool ($P_p$) follows Zhu *et al.*, (2016) and is estimated as the difference between the
input, plant uptake $F_P^{Up}$ (eq.26) and output of this pool, plant litter flux $F_P^{lit}$ (eq.28), with both fluxes
expressed in kg P m$^{-2}$ yr$^{-1}$ as follows:

$\frac{dP_p}{dt} = F_P^{up} - F_P^{lit}$          (eq.5)

The litter P pool ($P_{O_l}$) is estimated as a sum of $P_{DPM}$ and $P_{RPM}$ pools over soil layers (n). Each pool is formed by
the fluxes of plant litter input ($F_P^{lit}$) and the outgoing decomposed P ($dec_P^{lit}$) both expressed in kg P m$^{-2}$ yr$^{-1}$
(eq.28-29). Furthermore, the plant litter input is modified based on the plant type material ratio α (in order to
distribute the litter input based on the DPM/RPM fraction) as follows:

$\frac{dP_{DPM}}{dt} = F_{P_n}^{lit} \times \alpha - dec_{P_{DPM,n}}$          (eq.6)

$\frac{dP_{RPM}}{dt} = F_{P_n}^{lit} \times (1 - \alpha) - dec_{P_{RPM,n}}$          (eq.7)

$P_{O_l} = \sum_{n=1}^{N} P_{DPM_n} + \sum_{n=1}^{N} P_{RPM_n}$          (eq.8)

The soil organic pool ($P_{O_s}$) is represented as the sum of $P_{BIO}$ and $P_{HUM}$. These pools are estimated from the
difference between P inputs from total immobilized ($F_{immob_P}$) distributed between BIO and HUM based on
fixed fraction (0.46 for BIO, 0.54 for HUM) (Jenkinson *et al.*, 1990; Jenkinson and Coleman, 2008) and
desorbed P, $F_{P_{O_S}}^{desorp}$ and P outputs from mineralized ($F_{minl_P}$), and adsorbed P fluxes ($F_{P_{O_S}}^{sorp}$) (adsorption:
eq. 40 and desorption: eq.41) with all fluxes expressed in kg P m$^{-2}$ yr$^{-1}$ as follows:

$\frac{dP_{BIO}}{dt} = 0.46 \times F_{immob_{P_n}} + F_{P_{O_{S_{BIO,n}}}}^{desorp} - F_{minl_{P_{BIO,n}}} - F_{P_{O_{S_{BIO,n}}}}^{sorp}$      (eq.9)

$\frac{dP_{HUM}}{dt} = 0.54 \times F_{immob_{P_n}} + F_{P_{O_{S_{HUM,n}}}}^{desorp} - F_{minl_{P_{BIO,n}}} - F_{P_{O_{S_{HUM,n}}}}^{sorp}$      (eq.10)

$P_{O_s} = \sum_{n=1}^{N} P_{BIO_n} + \sum_{n=1}^{N} P_{HUM_n}$          (eq.11)


Description of the inorganic sorbed P pool ($P_{inorg-sorp}$) follows Wang *et al.*, (2007) and is represented as the
difference between the input flux of inorganic sorption ($F_{P_{in}}^{sorp}$) (eq. 37) and output fluxes of inorganic
desorption ($F_{P_{in}}^{desorp}$) (eq. 38) and occluded P ($F_P^{occ}$) (eq. 39), with all fluxes expressed in kg P m$^{-2}$ yr$^{-1}$ as
follows:

$\frac{dP_{inorg-sorp}}{dt} = \sum_{n=1}^{N} F_{P_{in_n}}^{sorp} - \sum_{n=1}^{N} F_{P_{in_n}}^{desorp} - \sum_{n=1}^{N} F_{P_n}^{occ}$      (eq.12)

The description of the occluded ($P_{occ}$) P pool follows Wang *et al.*, (2007) and Hou *et al.*, (2019 ) and is
represented as the sum of input fluxes of occluded P from both organic ($F_P^{or-occ}$) (eq. 42) and inorganic P
pools ($F_P^{occ}$) expressed in kg P m$^{-2}$ yr$^{-1}$, as follows:

$\frac{dP_{occ}}{dt} = \sum_{n=1}^{N} F_{P_n}^{occ} + \sum_{n=1}^{N} F_{P_n}^{or-occ}$          (eq.13)

The description of the organic sorbed P pool ($P_{org-sorp}$) follows Wang *et al.*, (2007) and is represented as the difference between the input flux of organic sorption ($F_{P_{O_{S_n}}}^{sorp}$) and output fluxes of organic desorption ($F_{P_{O_{S_n}}}^{desorp}$) and occluded P($F_{P_n}^{occ}$), with all fluxes expressed in kg P m$^{-2}$ yr$^{-1}$ as follows:

$$\frac{dP_{org-sorp}}{dt} = \sum_{n=1}^{N} F_{P_{O_{S_n}}}^{sorp} - \sum_{n=1}^{N} F_{P_{O_{S_n}}}^{desorp} - \sum_{n=1}^{N} F_{P_n}^{or-occ} \qquad (eq.14)$$

P from parent material ($P_{pm}$) pool follows Wang *et al.*, (2007) and depends on the weathering flux ($F_P^{w}$) (eq. 43) in kg P m$^{-2}$ yr$^{-1}$ as follows:

$$\frac{dP_{pm}}{dt} = -\sum_{n=1}^{N} F_{P_n}^{w} \qquad (eq.15)$$

### 2.2.2.   C and P fluxes

NPP in JULES is calculated as the difference between GPP and autotrophic respiration. In JULES-CNP, potential NPP represents the amount of C, available for tissue growth (C density increase) on a unit area, and spreading (vegetation cover increase as a result of reproduction and recruitment), i.e., to increase the area covered by the vegetation type, assuming no nutrient limitation. The reported NPP in the literature often includes other C fluxes related to the exudates, production of volatiles and non-structural carbohydrates (Malhi *et al.*, 2009; Chapin *et al.*, 2011; Walker *et al.*, 2021) which are challenging to measure (Malhi, Doughty and Galbraith, 2011). Therefore, actual NPP is for our purposes equal to Biomass Production (BP), and is calculated as potential NPP minus excess C (lost to the plant through autotrophic respiration), with the latter the C that cannot be used to grow new plant tissue due to insufficient plant nutrient supply. Hence, if the system is limited by the availability of N and/or P, NPP will be adjusted to match the growth that can be supported with the limited N or P supply, with any excess carbohydrate lost through excess C.
The total excess C term ($\psi_t$) (kg C m$^{-2}$ yr$^{-1}$) is calculated as:

$$\psi_t = \psi_g + \psi_s \qquad (eq.16)$$

where $\psi_g$ and $\psi_s$ are the excess C fluxes due to growth (g) and spread (s) and are assumed to be rapidly respired by plants.

Therefore, BP is calculated as the difference between potential NPP ($\Pi_c$) and total excess C:

$$BP = \Pi_c - \psi_t \qquad (eq.17)$$

The litter production in JULES before limitation is estimated as follows:

$$F_{C_n}^{lit} = \gamma_{leaf} C_{leaf} + \gamma_{root} C_{root} + \gamma_{wood} C_{wood} \qquad (eq.18)$$

where $\gamma$ is a temperature dependent turnover rate representing the phenological state (Clark *et al.*, 2011). P limitation is applied on the C litter production similar to the N scheme of JULES (JULES-CN) (Wiltshire *et al.*, 2021). In JULES-CN the N limitation effect on the litter production is captured by estimating the available C for litter production as a difference between the NPP and excess C (Wiltshire *et al.*, 2021).

Similar to other P-enabled models (Yang *et al.*, 2014; Goll *et al.*, 2017), JULES-CNP follows the same structure as its N model component. Description of the plant P and N demand follow Wang *et al.*, (2007) and are represented by the sum of demand ($\emptyset_t$) to sustain growth (P-related: ($\emptyset_{g_P}$), N-related: ($\emptyset_{g_N}$)) and to sustain vegetation spreading (to increment PFT fractional coverage) (P-related: ($\emptyset_{S_P}$), N-related: ($\emptyset_{S_N}$)) and is expressed in (P-related in kg P m$^{-2}$ yr$^{-1}$; N-related in kg N m$^{-2}$ yr$^{-1}$). The total demand for growth ($\emptyset_g$) and spreading ($\emptyset_s$) is controlled by the dominant demand between P ($\emptyset_{g_P}$) and N ($\emptyset_{g_N}$) as follows:

$$\emptyset_t = \emptyset_g + \emptyset_s \qquad (eq.19)$$

$\emptyset_{gP} = \frac{P_p}{C_V}\left(\Pi_c - \frac{dC_v}{dt} - \psi_g\right)$          (eq.20)
$\emptyset_{sP} = \frac{P_p}{C_V}\left(\Pi_c - \frac{dC_v}{dt} - \psi_s\right)$          (eq.21)
$\emptyset_{gN} = \frac{N_v}{C_V}\left(\Pi_c - \frac{dC_v}{dt} - \psi_g\right)$          (eq.22)
$\emptyset_{sN} = \frac{N_v}{C_V}\left(\Pi_c - \frac{dC_v}{dt} - \psi_s\right)$          (eq.23)
$\emptyset_g = \begin{cases} \emptyset_{gP} & \emptyset_{gP} \times \frac{C_V}{P_p} > \emptyset_{gN} \times \frac{C_V}{N_v} \\ \emptyset_{gN} & \emptyset_{gN} \times \frac{C_V}{N_v} > \emptyset_{gP} \times \frac{C_V}{P_p} \end{cases}$          (eq.24)
$\emptyset_s = \begin{cases} \emptyset_{sP} & \emptyset_{sP} \times \frac{C_V}{P_p} > \emptyset_{sN} \times \frac{C_V}{N_v} \\ \emptyset_{sN} & \emptyset_{sN} \times \frac{C_V}{N_v} > \emptyset_{sP} \times \frac{C_V}{P_p} \end{cases}$          (eq.25)
where $\frac{P_p}{C_V}$ is the inverse of whole plant C:P ratio, $\frac{N_v}{C_V}$ is inverse plant C:N ratio, $\frac{dC_v}{dt}$ is rate of change in plant C
(see Clark *et al.,* (2011) for more detail), $\Pi_c$ is nutrient-unlimited, or potential, NPP (kg C m$^{-2}$ yr$^{-1}$), $\psi_g$ is excess
C due to either P or N limitation for plant growth (kg C m$^{-2}$ yr$^{-1}$) and $\psi_s$ is excess C due to either P or N
limitation for vegetation spreading (kg C m$^{-2}$ yr$^{-1}$).

Equations 20 and 22 are solved by first setting $\psi_g = 0.0$ to find the total plant P (eq. 20) and N demand (eq.22).
If the P and N demand for growth are less than the available P and N and fractional coverage ($\lambda$) (NPP fraction
used for fractional cover increment; for detail see Wiltshire *et al.,* (2021)) at the considered timestep $\Delta t$ then
there is no limitation to growth ($i.e. \emptyset_{gP} < \frac{(1-\lambda)P_{avail}}{\Delta t}$; $\emptyset_{gN} < \frac{(1-\lambda)N_{avail}}{\Delta t}$). Where there is limited P and/or N
availability, the uptake equals the available P and N ($\emptyset_{gP} = \frac{(1-\lambda)P_{avail}}{\Delta t}$; $\emptyset_{gN} = \frac{(1-\lambda)N_{avail}}{\Delta t}$), and the plant
growth which cannot be achieved due to nutrient constraints will be deducted from potential NPP, here termed
excess C term ($\psi_g$), to give an actual NPP. Following Wiltshire et al., 2021, we assume excess C is respired by
the plant.

Similarly, in order to estimate the P and N demand for spreading (eq. 21 and 23), initially the excess C from
spreading is set to 0.0 ($\psi_s = 0.0$), i.e under the assumption that there is no nutrient limitation. If the P and N
demand for spreading are lower than the available P and N and fractional coverage ($\lambda$) ($\emptyset_{sP} <$
$\frac{(1-\lambda)P_{avail}}{\Delta t}$; $\emptyset_{sN} < \frac{(1-\lambda)N_{avail}}{\Delta t}$), then there is no limitation on spreading and in case of limited P and N
availability, the uptake equals the available P and N ($\emptyset_{sP} = \frac{(1-\lambda)P_{avail}}{\Delta t}$; $\emptyset_{sN} = \frac{(1-\lambda)N_{avail}}{\Delta t}$), and the excess C
for spread ($\psi_s$) is subtracted from potential NPP.

Plant P uptake ($F_p^{up}$) (arrow a in Fig 1) is estimated based on the P demand for growth and spreading ($\emptyset_t$) and
the root uptake capacity ($u^{max}$) (kg P kg$^{-1}$ C yr$^{-1}$), as follows:

$F_p^{up}{}_n = \begin{cases} \emptyset_t & \emptyset_t \leq u^{max} \\ u^{max} & \emptyset_t > u^{max} \end{cases}$          (eq.26)

Plant P uptake ($F_p^{up}$) varies spatially depending on the root uptake capacity ($u^{max}$) followed by Goll *et al.,*
(2017). Therefore, in regions with limited P supply, the plant P uptake is limited to the $u^{max}$ and consequently
impacts the excess C and BP.
The root uptake capacity depends on the maximum root uptake capacity ($v_{max}$) (kg P kg$^{-1}$ C yr$^{-1}$), root depth
($d_{root}$), the concentration of inorganic P at different soil depths ($P_{in}$), and a half saturation term at which half of
the maximum uptake capacity is reached using inorganic P at different soil depths ($P_{in}$), a scaling uptake ratio
($K_p$) (μmol P l$^{-1}$), unit conversion ($C_f$) (1 kg P$^{-1}$), and soil moisture (θ) (l m$^{-2}$), as follows:

$u^{max} = v_{max} \times d_{root} \times \sum_{n=1}^{N} P_{in_n} \times \left(\frac{1}{\sum_{n=1}^{N} P_{in_n} + c_f \times K_p \times \theta_n}\right)$          (eq.27)

Description of the litter production of P ($F_{P_n}^{lit}$) (arrow b in Fig 1) follows JULES-CN as in Wiltshire *et al.*,
(2021) and is calculated based on the litter flux of C (kg C m$^{-2}$ yr$^{-1}$) using leaf, root and wood turnovers (yr$^{-1}$),
and through the vegetation dynamics due to large-scale disturbance and litter production density, as follows:

$$F_{P_n}^{lit} = \left(\frac{(1-k_{leaf})\gamma_{leaf}C_{leaf}}{C:P_{leaf}}\right) + \left(\frac{(1-k_{root})\gamma_{root}C_{root}}{C:P_{root}}\right) + \left(\frac{\gamma_{wood}C_{wood}}{C:P_{stem}}\right) \qquad \text{(eq.28)}$$
where $\lambda$ is the leaf, root and stem re-translocation (at daily timestep) coefficient (Zaehle and Friend, 2010; Clark
*et al.*, 2011) and the related $C:P$ ratios for P fraction and $\gamma$ is a temperature dependent turnover rate representing
the phenological state (Clark *et al.*, 2011).
The decomposition of litter ($dec^{lit}$) (arrow c in Fig 1) depends on soil respiration ($R$) (kg C m$^{-2}$ yr$^{-1}$), the litter
C:P ratio ($C:P_{lit}$) at each soil layer (n) as follows:

$$dec_P{}^{lit} = \frac{\sum_{n=1}^{N} R_n}{C:P_{lit}} \qquad \text{(eq.29)}$$
where the $C:P_{lit}$ is calculated based on litter C pool (DPM and RPM) ($lit^C$) (kg C m$^{-2}$ yr$^{-1}$) and litter P pool
($P_{O_l}$) as follows:

$$C:P_{lit} = \frac{\sum_{n=1}^{N} lit_n{}^C}{P_{O_{l_n}}} \qquad \text{(eq.30)}$$
The mineralized ($F_{minl_P}$) (arrow d in Fig 1) and immobilized ($F_{immob_P}$) (arrow e in Fig 1) P fluxes are
calculated based on C mineralization and immobilization, C:P ratios of plant (i) (DPM/RPM) ($C:P_{plant}$) and
soil (HUM/BIO) ($C:P_{soil}$), soil pool potential respiration ($R_{POT_i}$) (kg C m$^{-2}$ yr$^{-1}$) and the respiration partitioning
fraction (*resp_frac*) as follows:

$$F_{minl_{P_n}} = \frac{\sum_{n=1}^{N} R_{POT_{i,n}}}{C:P_{plant}} \qquad \text{(eq.31)}$$

$$F_{immob_{P_n}} = \frac{\sum_{n=1}^{N} R_{i,n} \times resp\_frac}{C:P_{soil}} \qquad \text{(eq.32)}$$
The soil respiration from each soil layer ($R_{i,n}$) is estimated from potential soil respiration ($R_{POT_{i,n}}$) for the
DPM, RPM pools and the litter decomposition rate modifier ($F_{P_n}$) as follows:

$$R_{i,n} = R_{POT_{i,n}} \times F_{P_n} \qquad \text{(eq.33)}$$
where the description of $F_{P_n}$ for P pools ($F_{P_{P_n}}$) follows Wang *et al.*,(2007) and is estimated based on the soil
pool (BIO/HUM) mineralization ($minl_{P-BIO_n}$, $minl_{P-HUM_n}$) and immobilization ($immob_{P-BIO_n}$,
$immob_{P-HUM_n}$) (in kg P m$^{-2}$ yr$^{-1}$), soil inorganic P ($P_{inorg_n}$) (in kg P m$^{-2}$), and litter pools (DPM/RPM) demand
(in kg P m$^{-2}$ yr$^{-1}$) as follows:

$$F_{P_{P_n}} = \frac{(minl_{P-BIO_n} + minl_{P-HUM_n} - immob_{P-BIO_n} - immob_{P-HUM_n}) + P_{inorg_n}}{DEM_{DPM_n} + DEM_{RPM_n}} \qquad \text{(eq.34)}$$
The net demand associated with decomposition of litter pools ($DEM_{k,n}$) represents the P required by microbes
which convert DPM and RPM into BIO and HUM. The limitation due to insufficient P availability is estimated
based on the potential mineralization ($minl_{p-pot}$) and immobilization ($immob_{p-pot}$) (in kg P m$^{-2}$ yr$^{-1}$) of pools
(k) as follows:

$$DEM_{k,n} = immob_{p-pot,k} - minl_{p-pot,k} \qquad \text{(eq.35)}$$
The $F_{P_n}$ estimated for N pools ($F_{P_{N_n}}$) follows the same formulation as P (see Wiltshire *et al.*, 2021 for further
details) and the $F_{P_n}$ is estimated based on a higher rate modifier between N and P as follows:

$$F_{P_n} = \begin{cases} F_{P_{P_n}} & F_{P_{P_n}} > F_{P_{N_n}} \\ F_{P_{N_n}} & F_{P_{N_n}} > F_{P_{P_n}} \end{cases} \qquad \text{(eq.36)}$$

Description of the fluxes of adsorption ($F_{P_{in_n}}^{sorp}$) (arrow e in Fig 1) and desorption ($F_{P_{in_n}}^{desorp}$) (arrow f in Fig
1) of inorganic P in kg P m$^{-2}$ yr$^1$ follow Wang $et\ al.$, (2010) and are calculated based on soil inorganic ($P_{in_n}$) and
sorbed inorganic ($P_{inorg-sorbed_n}$) P pools and inorganic adsorption ($K_{sorp-in}$), desorption ($K_{desorp-in}$)
coefficients (kg P m$^{-2}$ yr$^{-1}$) and maximum sorbed inorganic ($P_{in-max}$) (kg P m$^{-2}$) as follows:

$$F_{P_{in_n}}^{sorp} = P_{in_n} \times K_{sorp-in} \times \frac{\left(P_{in-max_n} - P_{inorg-sorbed_n}\right)}{P_{in-max_n}} \qquad \text{(eq.37)}$$

$$F_{P_{in_n}}^{desorp} = P_{inorg-sorbed_n} \times K_{desorp-in} \qquad \text{(eq.38)}$$

Description of the occluded inorganic P flux ($F_{P_n}^{occ}$) (arrow g in Fig 1) follows Wang $et\ al.$, (2007) and Hou $et$
$al.$, (2019) and is calculated based on sorbed inorganic P pool and P occlusion rate ($K_{occ}$) (kg P m$^{-2}$ yr$^{-1}$) as
follows:

$$F_{P_n}^{occ} = P_{inorg-sorbed_n} \times K_{occ} \qquad \text{(eq.39)}$$

Description of the fluxes of adsorption ($F_{P_{O_{S_n}}}^{sorp}$) (arrow h in Fig 1) and desorption ($F_{P_{O_{S_n}}}^{desorp}$) (arrow i in Fig
1) of organic P follow Wang $et\ al.$, (2010) are calculated based on soil organic and sorbed organic P pools and
organic adsorption ($K_{sorp-or}$) (kg P m$^{-2}$ yr$^{-1}$), desorption ($K_{desorp-or}$) coefficients (kg P m$^{-2}$ yr$^{-1}$) and maximum
sorbed organic ($P_{org-max}$) (which corresponds to the sorbed soil P saturation, thus modifying the sorption rate
respectively) (kg P m$^{-2}$) as follows:

$$F_{P_{O_{S_n}}}^{sorp} = P_{O_{S_n}} \times K_{sorp-or} \times \frac{\left(P_{or-max_n} - P_{org-sorbed_n}\right)}{P_{or-max_n}} \qquad \text{(eq.40)}$$

$$F_{P_{O_{S_n}}}^{desorp} = P_{org-sorbed_n} \times K_{desorp-or} \qquad \text{(eq.41)}$$

Description of the occluded organic P flux ($F_{P_n}^{or-occ}$) (kg P m$^{-2}$ yr$^{-1}$) (arrow j in Fig 1) follows Wang $et\ al.$,
(2007) and Hou $et\ al.$, (2019) is calculated based on sorbed organic P pool ($P_{org-sorbed_n}$) and P occlude rate
($K_{occ}$) (kg P m$^{-2}$ yr$^{-1}$) as follows:

$$F_{P_n}^{or-occ} = P_{org-sorbed_n} \times K_{occ} \qquad \text{(eq.42)}$$

Description of the P flux from weathered parent material ($F_{P_n}^{w}$) (arrow k in Fig 1) follows Wang $et\ al.$, (2007)
and is calculated based on amount of P in the parent material ($P_{pm}$) and P weathering rate ($K_w$) (kg P m$^{-2}$ yr$^{-1}$) as
follows:

$$F_{P_n}^{w} = P_{pm_n} \times K_w \qquad \text{(eq.43)}$$

Description of P diffusion between soil layers ($F_{D_n}$) expressed in (kg P m$^{-2}$ yr$^{-1}$) (arrow l in Fig 1) follows Goll
$et\ al.$, (2017) and is calculated following Fick's second law and it is a function of the diffusion coefficient ($Dz$)
in m$^2$ s$^{-1}$, the concentration of inorganic P at different soil depths ($P_{in}$) in kg P m$^{-2}$, the distance ($z$) between the
midpoints of soil layers in metres and seconds to year unit conversion ($Yr$):

$$F_{D_n} = \frac{\partial}{\partial z}\left(D_{z_n} \frac{\partial P_{S_n}}{\partial z}\right) \times Yr \qquad \text{(eq.44)}$$





**Table 1.** Model variables

| Variable | Unit | Definition |
| --- | --- | --- |
| $\psi$ | kg C m$^{-2}$ yr$^{-1}$ | Excess C flux |
| $\emptyset$ | kg P m$^{-2}$ yr$^{-1}$ | Plant demand for uptake |
| $\Pi_c$ | kg C m$^{-2}$ yr$^{-1}$ | Potential NPP |
| $u^{max}$ | kg P kg$^{-1}$ C yr$^{-1}$ | Root uptake capacity |
| $DEM$ | kg P m$^{-2}$ yr$^{-1}$ | Plant pool P associated decomposition demand |
| $dec_P{}^{lit}$ | kg P m$^{-2}$ yr$^{-1}$ | Litter decomposition |
| $F_D$ | kg P m$^{-2}$ yr$^{-1}$ | Plant diffusion flux |
| $F_P$ | - | Plant litter decomposition rate modifier |
| $F_p{}^{lit}$ | kg P m$^{-2}$ yr$^{-1}$ | Plant litter flux |
| $F_p{}^{up}$ | kg P m$^{-2}$ yr$^{-1}$ | Plant uptake |
| $F_{P_{O_S}}{}^{sorp}$ | kg P m$^{-2}$ yr$^{-1}$ | Sorbed organic P flux |
| $F_{P_{in}}{}^{sorp}$ | kg P m$^{-2}$ yr$^{-1}$ | Sorbed inorganic P flux |
| $F_{P_{O_S}}{}^{desorp}$ | kg P m$^{-2}$ yr$^{-1}$ | Desorbed organic P flux |
| $F_{P_{in}}{}^{desorp}$ | kg P m$^{-2}$ yr$^{-1}$ | Desorbed inorganic P flux |
| $F_p{}^{occ}$ | kg P m$^{-2}$ yr$^{-1}$ | Occluded inorganic P flux |
| $F_p{}^{or\text{-}occ}$ | kg P m$^{-2}$ yr$^{-1}$ | Occluded organic P flux |
| $F_p{}^{w}$ | kg P m$^{-2}$ yr$^{-1}$ | Weathered P flux |
| $F_{immob\,P}$ | kg P m$^{-2}$ yr$^{-1}$ | Immobilized P flux |
| $lit_C$ | kg C m$^{-2}$ yr$^{-1}$ | C litter flux |
| $lit_{frac}$ | - | Litter fraction |
| $lit_{leaf}$ | kg C m$^{-2}$ yr$^{-1}$ | Leaf litter flux |
| $lit_{root}$ | kg C m$^{-2}$ yr$^{-1}$ | Root litter flux |
| $lit_{wood}$ | kg C m$^{-2}$ yr$^{-1}$ | Woody litter flux |
| $F_{minl\,P}$ | kg P m$^{-2}$ yr$^{-1}$ | Mineralized P flux |
| $P_p$ | kg P m$^{-2}$ | Plant P pool |
| $P_{O_l}$ | kg P m$^{-2}$ | Litter organic pool |
| $P_{O_s}$ | kg P m$^{-2}$ | Soil organic pool |
| $P_{in}$ | kg P m$^{-2}$ | Soil inorganic pool |
| $P_{inorg-sorp}$ | kg P m$^{-2}$ | Soil inorganic sorbed pool |
| $P_{org-sorp}$ | kg P m$^{-2}$ | Soil organic sorbed pool |
| $P_{occ}$ | kg P m$^{-2}$ | Soil occluded pool |
| $P_{pm}$ | kg P m$^{-2}$ | Parent material pool |
| $R$ | kg C m$^{-2}$ yr$^{-1}$ | Total respiration |
| $R_{POT}$ | kg C m$^{-2}$ yr$^{-1}$ | Total potential respiration |
| $R^s$ | kg C m$^{-2}$ yr$^{-1}$ | Soil respiration |
| $R_d$ | kg C m$^{-2}$ yr$^{-1}$ | Leaf dark respiration |
| $T_{ref}$ | K | Soil reference temperature |
| $T_s$ | K | Soil temperature |
| Veg$_c$ | kg C m$^{-2}$ | Sum of biomass |
| z | m | Soil depth |

**Table 2.** P Model parameters

| Parameter | Value | Unit | Eq. | Description | Source |
|---|---|---|---|---|---|
| | | | **C and N related** | | |
| $\alpha$ | 0.25 | - | 6 | Plant type material ratio | (Clark *et al.*, 2011) |
| $a_{wl}$ | 1.204 | kg C m$^{-2}$ | 50 | Allometric coefficient | calibrated |
| $\sigma_l$ | 0.0375 | kg C m$^{-2}$ per unit LAI | 48 | Specific leaf density | Clark *et al.*, 2011 |
| $b_{wl}$ | 1.667 | - | 50 | Allometric exponent. | Clark *et al.*, 2011 |
| $f_{dr}$ | 0.005 | - | 47 | Respiration scale factor | Calibrated |
| $resp\_frac$ | 0.25 | - | 32 | Respiration fraction | (Clark *et al.*, 2011) |
| $k_{leaf}$ | 0.5 | - | 28 | Leaf N re-translocation coefficient | (Zaehle and Friend, 2010) |
| $k_{root}$ | 0.2 | - | 28 | Root N re-translocation coefficient | (Zaehle and Friend, 2010) |
| $d_{root}$ | 3.0 | - | 27 | Root fraction in each soil layer | (Clark *et al.*, 2011) |
| $v_{int}$ | 7.21 | μmol CO$_2$ m$^{-2}$ s$^{-1}$ | 45 | Intercept in the linear regression between $V_{cmax}$ and $N_{area}$ | Calibrated (Clark *et al.*, 2011) |
| $v_{sl}$ | 19.22 | μmol CO$_2$ gN$^{-1}$ s-1 | 45 | Slope in the linear regression between $V_{cmax}$ and $N_{area}$ | Calibrated (Clark *et al.*, 2011) |
| $LMA$ | 131.571852 | g m-2 | 45 | Observed Leaf Mass per Area | Study site |
| $Leaf\ N$ | 1.79007596 | g g-1 | 45, 46 | Foliar N concentrations | Study site |
| | | | **P related** | | |
| $C{:}P_{soil}$ | 1299.6 | - | 32 | Soil C:P ratio | (Fleischer *et al.*, 2019) |
| $v_{max}$ | 0.0007 | kg P kg$^{-1}$ C yr$^{-1}$ | 27 | Maximum root uptake capacity | Calibrated (Goll *et al.*, 2017) |
| $P$ | 0.7083062 | g kg$^{-1}$ | 46 | Foliar P concentrations | Study site |
| $c_f$ | 3.1×10$^{-5}$ | 1 kg P$^{-1}$ | 27 | Conversion factor | (Goll *et al.*, 2017) |
| $D_z$ | 0.001 | m$^2$ s$^{-1}$ | 44 | Diffusion coefficient | (Burke *et al*, 2017) |
| $K_{occ}$ | 1.2×10$^{-5}$ | yr$^{-1}$ | 39, 42 | P occlusion rate | (Yang *et al.*, 2014) |
| $K_p$ | 3.0 | kg P l$^{-1}$ | 27 | Scaling uptake ratio | Calibrated |
| $K_{sorp-in}$ | 0.0054 | kg P m$^{-2}$ yr$^{-1}$ | 37 | Inorganic P adsorption coefficient | Calibrated (Hou *et al.*, 2019) |
| $K_{sorp-or}$ | 0.00054 | kg P m$^{-2}$ yr$^{-1}$ | 40 | Organic P adsorption coefficient | Calibrated |
| $K_{in-max}$ | 0.0075 | kg P m$^{-2}$ yr$^{-1}$ | 37 | Maximum sorbed inorganic P | Study site |
| $K_{or-max}$ | 0.0042 | kg P m$^{-2}$ yr$^{-1}$ | 40 | Maximum sorbed organic P | Study site |
| $K_w$ | 3×10$^{-6}$ | kg P m$^{-2}$ yr$^{-1}$ | 43 | P weathering rate | (Wang *et al.*, 2010) |

**2.3 Study sites**
This study primarily uses data from two nearby sites in Central Amazon in Manaus, Brazil. The main site from
here on termed *study site* (2°35′′21.08′′ S, 60°06′′53.63′′ W) (Lugli *et al.*, 2020) is for model development and
evaluation. The second site is the Manaus K34 flux site (2°36′′32.67′′ S, 60°12′′33.48′′ W) which provides
meteorological station data for running the model but also provides data for model evaluation. Our *study site* is
the main lowland tropical forest site maintained by the National Institute for Amazon Research (INPA).
Research at this site focuses on projects, combining experimental approaches (Keller *et al.*, 2004; Malhi *et al.*,
2009) with modelling (Lapola and Norby, 2014). We use detailed novel soil and plant P pool data from the *study*
*site* (Lugli *et al.*, 2020, 2021) for model parameterisation and calibration and carbon stock data for model
validation. The *study site* has a very similar forest, geomorphology, soil chemistry and species composition to
the well-known and studied K34 flux site (Araújo *et al.*, 2002). The average reported annual precipitation is
2431 (mm yr$^{-1}$), with a monthly range of 95 to 304 (mm month$^{-1}$), and averaged temperature is 26°C (Araújo *et*
*al.*, 2002). Soil type at this site is Geric Ferrosol with a high clay content and weathering activities (Malhi *et al.*,
471 2004).

In addition to the *study site* we use data from other P limited locations from the Amazon, Panama and Hawaii
(Table 3) for model evaluation. Old-growth forest sites in the Amazon are located across a fertility gradient
from west to east (AGP-01, SA3, CAX,) with detailed C cycle measurements available (Aragão *et al.*, (2009)).
The site in Panama is located in the Gigante Peninsula in the Barro Colorado Nature Reserve and is a 200 year
old semi-deciduous rainforest (Wright *et al.*, 2011) growing on Oxisols developed on Miocene basalt (Dieter,
Elsenbeer and Turner, 2010) with the topsoil a dominant clay texture (Turner and Condron, 2013). It is the
location of a long term running nutrient fertilization experiment since 1998 (Mirabello *et al.*, 2013). The site in
Hawaii (Hawaii Kokee) is a P limited chronosequence that developed on the 4 million year old oxisols soil
(Vitousek, 2004) and has a long term fertilization experiment. Site information is provided in Table 3.
**Table 3.** Test sites name, location and climate characterises.

| Site | Name | Location | | Climate | |
|---|---|---|---|---|---|
| | | Lat. | Lon. | Rainfall (mm yr$^{-1}$) | Temperature($^\circ$C) |
| *Study site* | AFEX project | -2.58 | -60.11 | 2431 | 26 |
| **AGP-01** | Agua pudre plot E | -3.72 | -70.3 | 2723 | 25.5 |
| **CAX** | Caxiuanã flux tower site | -1.72 | -51.5 | 2314 | 26.9 |
| **SA3** | Tapajós flux tower site | -2.5 | -55 | 1968 | 26.1 |
| **Gig. Pen.** | Gigante peninsula (control data) | -9.1 | -79.84 | 2600 | 26 |
| **Hawaii K.** | Hawaii Kokee (control data) | 22.13 | -159.62 | 2500 | 16 |

**2.4 Model parameterisation, calibration and evaluation at study site**
We use observations from the four control plots of the study site to parameterise, calibrate and evaluate different
processes in JULES (Table 4). The observations were collected at 4 soil depths and processed using the Hedley
sequential fractionation (Hedley, Stewart and Chauhan, 1982; Quesada *et al.*, 2010). Observed Leaf Mass per
Area (LMA), leaf N and leaf P estimated from fresh leaves were used as input parameters to JULES to estimate
photosynthetic capacity and respiration parameters. JULES vn5.5 (JULES CN in this study) estimates $V_{cmax}$
($\mu$mol m$^{-2}$ s$^{-2}$) based on Kattge et al. (2009) using foliar N concentrations in area basis ($nleaf$), as follows:
$$V_{cmax} = v_{int} + v_{sl} * nleaf \tag{eq.45}$$
where $v_{int}$ is the estimated intercept and $v_{sl}$ is the slope of the linear regression derived for the $V_{cmax}$ estimation.
We incorporated an additional P dependency on the estimation of $V_{cmax}$ following Walker et al. (2014) as
follows:
$$\ln(V_{cmax}) = 3.946 + 0.921\ln(N) + 0.121\ln(P) + 0.282\ln(N)\ln(P) \tag{eq.46}$$
Where N and P are foliar concentrations in area basis.
Implementation of eq. 46 resulted in higher $V_{cmax}$ than in the original version of JULES. A higher $V_{cmax}$ predicted
higher leaf and plant respiration (eq.47). Constrained by observations of NPP and plant respiration at the study
site, we modified one of the most uncertain parameters in the description of plant respiration ($f_{dr}$) (eq.47) which
is the scale factor for leaf dark respiration ($R_d$) as follows:
$$R_d = f_{dr} V_{cmax} \tag{eq.47}$$
The default value is 0.01 (Clark *et al.*, 2011), and for JULES-CNP simulations at our study site it was modified
to 0.005.
Observations of aboveground biomass were used to calibrate the non PFT dependent allometric relationships in
JULES (Clark et al 2011) (eq 48-50) for leaf, root and wood C. Specifically, the $a_{wl}$ parameter (eq 50) was
modified from 0.65 to 1.204 to match better tropical forest allometry:
$$C_{leaf} = \sigma_l L_b \tag{eq.48}$$
$$C_{root} = C_{leaf} \tag{eq.49}$$
$$C_{stem} = a_{wl} \, L_b{}^{b_{wl}} \qquad\qquad\qquad\qquad\qquad\qquad\qquad\qquad\qquad\qquad\qquad\qquad\text{(eq.50)}$$

Where $\sigma_l$ is specific leaf density (kg C m$^{-2}$ per unit LAI), $L_b$ is balanced (or seasonal maximum) leaf area index
(m$^2$ m$^{-2}$), $a_{wl}$ is allometric coefficient (kg C m$^{-2}$) and $b_{wl}$ is the allometric exponent.
Note that JULES-CNP uses the C3 and C4 photosynthesis model from Collatz et al., 1991; Collatz, Ribas-Carbo
and Berry, 1992, which does not include estimation of J$_{max}$.

JULES-CNP has fixed stoichiometry and C:P ratios of leaf and root (measured), and wood (estimated from fresh
coarse wood (Lugli, 2013)) which were taken from the *study site* and prescribed in JULES to simulate P
dynamics in the plant. The following belowground data were used to represent various soil P pools: Resin and
bicarbonate inorganic P (inorganic P: $P_{in}$), organic bicarbonate P (organic P: $P_{O_S}$), NaOH organic P (sorbed
organic P: $P_{org-sorp}$), NaOH inorganic P (sorbed inorganic P: $P_{inorg-sorp}$), residual P (occluded P: $P_{occ}$) and
HCL P (parent material P: $P_{pm}$) (Table 4). The measurements were collected between 2017 and 2018 in control
plots. All measurements were conducted in four soil layers (0-5 ,5-10, 10-20, 20-30 cm). However, to be
consistent with the JULES model soil layer discretization scheme, we defined 4 soil layers (0-10 cm, 10-30 cm,
30-100 cm and 100-300 cm) and we used the average between 0 and 30 cm to compare against the measurement
from the same depth for model evaluation.
Vegetation C stocks were derived based on tree diameter measurements at breast height, that are linked to
allometric equations and wood density databases to estimate the C stored in each individual tree, and then scaled
to the plot (Chave *et al.*, 2014).

The organic and inorganic soil P was assumed to be always at equilibrium with the relative sorbed pools (Wang,
Law and Pak, 2010). Thus, in order to cap P sorption and uptake capacity, the maximum sorption capacities
($P_{in-max_n}$, $P_{or-max_n}$, eq.37 and 39) (adopted from (Wang, Houlton and Field, 2007)) were prescribed using
maximum observed sorbed inorganic and organic P. Hence, the maximum sorption capacity defines the
equilibrium state of sorbed and free-soil P. Moreover, despite the initial representation of the parent material
pool in JULES and its depletion through weathering (eq. 43), as the magnitude of changes in the occluded and
parent material pools are insignificant over a short-term (20 years) simulation period (Vitousek *et al.*, 1997),
these two pools were prescribed using observations. Remaining parameters used to describe soil P fluxes (eqn.s
27-44) were prescribed using values from the literature (Table 4).

We used a combination of data from the *study site* and the nearby K34 site for model evaluation of C fluxes
(GPP, NPP) and C pools (soil and vegetation C, leaf, root and wood C) with no calibration of plant and soil
organic and soil inorganic P pools included (Table 4).

**Table 4.** Observations from study site (taken during 2017-2018) and from Manaus site K34 used for model parameterisation
and evaluation

| Process | Variables | Purpose of use | Reference and site |
|---|---|---|---|
| C associated | GPP | Evaluation | Fleischer et al., 2019, K34 |
| | NPP | Evaluation | Fleischer et al., 2019, K34 |
| | Soil C | Evaluation | Malhi et al., 2009, K34 |
| | CUE | Evaluation | Malhi et al., 2009, K34 |
| | Veg C | Evaluation | Study site |
| | Leaf C | Evaluation | Study site |
| | Wood C | Evaluation | Study site |
| | Root C | Evaluation | Study site |
| | LAI | Initialisation | Study site |
| | LMA | Parameterisation | Study site |
| P associated | Resin | Evaluation | Study site |
| | Pi Bic | Evaluation | Study site |
| | Po Bic | Evaluation | Study site |
| | Po NaOH | Calibration | Study site |
| | Pi NaOH | Calibration | Study site |
| | P residual | Parameterisation | Study site |
| | P HCL | Parameterisation | Study site |
| | Leaf N | Parameterisation | Study site |
| | Leaf P | Parameterisation | Study site |
| | Root P | Parameterisation | Study site |
| | Plant C:P ratio | Parameterisation | Study site |

### 2.4.1  Model parameterisation and evaluation at test sites

JULES-CNP was parameterised using reported C:P ratios and maximum sorbed organic and inorganic P for each test site (Table 5) as follows:

**Table 5.** Additional test sites data used for model parameterisation

|  | AGP-01[a,b] | CAX [a,b] | SA3 [a,b] | Gig. Pen. [c] | Hawaii K. [b,d] |
|---|---|---|---|---|---|
| $Leaf_{C:P}$ | 600 | 600 | 600 | 700 | 691.5 |
| $Root_{C:P}$ | 1000 | 1000 | 1000 | 1750 | 1100 |
| $Wood_{C:P}$ | 3000 | 3000 | 3000 | 5500 | 5937.5 |
| $Soil_{C:P}$ | 2000 | 2000 | 2000 | 800 | 2000 |
| $K_{or-max}$ | 0.001 | 0.001 | 0.001 | 0.0033 | 0.001 |
| $K_{in-max}$ | 0.001 | 0.001 | 0.001 | 0.0185 | 0.001 |

[a]C:P ratios from Wang, Law and Pak, 2010 and [b]maximum sorbed P capacities from Yang *et al.*, 2014.
[c]Mirabello *et al.*, 2013 [d] C:P ratios from Vitousek, 2004

Model evaluation at test sites, was performed using observed NPP, litterfall, autotropic respiration, biomass and soil C pools taken from different sources. We used NPP and litterfall for the Amazon sites from Aragão *et al.*, (2009) and for Gigante Peninsula from Chave *et al.*, (2003), Hawaii Kokee NPP as reported in Goll *et al.*, (2017) and litterfall as reported in (Yang *et al.*, 2014). Plant respiration was only available at two of Amazon sites (AGP and CAX) (Malhi *et al.*, 2009). The biomass and soil C pools for Amazon sites (CAX and SA3) are taken from  Malhi *et al.*, (2009) and biomass from AGP is taken from Jiménez *et al.*, (2009). The Gigante Peninsula biomass is taken from Chave *et al.*, (2003), soil C from Turner et al., (2015), and the Hawaii Kokee C pools are taking as reported in Yang et al., (2014).

### 2.5  JULES simulations

JULES was first applied at the K34 flux tower site using observed meteorological forcing data from 1999-2019 (Fleisher et a 2019) at half hourly resolution. The following meteorological variables are needed to drive JULES (model inputs) (Best *et al.*, 2011): atmospheric specific humidity (kg kg$^{-1}$), atmospheric temperature (K), air pressure at the surface (Pa), short and longwave radiation at the surface (W m$^{-2}$), wind speed (m s$^{-1}$) and total precipitation (kg m$^{-2}$ s$^{-1}$). Furthermore, the averaged measured LAI from study site was used to initialise the vegetation phenology module, but was allowed to vary in subsequent prognostic calculations. Soil organic and inorganic sorbed P pools were initialised with study site observations. The JULES-CNP simulations were initialized following the same methodology as in Fleischer et al., (2019), by the spin-up from1850 resulted in equilibrium state (Figure S1). The spin up was performed separately for three versions of JULES (C/CN/CNP) following the same procedure. Furthermore, the transient run was performed for the period 1851-1998 using time-varying CO$_2$ and N deposition fields. Finally, for the extended simulation period (1999-2019) two runs were performed, the first with ambient the second elevated CO$_2$ concentrations.

We evaluate the impact of including a P cycle in JULES using three model configurations (JULES C, CN and CNP). We apply JULES in all three configurations using present day climate under both ambient CO$_2$ and eCO$_2$. Ambient and eCO$_2$ were prescribed following Fleischer *et al.*, (2019), with present-day CO$_2$ based on global monitoring stations, and an abrupt (step) increase in atmospheric CO$_2$ of +200 ppm on the onset of the transient period (i.e., 1999). However, the comparison period is limited to 2017-18 for which the P measurements are available.

We compare simulated C fluxes (GPP, NPP, litterfall C), C stocks (total vegetation, fine root, leaf, wood, soil) and the CO$_2$ fertilization effect across model configurations. The CO$_2$ fertilization effect $\left(CO2_{fert-eff}\right)$ (eq.51) is calculated based on simulated vegetation C under ambient ($VegC\ (aCO_2)$) and eCO$_2$ ($VegC\ (eCO_2)$) as follows:

$$CO2_{fert-eff} = \frac{(VegC\ (eCO_2) -\ VegC\ (aCO_2)) \times 100}{VegC\ (aCO_2)}$$  (eq.51)

Furthermore, the net biomass increases due to $CO_2$ fertilization effect ($\Delta Cveg$) is estimated as follows:
$\Delta C_{veg} = \Delta BP - \Delta litterfall\ C$ (eq.52)
We studied the Water Use Efficiency (WUE) (eq. 53) at half-hourly timestep, then aggregated per month as one
of the main indicators of GPP changes (Xiao *et al.*, 2013), and soil moisture content (SMCL), as one of the
main controllers of maximum uptake capacity (eq. 27), in order to better understanding the changes in GPP, P
demand and uptake as well as excess C fluxes.
$WUE = GPP/Transpiration$ (eq.53)
Moreover, we also estimated the Carbon Use Efficiency (CUE) as an indicator of the required C for the growth
(Bradford and Crowther, 2013) as follows:
$CUE = BP/GPP$ (eq.54)
We use JULES-CNP to evaluate the extent of P limitation under ambient and eCO$_2$ at this rainforest site in
Central Amazon. P limitation is represented by the amount of C that is not used to grow new plant tissue due to
insufficient P in the system (excess C) (eq. 27). The excess C flux is highly dependent on the plant P and the
overall P availability to satisfy demand. We also explore the distribution of the inorganic and organic soil P and
their sorbed fraction within the soil layers and under ambient and eCO$_2$.
**2.5.1 Model sensitivity**
To test the sensitivity of the P and C related processes to individual model P parameters, six sets of simulations
were conducted independently with modified plant C:P stoichiometry (Plant C:P: *SENS1*), P uptake scaling
factor ($K_P$) (Kp: *SENS2*), inorganic (KP_sorb_in: *SENS3*) and organic (KP_sorb_or: *SENS4*) P adsorption
coefficients ($K_{sorp-or}$, $K_{sorp-in}$), and maximum inorganic (KP_sorb_in_max: *SENS5*) and organic
(KP_sorb_or_max: *SENS6*) sorbed P ($K_{or-max}$, $K_{in-max}$). These values were prescribed to vary between ±50%
of the observed values and their effect on C pools (plant and soil C) and fluxes (NPP and excess C), and P pools
(plant, soil, and soil sorbed P) was assessed. As the derived model parameters from measurements have their
own level of uncertainty, we took 50% change to test these parameters at reasonable degree. However, the
occluded and weathered P pools are prescribed for this model application, the occluded and weather P
coefficients (other two P-related model parameters) were not part of sensitivity tests.
Our model evaluation period is limited to years 2017-18 due to the P measurement availability. However, in
order to compare with 15 models studied by Fleischer *et al.*, (2019) we also studied the response of GPP, NPP
and BP to eCO$_2$ for both initial (1999) and 15 years periods (between 1999-2013).
**2.5.1    Simulations at test sites**
To perform JULES (C, CN, CNP) simulations at test sites we extracted the meteorological input data to drive
the model from a global dataset (CRU-NCEP)(Harris *et al.*, 2014) by selecting the closest grid cell to each site
when data were not available for a given site (Table 3). Soil texture ancillaries for each site were extracted from
a global soil data (HWSD) (Nachtergaele *et al.*, 2010). All simulations were initialised from a global JULES-CN
run (Wiltshire *et al.*, 2020) extracted for each site and further spun-up for 2000 years over the 1980-2000 period
for the three versions of JULES (C/CN/CNP). Finally, the transient (2000-2013) run was performed using the
output of the spin-up for each site.

## 3.  Results

### 3.1  Model application under ambient CO₂

#### 3.1.1  Calibration of simulated soil P pools at *study site*

The maximum sorption capacities ($P_{in-max_n}$, $P_{or-max_n}$, eq.37 and 40) were calibrated to the observed P pools. As a result, JULES-CNP could reproduce the measured soil P pools (Fig. 2 and Table 6). Simulated inorganic soil P and sorbed organic and inorganic soil P closely matched the observations (Table 7 and Fig. 2). However, simulated organic soil P overestimates the observations by 60 %.

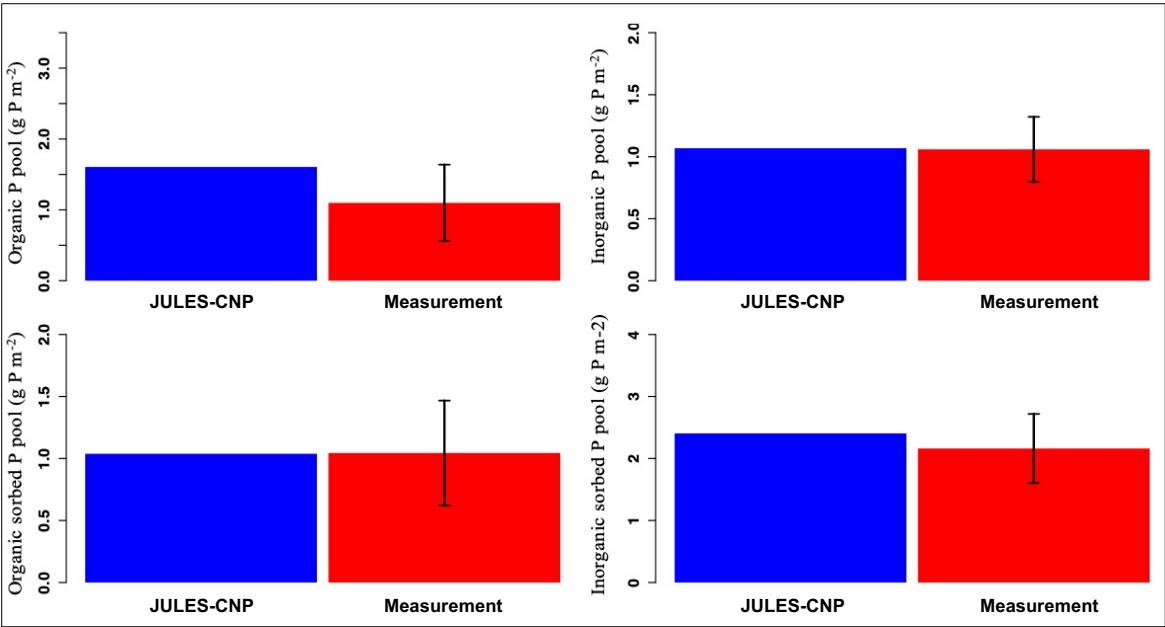

**Figure. 2-** Modelled vs measured soil phosphorus pools under ambient CO₂ (for the soil depth of 0-30cm). Black line represents standard deviation

**Table 6.** Observed and simulated phosphorus pools and fluxes. Occluded and weathered P pools were prescribed using the observed values (between period 2017-18).

| | Phosphorus pools and fluxes | | |
|---|---|---|---|
| | **Measured** | **Modelled Ambient CO₂** | **Modelled Elevated CO₂** |
| **Organic P** (g P m$^{-2}$) | 1.09±0.53 | 1.6 | 1.57 |
| **Inorganic P** (g P m$^{-2}$) | 1.05±0.33 | 1.07 | 0.96 |
| **Sorbed organic P** (g P m$^{-2}$) | 1.04±0.42 | 1.04 | 1.03 |
| **Sorbed inorganic P** (g P m$^{-2}$) | 2.1±0.55 | 2.4 | 2.4 |
| **Occluded P** (g P m$^{-2}$) | 7.98±2.38 | prescribed | prescribed |
| **Weathered P** (g P m$^{-2}$) | 0.59±12 | prescribed | prescribed |
| **Total vegetation P** (g P m$^{-2}$) | 4.15 | 4.66 | 5.11 |
| **Soil P** – 30cm (g P m$^{-2}$) | 13.85 | 14.7 | 14.56 |
| **Total ecosystem P** (g P m$^{-2}$) | - | 35.97 | 35.97 |
| **P litter flux** (g P m$^{-2}$yr$^{-1}$) | 0.3 | 0.28 | 0.29 |

### 3.1.2 Model evaluation

JULES-CNP could reproduce the plant and soil C (Figure 2 and Table 7) and N pools and fluxes (Figure S6 and Table 8) under ambient $CO_2$. Our results show that simulated GPP, is within the range of measurement (3.02 kg C $m^{-2}$ $yr^{-1}$ model vs 3-3.5 kg C $m^{-2}$ $yr^{-1}$ observed, respectively, Table 7).

Simulated NPP, is close to the measured values (NPP: 1.14 - 1.31 observed vs 1.26 modelled kg C $m^{-2}$ $yr^{-1}$) with autotropic respiration (RESP) also closely following the observations (1.98 observed vs 1.81 modelled kg C $m^{-2}$ $yr^{-1}$). Biomass production is estimated as a difference between NPP and the amount of C which is not fixed by plants due to the insufficient P in the system (excess C) (eq. 27). The excess C flux depends on the plant P and the overall P availability to satisfy demand (Table 7). The simulated flux of excess C is 0.3 kg C $m^{-2}$ $yr^{-1}$ under ambient $CO_2$. In JULES-CNP this flux is subtracted from NPP in order to give the BP (eq. 17) (Table 7). Our simulated litterfall overestimates the observations by 32%, however simulated vegetation and its components (fine root, leaf and wood) and soil C stocks match well the observations (Table 7).

**Table 7.** Observed and simulated carbon pools and fluxes with JULES-CNP (between period 2017-18)

| Carbon pools and fluxes | | | |
|---|---|---|---|
| | Measured | Modelled Ambient $CO_2$ | Modelled Elevated $CO_2$ |
| **GPP** (kg C $m^{-2}$ $yr^{-1}$) | 3.0 – 3.5 | 3.06 | 3.9 |
| **NPP$_{pot}$** (kg C $m^{-2}$ $yr^{-1}$) | - | 1.27 | 1.77 |
| **Plant respiration** (kg C $m^{-2}$ $yr^{-1}$) | 1.98 | 1.78 | 2.12 |
| **Excess C flux** (kg C $m^{-2}$ $yr^{-1}$) | - | 0.30 | 0.81 |
| **Biomass Production** (kg C $m^{-2}$ $yr^{-1}$) | 1.14±0.12 | 0.96 | 0.94 |
| **Litter C flux** (kg C $m^{-2}$ $yr^{-1}$) | 0.69±0.15 | 0.91 | 0.83 |
| **Leaf C** (kg C $m^{-2}$) | 0.37±0.2 | 0.38 | 0.40 |
| **Wood C** (kg C $m^{-2}$) | 22.01 | 22.4 | 24.71 |
| **Root C** (kg C $m^{-2}$) | 0.37±0.2 | 0.38 | 0.40 |
| **Vegetation C** (kg C $m^{-2}$) | 22.75±0.3 | 23.16 | 25.52 |
| **Soil C stock** (kg C $m^{-2}$) | 12.7 | 13.2 | 12.71 |
| **LAI** ($m^2$ $m^{-2}$) | 5.6±0.36 | 5.77 | 6.12 |

### 3.1.3 Comparison of JULES C, CN and CNP under ambient $CO_2$ at *study site*

We compare simulated C pools and fluxes from JULES-C, JULES-CN and JULES-CNP (Figure 3). There is no difference between C stocks and fluxes in simulations from JULES C and CN indicating that there is no N limitation at this tropical site in the CN simulations. However, simulated BP and litter flux of C by JULES C/CN are higher than in JULES-CNP but also overestimate the observations (litter flux of JULES C/CN: 1.18, JULES-CNP: 0.91 and obs 0.69 (kg C $m^{-2}$ $yr^{-1}$) and BP of JULES C/CN: 1.24, JULES-CNP: 0.96 and obs 1.14-1.31 (kg C $m^{-2}$ $yr^{-1}$), respectively. By including P cycling in JULES an excess C flux of 0.3 (kg C $m^{-2}$ $yr^{-1}$) is simulated, indicating a 24% P limitation to BP at this site according to JULES-CNP, which represents a 29% decrease in BP compared to JULES-C/CN. Consequently, the total vegetation C stock for models without P inclusion is higher than the CNP version (+3% difference) due to the lack of representation of P limitation. The simulated soil C stock in JULES C and JULES CN is also higher than in the CNP version (JULES C/CN: 13.93 vs. JULES-CNP: 13.18 (kg C $m^{-2}$ $yr^{-1}$)) and higher than the observations. Moreover, CUE in JULES C/CN (eq.54) is higher than observations and JULES-CNP (JULES C/CN: 0.38 vs. JULES-CNP: 0.31, obs: 0.34 ±0.1(dimensionless).

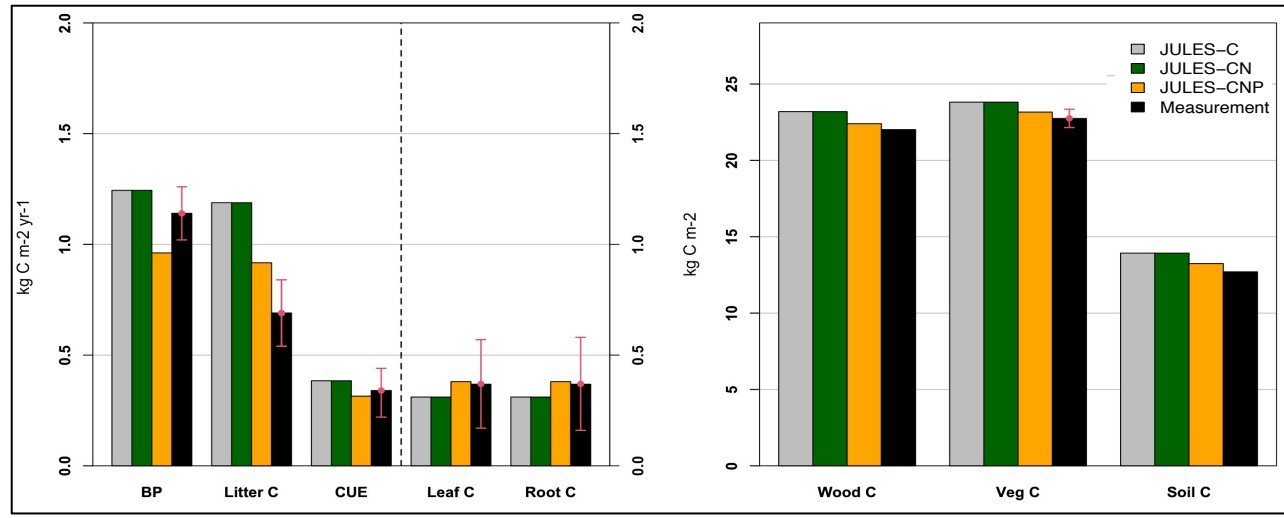

**Figure. 3-** JULES C, CN, CNP modelled vs measured C pools (Leaf, root, wood, Veg and Soil C) (in kg C m$^{-2}$) and fluxes (BP and Litter C) (in kg C m$^{-2}$ yr$^{-1}$) and CUE under ambient $CO_2$. Note that CUE is unitless.

### 3.1.4 Model evaluation at test sites under ambient $CO_2$

Evaluation of JULES C, CN and CNP at five test sites against the observed C pools and fluxes demonstrate that the inclusion of P processes improved the simulation of C pools and fluxes across all test sites (Figure 4). At all Amazon sites JULES C and CN overestimated BP compared to JULES-CNP which estimated lower BP values which were closer to the measurements for AGP (JULES-C: +35%, JULES-CN: +33%; JULES-CNP: +21%), CAX (JULES-C: +45%, JULES-CN: +44%; JULES-CNP: +7%) and SA3 (JULES-C: +27%, JULES-CN: +26%; JULES-CNP: -23%). Moreover, at Gigante Peninsula the C and CN versions overestimated BP (+42% and +40%, respectively), and CNP slightly underestimated BP (-15%). Furthermore, at the Hawaii Kokee site, all three versions of JULES underestimated the BP (C:-8%, CN:-8%, CNP: -32%). The litterfall and respiration fluxes in JULES-CNP have decreased compared to the JULES C and CN versions which overestimated both fluxes at all the test sites compared to the measurements. The litterfall flux comparisons show a significant overestimation using JULES C and CN versions across all the tested sites. Along the Amazon sites inclusion of P limitation reduced the litterfall flux but still overestimated (AGP: +50%, CAX: +24% and SA3: +16%) and at Gigante Peninsula and Hawaii Kokee slightly underestimated (Gigane Peninsula: -9% and Hawaii Kokee -19%). The respiration measurements were only available at two Amazon sites (CAX and SA3) at which inclusion of P limitation resulted in a well estimated flux at both sites compared to the JULES C/CN versions (CAX site: C-only: +38%, CN: +38%, CNP: -1%; SA3 site: C-only: +38%, CN: +38%, CNP: -2%). The total vegetation biomass also reduced using JULES-CNP compared to the other versions and yield closer values to the measurements across all the sites. However, except at the AGP site in which all three versions of JULES slightly underestimated the biomass (C: -1%, CN: -1%, CNP: -6%), at the other test sites JULES-CNP estimated lower biomass pools compared to the other versions which overestimated total vegetation biomass. Similarly, the soil C pool was overestimated prior to P limitation inclusion in JULES at the test sites, and the JULES-CNP estimated a closer value compared to the measurements (slight underestimation at CAX and SA3 sites: -5% and -18% respectively, and close values at Gigante Peninsula and Hawaii Kokee: +3% and +4%, respectively).

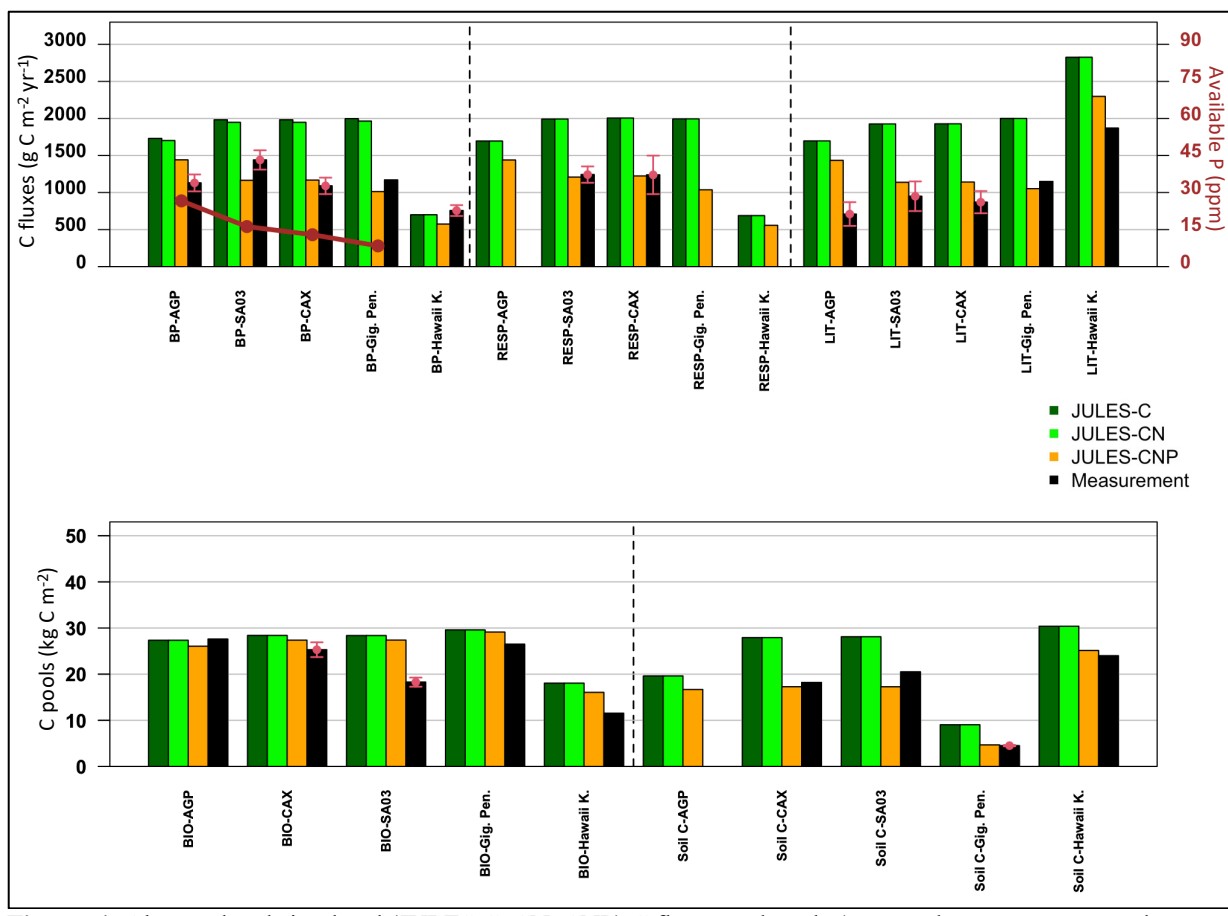

**Figure. 4-** Observed and simulated (JULES C, CN, CNP) C fluxes and pools (averaged measurements: red
points, sd: red arrows) and available observed P (dark red points and lines (reported in ppm)) at test sites across
the Amazon (AGP, SA03, CAX), Gigante Peninsula (Gig. Pen.) and Hawaii Kokee (Hawaii K.).
### 3.1.5 Model sensitivity
The results indicate that among all the corresponding C and P pools and fluxes, the excess C flux – which
demonstrates P limitation to growth – shows the highest sensitivity to changes in C:P ratios (Figure 5-a), $K_P$
(Figure 5-b), and $K_{or-max}$ (Figure 5-c) and $K_{in-max}$ (Figure 5-d). A decrease in plant C:P results in a large
increase in excess C. This is due to the higher plant P demand as a result of lower plant C:P ratios. An increase
in the uptake factor and maximum sorbed organic and inorganic P also results in an increase in excess C. This is
due to the higher uptake demand through higher uptake capacity (due to higher $K_P$) and lower available P for
uptake due to higher organic and inorganic sorbed P (due to higher $K_{or-max}$, $K_{in-max}$). Since the total P in the
system is lower than the plant demand, the uptake capacity and sorbed P, higher P limitation is placed on growth
(decreasing BP) which results in an increase in excess C and decrease in plant C, but also soil C which is a result
of lower litter input (Figure 5). Total soil P shows low sensitivity to changes in plant C:P and uptake factor but
high sensitivity to maximum inorganic sorbed P. Moreover, sorbed P shows middle to high sensitivity to
maximum organic and inorganic sorbed P respectively (Figure. S5). Nevertheless, organic and inorganic P
adsorption coefficients ($K_{sorp-or}$, $K_{sorp-in}$) show no sensitivity to C and P pools and fluxes. This is due to
limiting the organic and inorganic P sorption terms to be controlled only by maximum sorption capacity, hence
no effect applied by organic and inorganic adsorption coefficients.

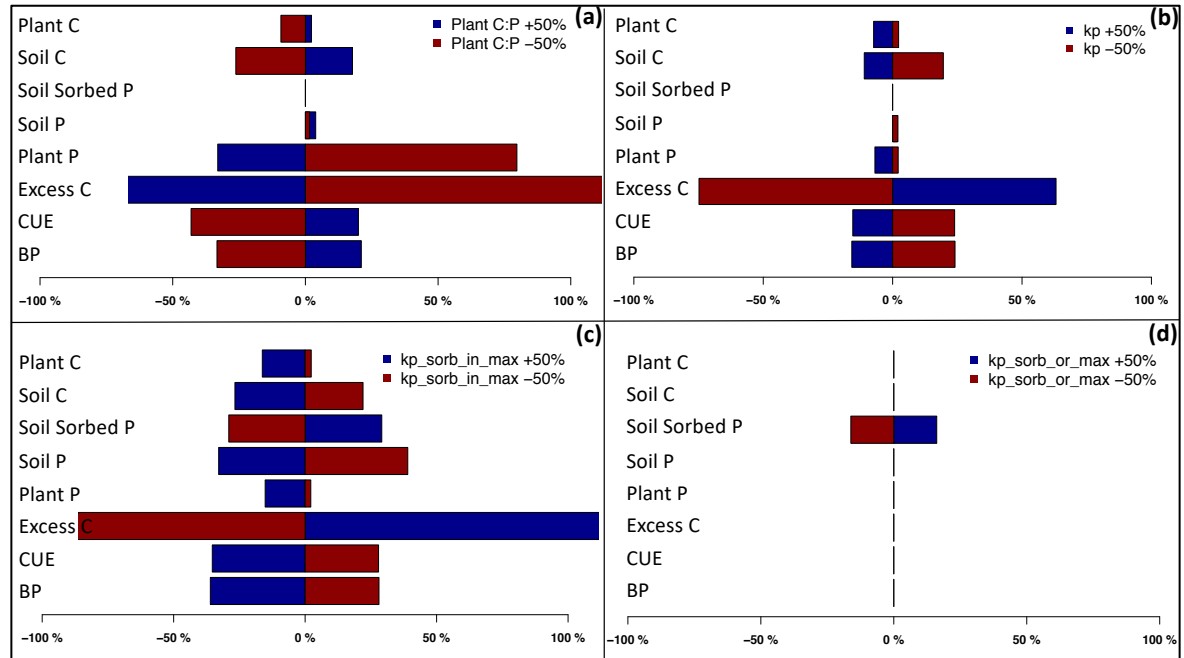

**Figure. 5-** Sensitivity of C and P pools to variation in key model parameters: prescribed tissue C:P (a), Kp(b), Kp_sorb_in(c), Kp_sorb_or(d) under ambient $CO_2$.

## 3.2    Model application under elevated $CO_2$

### 3.2.1    Simulated plant and soil C and P pools and fluxes -JULES-CNP: eCO₂ vs ambient CO₂

The eCO₂ simulation using JULES-CNP yields a higher GPP compared to the ambient $CO_2$ (0.83 (kg C $m^{-2}$ $yr^{-1}$) increase), as a result of $CO_2$ fertilization. Moreover, due to the GPP increase, NPP and RESP also increased compared to ambient CO₂ (NPP: 0.49 and RESP:0.3 (kg C $m^{-2}$ $yr^{-1}$) increase) (Table 7). The total simulated vegetation C pool increases under eCO₂ compared to ambient $CO_2$ (0.41 kg C $m^{-2}$), hence the estimated plant P (estimated as a fraction of C:P ratios) increases as well (+0.45 (g P $m^{-2}$)) (Fig. 6, Table 6). Thus, the simulated plant P demand is higher, and as the total available soil P for uptake is limited, the simulated excess C flux increases to 0.51(kg C $m^{-2}$ $yr^{-1}$).  Moreover, despite the higher NPP under eCO₂ compared to simulated NPP under ambient CO₂, due to the substantial increase in simulated excess C, the BP is similar to the ambient CO₂ (2% difference).

The simulated organic soil P under eCO₂ were close to those under ambient CO₂ (1.6 g P $m^{-2}$) (Table 7). This is due to the same parameterization of the output fluxes from this pool for eCO₂ and ambient CO₂. The simulated pool of inorganic P under eCO₂ decreases compared to the ambient CO₂ by 0.11 (g P m$^{-2}$) due to the increased plant P pools and slight increase in uptake (+0.13 %).
However, the simulated sorbed organic and inorganic soil P from eCO₂ are similar to those simulated under the ambient CO₂ which is due to the same parameterization of sorption function (maximum sorption capacity) from the ambient CO₂ run as explained in calibration section. Moreover, the modelled occluded and weathered soil P were similar to those in the ambient CO₂ simulation (Table 7) which is due to the same prescribed observational data that was used for this simulation.

### 3.2.2    Comparison of JULES C, CN and CNP under elevated CO₂

JULES C/CN show higher vegetation and soil C pools, BP and litter flux compared to JULES-CNP: (Table 8, Figure. S2). Under eCO₂, simulated NPP using JULES C-CN is 4.5% higher than JULES-CNP and the BP with JULES- C/CN is 96.8% higher than in JULES-CNP which simulates an excess C flux of 0.81 (kg C $m^{-2}$ $yr^{-1}$) equivalent to 46% P limitation under eCO₂. As a result of P limitation and eCO₂, the simulated $CO_2$ fertilization effect estimated based on changes in biomass under ambient and eCO₂ was reduced from 13% with JULES-C/CN to 10% JULES-CNP. Moreover, the CUE from JULES C/CN is 87.5% higher than the JULES-CNP as a result of high P limitation over biomass production.

**Table 8.** Simulated C pools and fluxes with JULES C/CN and difference in percentage with JULES-CNP model under eCO₂. A positive % means larger respective values simulated with JULES C and JULES CN than with JULES-CNP (between period 2017-18).

|  | GPP | NPP | BP | CUE | Litter C | Leaf C | Root C | Wood C | Soil C |
|---|---|---|---|---|---|---|---|---|---|
| **JULES C/CN** | 4.1 | 1.85 | 1.85 | 45% | 1.77 | 0.42 | 0.42 | 26.1 | 19.2 |
| **JULES-CNP** | 3.9 | 1.77 | 0.94 | 24% | 0.83 | 0.4 | 0.4 | 24.71 | 12.71 |
| **ΔC/CN: CNP** | 5.1% | 4.5% | 96.8% | 87.5% | 113.3% | 5% | 5% | 5% | 51.1% |

### 3.2.2.1 Inter-models under elevated CO₂

Following Fleischer *et al.*, (2019), we report the simulated response to eCO₂ for year 1999 (initial: CO₂ effect) and 1999-2013 (15 years: final effect) which are different than our evaluation period (2017-18). Using JULES-C and JULES-CN under eCO₂, simulated GPP and NPP during the 1st year increase by 30% and 61% respectively and by 28% and 52% after 15 years (Figure. 6). However, using JULES-CNP, eCO₂ increases simulated GPP, NPP and BP responses during the 1st year by 29%, 51% and 20% and by 28%, 43% and 7%, after 15 years, respectively.

Corresponding simulated CUE during the 1st year and 15 years shows an increase of 24% and 20% in response to eCO₂ using JULES C/CN, respectively. However, using JULES-CNP, simulated CUE for the 1st and after 15 years is reduced by 7% and 17% in response to eCO₂.

Simulated total biomass (leaf, fine root and wood C) ($\Delta C_{veg}$) using JULES- C/CN for the 1st and 15 years of eCO₂ increased by 9% and 13% respectively. However, using JULES-CNP $\Delta C_{veg}$ only increases by 0.5% and 9% for 1st and 15 years of eCO₂, respectively.

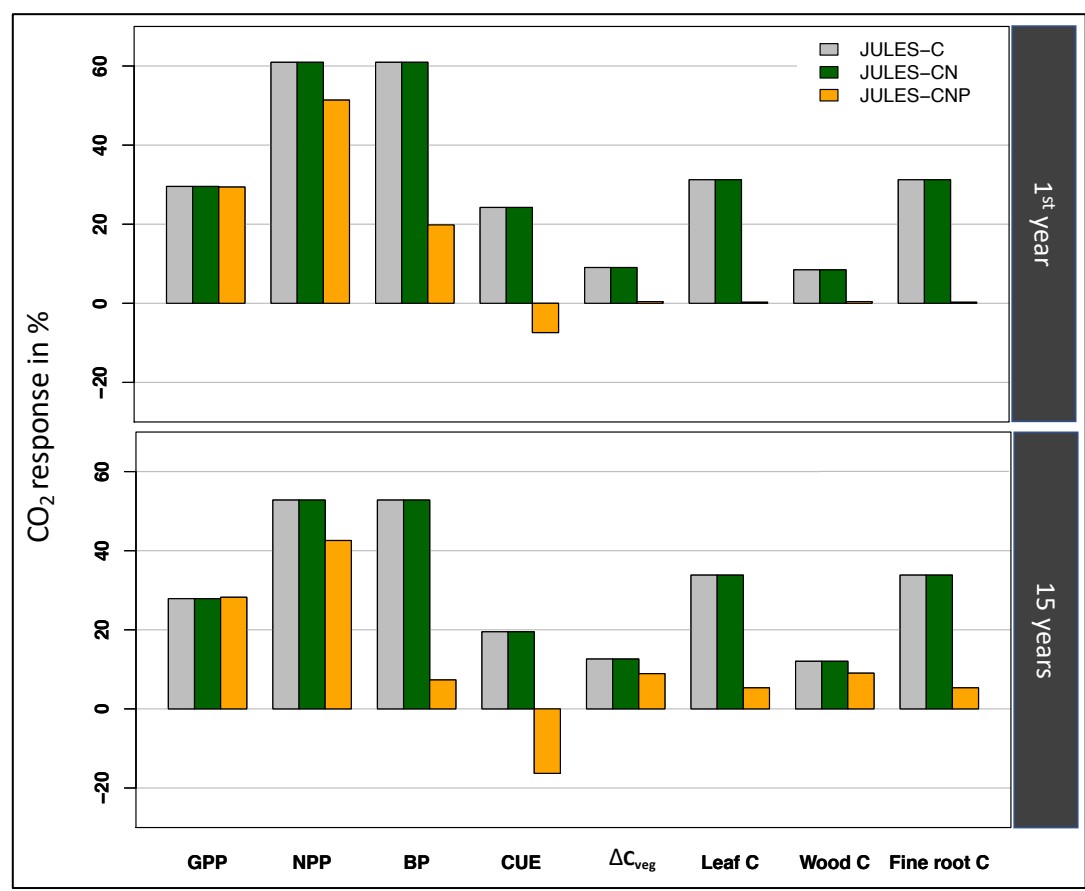

**Figure. 6-** Relative effect of eCO₂ on simulated GPP, NPP, BP, CUE, $\Delta C_{veg}$, leaf C, wood C and fine root C, using three versions of JULES model in 1st (initial response) and 15 years periods (final response).

### 3.3 Plant P Demand, uptake and excess C under ambient and elevated CO₂

To understand further the CP-cycle dynamics, we studied the monthly averaged plant P demand and the relative (limited) P uptake (eq. 26) under both ambient and elevated $CO_2$ conditions (Figure. 7).

Under ambient $CO_2$ condition the highest GPP is estimated at $0.29\pm0.016$ kg C m⁻² month⁻¹ in July and the lowest at $0.17\pm0.051$kg C m⁻² month⁻¹ in October (Figure. 7-a). The estimated WUE and SMCL in October is among the lowest estimated monthly values at $2.3\pm0.51$ kg $CO_2$/kg $H_2O$ and $526.2\pm31$ kg m⁻² respectively (Figure. 7-c). The highest P demand is estimated at $0.4\pm0.02$ g P m⁻² month⁻¹ in July and the lowest demand at $0.2\pm0.08$ g P m⁻² month⁻¹ in October. Consequently, the highest and lowest uptake ($0.32\pm0.01$ and $0.19\pm0.07$ g P m⁻² month⁻¹, respectively). The excess C for the highest and lowest GPP and demand periods are estimated at $0.4\pm15$ and $0.04\pm0.07$ kg C m⁻² month⁻¹, respectively.

However, similar to ambient $CO_2$, under eCO₂ condition the highest estimated GPP is in July at $0.36\pm0.017$ kg C m⁻² month⁻¹ and lowest for October $0.25\pm0.062$ kg C m⁻² month⁻¹ (Figure. 7-b). The estimated WUE and soil moisture content (SMCL) for the lowest GPP period is among the lowest monthly estimated values at $3.5\pm0.74$ kg $CO_2$/kg $H_2O$ and $552\pm33$ kg m⁻² for October respectively (Figure. 7-d). The highest P demand is estimated for July at $0.51\pm0.02$ g P m⁻² month⁻¹ with the uptake flux of $0.31\pm0.02$ g P m⁻² month⁻¹ and the lowest demand is estimated for October at $0.32\pm0.1$ g P m⁻² month⁻¹ with the estimated uptake flux of $0.26\pm0.06$ g P m⁻² month⁻¹. The highest excess C flux is also for July at $1.01\pm0.17$ kg C m⁻² month⁻¹ and lowest for October $0.27\pm0.29$ kg C m⁻² month⁻¹, respectively.

However, despite the P limitation in both eCO₂ and ambient $CO_2$ conditions, the P uptake flux under eCO₂ is higher than the ambient $CO_2$ condition. This is due to the higher WUE and increased SMCL (controlling uptake capacity (eq. 27)) under eCO₂ condition, hence more water availability during the dry season to maintain productivity and critically transport P to the plant (see eq. 27), compared to ambient $CO_2$ condition (Figure. 7-c and d). Additionally, in JULES both the vertical discretisation (Burke, Chadburn and Ekici, 2017) and mineralisation terms (Wiltshire *et al.*, 2021) depend on the soil moisture and temperature. Thus, higher P concentration and uptake under eCO₂ condition.

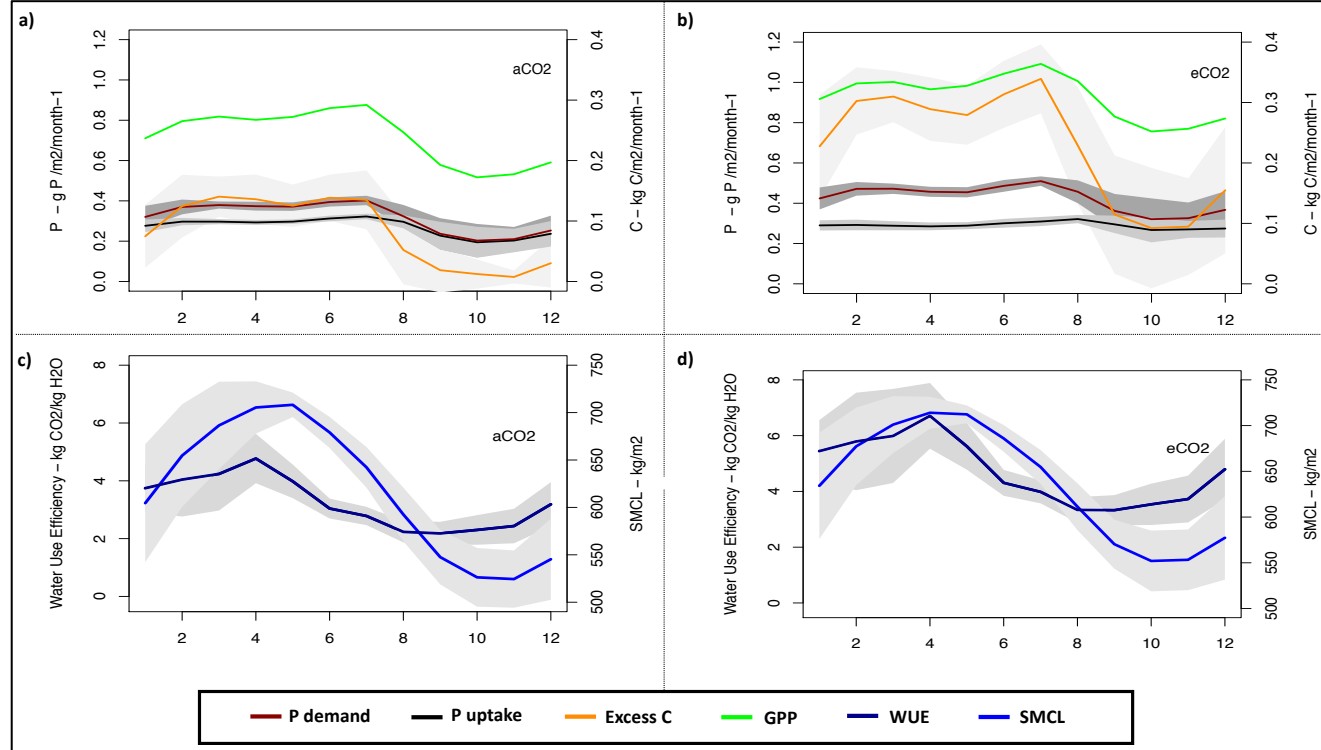

**Figure. 7-** Simulated monthly plant P demand and uptake (g P m⁻² month⁻¹), excess C and GPP (kg C m⁻² month⁻¹) under a) aCO₂ and b) eCO₂, water use efficiency (g m⁻² month⁻¹) under c) ambient $CO_2$ (aCO₂) and d) eCO₂ conditions. The grey area represents the standard deviation.

### 3.4 Soil P pools profile under ambient $CO_2$ and elevated $CO_2$

We explored the distribution of the inorganic and organic soil P and their sorbed fraction within the soil layers and under different $CO_2$ conditions (Figure. S3). Both the ambient and $eCO_2$ simulations have a close inorganic soil P distribution at the topsoil layer (0-30cm) (0.85 vs. 0.9 (g P m$^{-2}$) respectively) as well as similar organic soil P distribution (0.85 vs 0.9 (g P m$^{-2}$) respectively).

However, the organic soil P and sorbed forms of inorganic and organic soil P profiles are not changing significantly between different sets due to the similar parameterization of the processes that control these pools (processes which are related to the physical aspects of soils, hence not changing under $eCO_2$ condition) and the same parameter values used for both ambient and $eCO_2$ runs.

Moreover, the soil P within 30cm soil depth for ambient and $eCO_2$ conditions is at 14.7 (g P m$^{-2}$) and 14.56 (g P m$^{-2}$) respectively, and the total ecosystem P for both ambient and $eCO_2$ conditions is at 35.97 (g P m$^{-2}$). However, the slightly lower soil P in the $eCO_2$ condition is due to the higher plant P demand compared to the ambient condition, hence the higher allocated P vegetation (10%) under $eCO_2$ condition.

## 4. Discussion

Studies show the significant role of the tropical forests, and Amazonia in particular, in C uptake and regulating atmospheric $CO_2$ (Brienen *et al.*, 2015; Phillips *et al.*, 2017). As soil P availability is low in the majority of Amazonia (Quesada *et al.*, 2012), the competition for nutrients by both plant and soil communities is high (Lloyd *et al.*, 2001). The responses of these communities to $eCO_2$ under P limited conditions remains uncertain (Fleischer *et al.*, 2019). These responses in P enabled models are represented in different ways regarding the excess C which is not used for plant growth due to P limitation. Either growth is directly downregulated taking the minimum labile plant C, N and P (Goll *et al.*, 2017), or photosynthesis is downregulated via $V_{cmax}$ and $J_{max}$ (Comins and McMurtrie, 1993; Yang *et al.*, 2014; Zhu *et al.*, 2016) and models like JULES-CNP downregulate NPP via respiration of excess carbon that cannot be used for growth due to plant nutrient constraints (Haverd *et al.*, 2018). The estimated CUE depends on the modelling approach. Models that down regulate the photosynthetic capacity and GPP consequently (Comins and McMurtrie, 1993; Yang *et al.*, 2014; Zhu *et al.*, 2016) simulate a positive CUE response to $CO_2$ fertilization while models that down regulate the NPP and respire the excess C (Haverd *et al.*, 2018) simulate a negative CUE response (Fleischer *et al.*, 2019) which is in line with field studies showing lower CUE when nutrient availability declines (Vicca *et al.*, 2012b). However, this remains a major uncertainty in understanding the implication of P limitation on terrestrial biogeochemical cycles.

The JULES-CNP structure represent key P processes in both plant and soil pools and can be applied to the Amazon region using existing soil (Quesada *et al.*, 2011) and foliar structural and nutrient (Fyllas *et al.*, 2009) data for parameterisation. The model can be applied globally and under future climate projections using global soil P data (Sun *et al.*, 2021) for model initialization and PFT-specific plant (Zechmeister-Boltenstern et al. 2015) and soil stoichiometries (Zechmeister-Boltenstern et al. 2015; Tipping et al. (2016), sorption and weathering ratios (based on lithological class specific from the GliM lithological map (Hartmann and Moosdorf, 2012) and soil shielding from Hartmann et al., (2014)).

### 4.1. Evaluation of model performance

At the study site, JULES-CNP could reproduce the magnitude of soil organic and inorganic P pools and fluxes. The relative distribution of total organic P, total inorganic P and residue P fractions of total P in soils under Brazilian Eucalyptus plantations (Costa *et al.*, 2016) shows inorganic P fraction of 28% from total soil P which is close to our estimation of 24% and organic P fraction of 30% from total soil P which is higher than our estimated fraction of 18%. Thus, we may need to improve the process representation or parameters that control the organic P concentration, such as litter flux and decomposition, soil organic P mineralization, and immobilization in the future.

Our estimated maximum P uptake, which represents the actual available P for plant uptake (Goll *et al.*, 2017), for both ambient and $eCO_2$, is highly correlated with the plant P demand ($R^2 = 0.96$ and 0.52 respectively). The plant P demand depends on the GPP changes which are reflected by the WUE (Hatfield and Dold, 2019). Hence, under ambient $CO_2$, JULES-CNP simulates lower GPP and plant P demand during the dry season than during the wet season. Sufficient P uptake during these periods results in the lowest P limitation, thus the lowest simulated excess C. Nevertheless, under $eCO_2$ the same pattern is simulated but a higher availability of soil P due to the stomatal closure in the dry season. Hence, due to the plant's more efficient water usage, the soil

moisture in the dry season is higher (Xu *et al.*, 2016) which impacts our capped P uptake flux (eq. 27) and
increases the uptake capacity respectively.
Overall, JULES-CNP reproduced the observed C pools and fluxes which are in the acceptable ranges compared
to the measurements. However, using the JULES default $V_{cmax}$ estimation method (eq. 40), the model slightly
underestimates the total GPP (2.9 kg C m$^{-2}$ yr$^{-1}$ vs. 3-3.5 kg C m$^{-2}$ yr$^{-1}$). Therefore, in this version of the model,
we used the improved $V_{cmax}$ estimation method based on N and P (eq. 46) which resulted in a final estimated
GPP closer to the measurements (3.06 kg C m$^{-2}$ yr$^{-1}$).
Our results show an increase in GPP (21%) in response to eCO$_2$ which is higher than the average increase of
GPP reported in mature eucalyptus forests (11%), also growing under low P soils at the free air CO$_2$ enrichment
experiment (EucFACE) facility in Australia (Jiang *et al.*, 2020). This can be related to the lower decrease of
biomass growth response estimated by JULES-CNP (-3%) compared to the measurements from mature
eucalyptus forests (-8%) (Ellsworth *et al.*, 2017), due to the P limitation which was shown to impact the above-
ground biomass growth response in mature forests (Körner *et al.*, 2005; Ryan, 2013; Klein *et al.*, 2016).
In order to estimate the biomass production (BP), we deducted the excess C fluxes from NPP. Using JULES
C/CN models, the simulated biomass productivity enhancement due to eCO$_2$ (49%) is in the middle range of the
reported for different biomes by Walker *et al.*, (2021). Moreover, our estimated difference of BP between
ambient and eCO$_2$ conditions (2%) is close to the estimated difference for mature forests (3%) (Jiang *et al.*,
959 2020).
A global estimation for tropical forests using the CASA-CNP model which includes N and P limitations on
terrestrial C cycling, shows that NPP is reduced by 20% on average due to the insufficient P availability (Wang,
Law and Pak, 2010) which is close to our estimated P limitation of 24%. This finding is in line with a field study
that shows a strong correlation between the total NPP and the soil available P (Aragão *et al.*, 2009).
The estimated decrease of NPP in response to eCO$_2$ as a result of P limitation is in line with the findings from
CLM-CNP model at five tropical forests (Yang *et al.*, 2014) which indicates the CO$_2$ fertilization dependency
on the processes that affect P availability or uptake.
Our estimated CUE (0.31) is close to that by Jiang *et al.* (2020) for mature eucalyptus forests (0.31±0.03), as
well as to the measurement for our study site (0.34 ±0.1). There is currently a lack of representation of stand age
in JULES-CNP which can significantly affect CUE (e.g. mature trees are less responsive to the nutrient
limitations) (De Lucia *et al.*, 2007; Norby *et al.*, 2016). However, a recent development of Robust Ecosystem
Demography (RED) model in JULES (Argles *et al.*, 2020) and its integration into JULES-CNP in the future can
address this issue.
Under low P availability, all available P is considered to be adsorbed or taken by plant and microbes for further
consumption, with leaching considered to be minor within the time scales of our study period (Went and Stark,
1968; Bruijnzeel, 1991; Neff, Hobbie and Vitousek, 2000). Despite studies that show the possibility of P
fixation as a source of available P for plants (Van Langenhove *et al.*, 2020; Gross *et al.*, 2021), due to the strong
fixation of P in the soil (Aerts & Chapin, 2000; Goodale, Lajtha,Nadelhoffer, Boyer, & Jaworski, 2002), the P
deposited is unlikely to be available to plants in the short term (de Vries  et al., 2014), for this reason this
version of JULES-CNP did not include P deposition. However both P deposition and leaching are likely to have
a very important role on sustaining the productivity of tropical forests in the Amazon over longer time scales
(Van Langenhove *et al.*, 2020) and needs to be considered in future studies. Moreover, biochemical
mineralisation is also not included in the current version of JULES-CNP which only accounts for total
mineralization. However, models that include this process show no significant difference between total and
biochemical mineralized P which can be due to complexity of identifying the inclination of mineralization
versus uptake (Martins *et al.*, 2021). Lastly, in order to capture plant internal nutrient impact on the C storage,
future work should focus on implementing recent developments including Non-Structural Carbohydrate pools
(NSC) (Jones *et al.*, 2020) in JULES-CNP.

### 4.1.1. Evaluation of model performance at test sites

Overall, inclusion of P processes in JULES-CNP improved the previously overestimated C fluxes and pools using JULES-C and -CN versions. Generally, the biomass productivity tends to follow the observed P availability (Figure 4), where the sites with higher available P for uptake simulated higher productivity which is in line with observations across P availability in the Amazon (Aragão *et al.*, 2009). Nevertheless, this tendency could be altered if the natural conditions in these forests are perturbated. For instance, in case of the high mortality events in these P limited sites (Malhi *et al.*, 2009; Pyle *et al.*, 2009), regrowing forests developing over the highly weathered oxisols with limited available P (Davidson *et al.*, 2004), results in the shifting limitation from P to N (Herbert, Williams and Rastetter, 2003). Hence, the controlling processes under N limitation will be N-related and processes such as N leaching or outgassing (Yang *et al.*, 2014) will define the productivity. This shifting in limitation condition is not represented by JULES-CNP, therefore at few tested sites the model overestimated the P limitation, thus underestimated the productivity below the measured values. Moreover, the higher (than other sites) BP in JULES C/CN at the the Gigante Peninsula is related to the higher solar radiation in the forcing data at this site (Figure S8).

The estimated litterfall and respiration fluxes were considerably lower with JULES-CNP than JULES-C and -CN due to the lower simulated NPP with the former in closer agreement with the observations at all sites. Consequently, the total vegetation and soil C pools have lower values under the P limitation (Malhi *et al.*, 2009), which could not be captured by JULES-C and -CN and successfully represented by JULES-CNP.

As shown in Figure 5, JULES-CNP is highly sensitive to the five parameters needed to run JULES-CNP in addition to JULES-C and JULES-CN which were prescribed for simulations at test sites. The successful model performance at these sites demonstrates the importance of these parameters in JULES-CNP with implications for global scale simulations.

### 4.2. Inter-models Comparison of JULES C, CN and CNP

The comparison of simulated GPP enhancement across JULES versions for the 1st year is within the middle range of the 1st year $CO_2$ responses of the C/CN models studied by Fleischer *et al.*, (2019) evaluating simulated $eCO_2$ effects at a site in Manaus using the same meteorological forcing and methodology used in this study for a range of DGVM's. However, comparison for 15 years of $eCO_2$, shows that the simulated response with JULES-CNP is on the higher end of Fleischer *et al.*, (2019) study which is due to the higher estimated biomass growth by JULES-CNP (Table S1). Similarly, using JULES-CNP our estimated GPP enhancement is on the higher end of model estimations in Fleischer *et al.*, (2019). Moreover, comparing the GPP responses between different versions of (JULES C/CN and CNP), the JULES-CNP shows a slightly higher response to $CO_2$ fertilization associated with the higher WUE changes (Xiao *et al.*, 2013) (Figure. S4). This is due to the higher sensitivity of the plant to water availability than P availability in the P limited system (He and Dijkstra, 2014). Hence, under $eCO_2$ due to water-saving strategy of plants and stomatal closure (Medlyn *et al.*, 2016), simulated transpiration is decreased (Sampaio *et al.*, 2021) and photosynthesis is enhanced compared ambient $CO_2$ .

To that end, the monthly changes of WUE in JULES-CNP are highly correlated to the GPP, hence the lowest and highest WUE follow the same periods as GPP similar to responses captured with models studied by Fleischer *et al.*, (2019) (Table. S1).

Our estimated NPP enhancement using JULES C/CN models for both 1st and 15 years period is within the middle range of the models in Fleischer *et al.*, (2019). Nevertheless, JULES-CNP response of BP is in the lower band of the CNP models in Fleischer *et al.*, (2019) and close to the estimations from CABLE (Haverd *et al.*, 2018) and ORCHIDEE (Goll *et al.*, 2017) models, which may be due to the similar representation of P processes and limitation between these models. However, our results show a 29% decrease in NPP using JULES-CNP compared to JULES-C/CN which is smaller than the differences between the CLM-CNP and CLM-CN versions (51% decrease) (Yang *et al.*, 2014). The lower estimated decrease in JULES highlights the need to further study the fully corresponding plant C pools and fluxes to the changes in soil and plant P. Therefore, future work should be focused on the improvement of the total P availability and the plant C feedbacks. Moreover, there are other environmental factors such as temperature which shows a possible impact on the $CO_2$ elevation and the changes of NPP (Baig *et al.*, 2015) which needs further improvement in our model.

The CUE estimations of 1st year and 15 years response to $CO_2$ elevation from JULES C/CN are in the middle range of C/CN models in Fleischer *et al.*, (2019). However, the estimated CUE using JULES-CNP for 1st and 15 years are in the low range of CNP models reported by Fleischer *et al.*, (2019) which is due to the same reason discussed for NPP comparison.

Finally, our estimated total biomass enhancement ($\Delta Cveg$) using JULES C/CN for the 1st and 15 years are in the
middle range of C/CN models from Fleischer *et al.*, (2019) and in lower range of CNP models from Fleischer *et*
*al.*, (2019) using JULES-CNP. Nevertheless, while JULES-CNP includes the trait-based parameters (Harper *et*
*al.*, 2016), other functions such as flexible C allocation and spatial variation of biomass turnover are still
missing and future model improvement should be focused on their inclusion.
**5.   Conclusion**
Land ecosystems are a significant sink of atmospheric $CO_2$, ergo buffering the anthropogenic increase of this
flux. While tropical forests contribute substantially to the global land C sink, observational studies show that a
stalled increase in carbon gains over the recent decade (Brienen *et al.*, 2015; Hubau *et al.*, 2020). However
modelling studies that lack representation of P cycling processes predict an increasing sink (Fernández-Martínez
*et al.*, 2019; Fleischer *et al.*, 2019). This is particularly relevant for efforts to mitigate dangerous climate change
and assumptions on the future efficacy of the land C sink. Therefore, in this study, we presented the full
terrestrial P cycling and its feedback on the C cycle within the JULES framework. Our results show that the
model is capable of representing plant and soil P pools and fluxes at a site in Central Amazon and across the
extended P limited test sites in Amazon, Gigante Peninsula and Hawaii chronosequence provided with site level
data for model parameterisation. Moreover, the model estimated a significant NPP limitation under ambient
$CO_2$, due to the high P deficiency at these sites which is representative of Central Amazon and tropical P limited
sites, and elevated $CO_2$ resulted in a further subsequent decrease in the land C sink capacity relative to the
model without P limitation. While our study is a step toward the full nutrient cycling representation in ESMs, it
can also help the empirical community to test different hypotheses (i.e., dynamic allocation and stoichiometry)
and generate targeted experimental measurements (Medlyn *et al.*, 2015).
*Code availability*
The modified version of JULES vn5_5 and the P extension developed for this paper are freely available on Met
Office Science Repository Service:
https://code.metoffice.gov.uk/svn/jules/main/branches/dev/mahdinakhavali/vn5.5_JULES_PM_NAKHAVALI/
after registration (http://jules-lsm.github.io/access_req/JULES_access.html) and completion of software license
form. Codes for compiling model available at: (https://doi.org/10.5281/zenodo.5711160). Simulations were
conducted using two sets of model configurations (namelists): ambient $CO_2$ condition
(https://doi.org/10.5281/zenodo.5711144) and elevated $CO_2$ condition
(https://doi.org/10.5281/zenodo.5711150).
*Data availability*
The model outputs related to the results in this paper are provided on Zenodo repository
(https://doi.org/10.5281/zenodo.5710898). All the R scripts used for processing the model outputs and
producing results in form of table or figures are provided on Zenodo repository
(https://doi.org/10.5281/zenodo.5710896).
*Author contributions.* MAN, LMM, SS, SEC, CAQ, AJW, IAP, KMA and DBC developed the model, per-
formed simulations and analysis. CAQ, FVC, RP, LFL, KMA, GR, LS, ACMM, JSR, RA and JLC provided the
measurements for the model parasitisation and evaluation. MAN, LMM, SS, IAP, SEC, FVC, RP, LFL, KMA
and DBC contributed in writing the manuscript.
*Competing interests.* The authors declare no competing interests
*Acknowledgment*s. This work and its contributors (MAN, LMM, KMA and IPH) were supported by the UK
Natural Environment Research Council (NERC) grant no. NE/LE007223/1. MAN, LMM, SS, IPH were also
supported by the Newton Fund through the Met Office Climate Science for Service Partnership Brazil (CSSP
Brazil). LMM acknowledges support from the Natural Environment Research Council, grant NEC05816 LTS-
M-UKESM. LFL was also supported by AmazonFACE programme (CAPES) and the National Institute of
Amazonian Research, grant no: 88887.154643/2017-00. The authors acknowledge contributions from Celso
Von Randow towards data curation of the meteorological forcing used in this study and Daniel Goll for
modelling insight. We would like to thank Alessandro C. de Araújo and the Large-Scale Biosphere-Atmosphere
Program (LBA), coordinated by the National Institute for Amazon Researches (INPA), for the use and
availability of data. We thank Jefferson Goncalves de Souza for processing the data required for additional sites
simulation.

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
