# Peer review of "Representation of phosphorus cycle in Joint UK Land 2 Environment Simulator (vn5.5\_JULES-CNP)"

_Geoscientific Model Development, 2021_

## Author Response (AR1)

We thank Dr Mingkai Jiang and the two anonymous reviewers for their thorough and constructive comments and suggestions. We modified the text and supporting documents accordingly. The changes include additional explanation on the novelty of model, N:P interactions and clarification of the equations and terms requested by the reviewers. Reviewer's comments are in bold black, responses to the reviewer are in blue italic font with added text to the manuscript in quotations.

**RC1:**

**This is a straightforward paper evaluating the inclusion of P cycle processes in JULES. The work is obviously important for the importance of P in estimating the global C sink and the important role of JULES in the modelling community. I have several comments to hopefully help further strengthen the manuscript.**

*We thank Dr Mingkai Jiang for his positive comments. We have addressed the comments as described below.*

**Given that the site is a nutrient fertilization experiment, it seems to be a missed opportunity to not evaluate the model performance in response to nutrient fertilization. The evaluation of the CO2 response is obviously still important, but I don't see enough rationale to do it given that there is limited data available to actually evaluate the simulated CO2 responses. Can we learn more by simulating the nutrient fertilization response where there is actual data?**

*We agree with the reviewer that additional evaluation of JULES-CNP against treatment (fertilization) data from the Amazon nutrient fertilization experiment (AFEX) is of high value. AFEX is an ongoing project and only changes in fine root dynamics have been published so far, so we don't have all the datasets needed for model evaluation. Therefore, it is not possible evaluate the model against the nutrient addition response yet but this is planned in a future model application study.*

*Given the fact that rightly two reviewers have asked about this, to avoid confusion, we have decided to simply refer to our 'study site' in Amazonia where we have access to key soil and plant data, rather than explicitly elaborate that this is the control site from AFEX.*

*We have modified lines 119-125 at the end of the introduction accordingly:*
"Here, we describe the development and implementation of the terrestrial P cycle in the Joint UK Land Environment Simulator (JULES) (Clark *et al.*, 2011), the land component of the UK Earth System Model (UKESM), following the structure of the prior N cycle development (Wiltshire *et al.*, 2021). The model (JULES-CNP) is parameterized and calibrated using novel in situ P soil and plant data from a well-studied forest site in Central Amazon near to Manaus, Brazil with soil P content representative of 60% of soils across the Amazon basin. We then evaluate the model against carbon stocks and fluxes from data sets from our study site and the nearby K34 field site"

*We also modified the following lines in the methods section 2.3; study sites in lines 460-471*
"This study uses data from two nearby sites in Central Amazon in Manaus, Brazil. The main site from here on termed *study site* (2°35′′21.08′′ S, 60°06′′53.63′′ W) (Lugli *et al.*, 2020) is for model development and evaluation. The second site is the Manaus K34 flux site (2°36′′32.67′′ S, 60°12′′33.48′′ W) which provides meteorological station data for running the model but also provides data for model evaluation.

We use detailed novel soil and plant P pool data from the *study site* (Lugli *et al.*, 2020, 2021) for model parameterisation and calibration and carbon stock data for model validation. The *study site* has a very similar forest, geomorphology, soil chemistry and species composition to the well-known and studied K34 eddy covariance flux site (Araújo *et al.*, 2002)."

**Some details of the model description is not available. For example, how P interacts with N to affect allocation, plant growth, stoichiometry and nutrient uptake?**

*Many thanks for spotting this. This was omitted due to the lack of N limitation at our study site. Details of the N and P interaction in JULES CNP in the revised are described in the methods section in lines 284-318 as follows:*

"Similar to other P-enabled models (Yang *et al.*, 2014; Goll *et al.*, 2017), JULES-CNP follows the same structure as its N model component. Description of the plant P and N demand follow Wang *et al.*, (2007) and are represented by the sum of demand ($\emptyset_t$) to sustain growth (P-related: ($\emptyset_{g_P}$), N-related: ($\emptyset_{g_N}$)) and vegetation spreading (to increment PFT fractional coverage) (P-related: ($\emptyset_{S_P}$), N-related: ($\emptyset_{S_N}$)) and is expressed in (P-related in kg P m$^{-2}$ yr$^{-1}$; N-related in kg N m$^{-2}$ yr$^{-1}$). The total demand for growth ($\emptyset_g$) and spreading ($\emptyset_s$) is controlled by the dominant demand between P ($\emptyset_{g_P}$) and N ($\emptyset_{g_N}$) as follows:

$$\emptyset_t = \emptyset_g + \emptyset_s \qquad (eq.19)$$

$$\emptyset_{gP} = \frac{P_p}{C_V}\left(\Pi_c - \frac{dC_v}{dt} - \psi_g\right) \qquad (eq.20)$$

$$\emptyset_{SP} = \frac{P_p}{C_V}\left(\Pi_c - \frac{dC_v}{dt} - \psi_s\right) \qquad (eq.21)$$

$$\emptyset_{gN} = \frac{N_v}{C_V}\left(\Pi_c - \frac{dC_v}{dt} - \psi_g\right) \qquad (eq.22)$$

$$\emptyset_{SN} = \frac{N_v}{C_V}\left(\Pi_c - \frac{dC_v}{dt} - \psi_s\right) \qquad (eq.23)$$

$$\emptyset_g = \begin{cases} \emptyset_{gP} & \emptyset_{gP} \times \frac{C_V}{P_p} > \emptyset_{gN} \times \frac{C_V}{N_v} \\ \emptyset_{gN} & \emptyset_{gN} \times \frac{C_V}{N_v} > \emptyset_{gP} \times \frac{C_V}{P_p} \end{cases} \qquad (eq.24)$$

$$\emptyset_s = \begin{cases} \emptyset_{SP} & \emptyset_{SP} \times \frac{C_V}{P_p} > \emptyset_{SN} \times \frac{C_V}{N_v} \\ \emptyset_{SN} & \emptyset_{SN} \times \frac{C_V}{N_v} > \emptyset_{SP} \times \frac{C_V}{P_p} \end{cases} \qquad (eq.25)$$

where $\frac{P_p}{C_V}$ is the inverse of whole plant C:P ratio, $\frac{N_v}{C_V}$ is inverse plant C:N ratio, $\frac{dC_v}{dt}$ is rate of change in plant C (see Clark *et al.,* (2011) for more detail), $\Pi_c$ is nutrient-unlimited, or potential, NPP (kg C m$^{-2}$ yr$^{-1}$), $\psi_g$ is excess C due to either P or N limitation for plant growth (kg C m$^{-2}$ yr$^{-1}$) and $\psi_s$ is excess C due to either P or N limitation for vegetation spreading (kg C m$^{-2}$ yr$^{-1}$).

Equations 20 and 22 are solved by first setting $\psi_g = 0.0$ to find the total plant P (eq. 20) and N demand (eq.22). If the P and N demand for growth are less than the available P and N and fractional coverage ($\lambda$) (NPP fraction used for fractional cover increment; for detail see Wiltshire *et al.,* (2021)) at the considered timestep $\Delta t$, then there is no limitation to growth ($i.e. \emptyset_{gP} < \frac{(1-\lambda)P_{avail}}{\Delta t}$; $\emptyset_{gN} < \frac{(1-\lambda)N_{avail}}{\Delta t}$). Where there is limited P and/or N availability, the uptake equals the available P and N ($\emptyset_{gP} = \frac{(1-\lambda)P_{avail}}{\Delta t}$; $\emptyset_{gN} = \frac{(1-\lambda)N_{avail}}{\Delta t}$), and the plant growth which cannot be achieved due to nutrient constraints will be deducted from potential NPP, here termed excess C term ($\psi_g$), to give an actual NPP. Following Wiltshire et al., 2021, we assume excess C is respired by the plant.

Similarly, in order to estimate the P and N demand for spreading (eq. 21 and 23), initially the excess C from spreading is set to 0.0 ($\psi_s = 0.0$), i.e under the assumption that there is no nutrient limitation. If the P and N demand for spreading are lower than the available P and N and fractional coverage ($\lambda$) ($\emptyset_{SP} < \frac{(1-\lambda)P_{avail}}{\Delta t}$; $\emptyset_{SN} < \frac{(1-\lambda)N_{avail}}{\Delta t}$), then there is no limitation on spreading and in case of limited P and N availability, the uptake equals the available P and N ($\emptyset_{SP} = \frac{(1-\lambda)P_{avail}}{\Delta t}$; $\emptyset_{SN} = \frac{(1-\lambda)N_{avail}}{\Delta t}$), and the excess C for spread ($\psi_s$) is subtracted from potential NPP."

*And lines366-388:*

"The soil respiration from each soil layer ($R_{i,n}$) is estimated from potential soil respiration ($R_{POT_{i,n}}$) for the DPM, RPM pools and the litter decomposition rate modifier ($F_{P_n}$) as follows:

$$R_{i,n} = R_{POT_{i,n}} \times F_{P_n} \tag{eq.33}$$

where the description of $F_{P_n}$ for P pools ($F_{PP_n}$) follows Wang *et al.*,(2007) and is estimated based on the soil pool (BIO/HUM) mineralization ($minl_{P-BIO_n}$, $minl_{P-HUM_n}$) and immobilization ($immob_{P-BIO_n}$, $immob_{P-HUM_n}$) (in kg P m$^{-2}$ yr$^{-1}$), soil inorganic P ($P_{inorg_n}$) (in kg P m$^{-2}$), and litter pools (DPM/RPM) demand (in kg P m$^{-2}$ yr$^{-1}$) as follows:

$$F_{PP_n} = \frac{(minl_{P-BIO_n} + minl_{P-HUM_n} - immob_{P-BIO_n} - immob_{P-HUM_n}) + P_{inorg_n}}{DEM_{DPM_n} + DEM_{RPM_n}} \tag{eq.34}$$

The net demand associated with decomposition of litter pools ($DEM_{k,n}$) represents the P required by microbes which convert DPM and RPM into BIO and HUM. The limitation due to insufficient P availability is estimated based on the potential mineralization ($minl_{p-pot}$) and immobilization ($immob_{p-pot}$) (in kg P m$^{-2}$ yr$^{-1}$) of pools (k) as follows:

$$DEM_{k,n} = immob_{p-pot,k} - minl_{p-pot,k} \tag{eq.35}$$

The $F_{P_n}$ estimated for N pools ($F_{PN_n}$) follows the same formulation as P (see Wiltshire *et al.*, 2021 for detail) and the $F_{P_n}$ is estimated based on a higher rate modifier between N and P as follows:

$$F_{P_n} = \begin{cases} F_{PP_n} & F_{PP_n} > F_{PN_n} \\ F_{PN_n} & F_{PN_n} > F_{PP_n} \end{cases} \tag{eq.36}"$$

**Can the authors spend some efforts highlighting the novelty of this work? I understand that adding P cycle into JULES is a significant work, but some readers could take it as "yet another model with P cycle". Considering the spread of the simulated CO2 responses among CNP models in Fleischer et al. 2019, having yet another model doesn't necessarily reduce the model spread and our knowledge gap. Is there anything specific about having a P cycle in JULES would help to resolve? In other words, what unique features JULES-CNP can provide relative to other models?**

*In the most recent CMIP6, only one model includes P (ACCESSESM1.5 model) (Arora et al., 2020) and only few studies focused on P cycling in soil and between plant and soils for tropical forests particularly (Yang et al., 2014; Zhu et al., 2016; Goll et al., 2017). There are no dedicated model-data nutrient cycling studies specifically for the Amazon forest with poor soils and limited P availability that is hypothesized to be among the key limiting nutrients to plant growth (Vitousek et al., 1997, 2010; Hou et al., 2020). As acknowledged by RC2 the inclusion of the P-related processes in models is critical to our ability to predict ecosystem responses to eCO2 and climate change in low-P systems such as the tropics and some land surface models are now adding these processes. We created a model suitable for global application, and it is appropriate to first test it at a site-level where data are available. We included detailed P processes such as: litter production and decomposition, mineralisation, immobilisation, ad/desorption, occlusion, weathering and plant uptake and we use the actual site measurements from soil pools from our study site to constrain the model in contrast with other P model development which use soil-specific data (Goll et al., 2017) or simplified approach (fluxes estimation based on equilibrium state of other fluxes (Wang et al., 2018)). At this site-level it is standard practise to parameterise and calibrate the model where possible, to be able to then evaluate model performance for key variables. This information is added in lines 119-125 as follows:*

"Here, we describe the development and implementation of the terrestrial P cycle in the Joint UK Land Environment Simulator (JULES) (Clark *et al.*, 2011), the land component of the UK Earth System Model (UKESM), following the structure of the prior N cycle development (Wiltshire *et al.*, 2021). The model (JULES-CNP) is parameterized and calibrated using novel in situ P soil and plant data from a well-studied forest site in Central Amazon near to Manaus, Brazil with soil P content representative of 60% of soils across the Amazon basin. We then evaluate the model against carbon stocks and fluxes from data sets from our study site and the nearby K34 field site."

Also

*JULES_CNP model structure was designed to match soil P pools for which data is available for the whole Amazon region, with a global data set also available for global applications. This is added in lines 827-833 in the discussion section 4:*

"Our new developments include major P processes in both plant and soil pools and can be applied to the Amazon region using existing soil (Quesada *et al.*, 2011) and foliar structural and nutrient (Fyllas *et al.*, 2009) data for parameterisation. Moreover, JULES CNP can be applied at the global scale and for future projections using global soil P data (Sun *et al.*, 2021) for model initialization and PFT-specific plant stoichiometries (Zechmeister-Boltenstern et al. 2015), soil stoichiometries (Zechmeister-Boltenstern et al. 2015; Tipping et al. (2016), sorption and weathering ratios (based on lithological class specific from the GliM lithological map (Hartmann and Moosdorf, 2012) and soil shielding from Hartmann et al., (2014))."

**Specific comments:**

**L31: Unclear – what does this 60% mean? Is it the 60th percentile of P availability (doesn't sound like a particularly P limited site)? Or does it mean the total P/available P at the site represents 60% of soil across the Amazon? Be good to make it clearer.**

*Indeed, our study soil P content is representative of 60% of soils across the Amazon basin. This is clarified in the revised version as follows:*

"…a low fertility site, with a soil P content representative of 60% of soils across the Amazon basin."

**L32: What is the eCO2 treatment? Show a number is useful.**

*This value is +200 ppm, over ambient [~400 ppm] We have added the value, see below:*

"We then apply the model under elevated $CO_2$ (600 ppm) at our study site to quantify the impact of P limitation on $CO_2$ fertilization."

**L32: The model is able to reproduce observed plant and soil P pools under ambient conditions – to what extent are these values provided as parameters and/or targeting values to tune the model?**

*Please note that, the plant P pools are calculated based on observations of C:P ratios, and soil P pools have been either prescribed (occluded and weathered) or calibrated via a single parameter (maximum sorption capacity) (sorbed organic and inorganic P) or evaluated (organic and inorganic P). Moreover, the model is able to reproduce observed fluxes and stock of C. We have now corrected Table 3 and clarified this in lines 32-34 as follows:*

"Novel soil and plant P pool observations are used for parameterisation and calibration and the model is evaluated against C fluxes and stocks, and for those soil P pools not used for parameterisation/calibration."

*And lines 515-517:*

"JULES-CNP has fixed stoichiometry and C:P ratios of leaf and root (measured), and wood (estimated from fresh coarse wood (Lugli, 2013)) which were taken from the *study site* and prescribed in JULES to simulate P dynamics in the plant"

*And lines 530-532:*

"Thus, in order to cap P sorption and uptake capacity, the maximum sorption capacities ($P_{in-max_n}$, $P_{or-max_n}$, eq.37 and 39) (adopted from (Wang, Houlton and Field, 2007)) were prescribed using maximum observed sorbed inorganic and organic P."

*And lines 533-535:*

"Moreover, as the magnitude of changes in the occluded and parent material pools are insignificant over a short-term (20 years) simulation period (Vitousek *et al.*, 1997), these two pools were prescribed using observations."

**Abstract: Given that you evaluated the model at a nutrient fertilization site, does it make sense to apply the nutrient treatment to the model and try to evaluate the model against observation? It's quite unclear what to learn from evaluating the model against a CO2 treatment where you don't have any data. The rationale about the CO2 effect isn't very clear in the abstract.**

*Having 1) developed and parameterised our P model with plant and soil observations from a site in the Amazon (Manaus) and 2) evaluated for C fluxes under ambient $CO_2$, the rationale behind the simulations under $eCO_2$ was to quantify the impact of P limitation on $CO_2$ fertilization in the Amazon in JULES-CNP and compare against the ensemble of models some of which include P cycle in Fleischer et al., 2019 which was performed as part of the AmazonFACE intercomparison, a nearby site to our study site.*

*In response to the four above specific comments, we have modified the abstract to read as follows in lines 30-36:*
"We evaluate JULES-CNP using in situ data collected at a low fertility site in the Central Amazon, with a soil P content representative of 60% of soils across the Amazon basin, to parameterise, calibrate and evaluate JULES-CNP. Novel soil and plant P pool observations are used for parameterisation and calibration and the model is evaluated against soil P pools and C fluxes and stocks. We then apply the model under elevated $CO_2$ (600 ppm) at our study site to quantify the impact of P limitation on $CO_2$ fertilization. We compare our results against current state of the art CNP models using the same methodology that was used in the AmazonFACE model intercomparison study."

**L42: net primary production.**

*Corrected.*

**L43: C, not carbon.**

*Corrected.*

**L47: It's not either or. There are other nutrients involved.**

*Corrected.*

**L99: Hou et al. 2019 – are you sure this is a DGVM?**

*The references are corrected in the revised version.*

**Figure 1. Do you have P leaching flux and biochemical P mineralization flux? What about atmospheric P deposition?**

*Due to the low P availability at this site, all the available P is considered to be returned to plants, thus leaching is to be minor within the timescales simulated (Went and Stark, 1968; Bruijnzeel, 1991; Neff, Hobbie and Vitousek, 2000). However, the current version of JULES CNP only includes the total mineralization. Nevertheless, even the models which include this process show no significant difference between biochemical P mineralization to total P mineralization (Goll et. al., 2017). Also, JULES-CNP does not include atmospheric P deposition which is envisaged in future model developments and applications. This information is now added to discussion in the revised version in discussion section 4.1 – lines 888-895:*
""Moreover under low P availability, all available P is considered to be adsorbed or taken by plant and microbes for further consumption, with leaching considered to be minor within the time scales of our study period (Went and Stark, 1968; Bruijnzeel, 1991; Neff, Hobbie and Vitousek, 2000).
Due to the strong fixation of P in the soil (Aerts & Chapin, 2000; Goodale, Lajtha,Nadelhoffer, Boyer, & Jaworski, 2002), the P deposited is unlikely to be available to plants in the short term (de Vries et al., 2014), for this reason this version of JULES CNP did not include P deposition. However both P deposition and leaching are likely to have a very important role on sustaining the productivity of tropical forests in the Amazon over longer time scales (Van Langenhove et al., 2020) and needs to be considered in future studies."

**From the equation it seems that you would have a continuous build-up of P in the system, because there is no way out from the occluded P pool.**

*In our simulations we deactivated the occluded flux and the reported occluded pool is prescribed in the model using the observed value. Thus, we have a closed P system balanced between vegetation and active soil pools. This is further clarified in the revised version as follows:*
"Moreover, as the magnitude of changes in the occluded and parent material pools are insignificant over a short-term (20 years) simulation period (Vitousek et al., 1997), these two pools were prescribed using observations for these two pools."

**L239: This exudate term is confusing. Can I consider it as part of autotrophic respiration? It's a missed opportunity where this exudate term isn't pumped into soil to facilitate plant-soil interaction, which could potentially be used to alleviate plant nutrient stress. See for example (Jiang et al., 2020).**

*The reviewer is right. Since in the model this excess does not have an explicit functional role in facilitating plant-soil interaction we now refer to it as "excess C" throughout the whole text, figures and tables. Following JULES-CN (Wiltshire et al., 2021) we simply assume it is part of the autotrophic respiration.*

**L284: What timestep does retranslocation occur?**

*The retranslocation occurs at a daily timestep. This is now added to the revised version in the 2.2.2 section in line 342-344 as follows:*
"where $\lambda$ is the leaf, root and stem re-translocation (at daily timestep) coefficient (Zaehle and Friend, 2010; Clark et al., 2011) and the related $C:P$ ratios for P fraction and $\gamma$ is a temperature dependent turnover rate representing the phenological state (Clark et al., 2011)."

**Table 2: Would be good to show P retranslocation coefficient.**

*We added these coefficients to the table2 in the revised version.*

**Method: Description on N-P interaction is missing. I suppose P is not entirely independent from N processes? I think it should be useful to describe how P affects N, and vice versa in the model.**

*We added detailed explanation on the N-P interaction as stated in the general comment part above and it can be found in the methods section (2.2.2) in lines 284-388*

**L404: What about N only model?**

*This is now clarified in the methods in lines 481-482 as follows:*
"JULES vn5.5 (JULES CN in this study) estimates Vcmax ($\mu$mol m$^{-2}$ s$^{-2}$) based on Kattge et al. (2009) using foliar N concentrations in area basis"

**What about Jmax?**

*In JULES vn5.5 that JULES CNP was developed on, we use Collatz C3 and C4 photosynthesis model (Collatz et al., 1991; Collatz, Ribas-Carbo and Berry, 1992) which does not include the estimation of Jmax. This information is now clarified in the methods in lines 513-514 as follows:*
"Note that JULES CNP uses C3 and C4 photosynthesis model from Collatz et al., 1991; Collatz, Ribas-Carbo and Berry, 1992, which does not include estimation of $J_{max}$."

**L428: I may have missed this, but does the model assume fixed or variable CP ratios?**

*We have clarified this in lines 515-517 in methods section 2.4:*
"JULES-CNP has fixed stoichiometry and C:P ratios of leaf and root (measured), and wood (estimated from fresh coarse wood (Lugli, 2013)) which were taken from the *study site* and prescribed in JULES to simulate P dynamics in the plant."

**L450: What does the symbol -//- mean?**

*We have removed this symbol and complete the table for clarity as follows:*
**Table 3.** Observations from study site (taken during 2017-2018) and from Manaus site K34 used for model parameterisation and evaluation

| Process | Variables | Purpose of use | Reference and site |
|---------|-----------|----------------|--------------------|
| C associated | GPP | Evaluation | Fleischer et al., 2019, K34 |
| | NPP | Evaluation | Fleischer et al., 2019, K34 |
| | Soil C | Evaluation | Malhi et al., 2009, K34 |
| | CUE | Evaluation | Malhi et al., 2009, K34 |
| | Veg C | Evaluation | Study site |
| | Leaf C | Evaluation | Study site |
| | Stem C | Evaluation | Study site |
| | Root C | Evaluation | Study site |
| | LAI | Initialisation | Study site |
| | LMA | Parameterisation | Study site |
| P associated | Resin | Evaluation | Study site |
| | Pi Bic | Evaluation | Study site |
| | Po Bic | Evaluation | Study site |
| | Po NaOH | Calibration | Study site |
| | Pi NaOH | Calibration | Study site |
| | P residual | Parameterisation | Study site |
| | P HCL | Parameterisation | Study site |
| | Leaf N | Parameterisation | Study site |
| | Leaf P | Parameterisation | Study site |
| | Root P | Parameterisation | Study site |
| | Plant C:P ratio | Parameterisation | Study site |

**L469: What does a step increase mean?**

*Step increase refers to a sudden change (increase) in atmospheric $CO_2$ ie. going from ambient to elevated to $CO_2$ conditions in one model timestep. This is now clarified in the revised version as follows:*
"with present-day $CO_2$ based on global monitoring stations, and an abrupt (step) increase in atmospheric $CO_2$ of +200 ppm on the onset of the transient period (i.e., 1999)."

**L664: What is SMCL?**

*Soil moisture. It is corrected in the revised version:*
"The estimated WUE and soil moisture content (SMCL) for the lowest GPP period"

**L767: Jiang et al., not Jing :)**

*I am really sorry for the typo on your name. It has now been corrected.*


**RC2:**
**This study implements phosphorus cycle processes into the JULES land surface model. The authors calibrate and test their model using observation from the Amazon Fertilization Experiment site and perform a theoretical elevated OC2 experiment. The inclusion of the P cycle in models is critical to our ability to predict ecosystem responses to elevated CO2 and climate change in low-P systems such as the tropics and subtropics, and the majority of land surface models are now adding these processes.**

*We thank the reviewer for their positive comments and suggestions. We have addressed the comments as described below.*

**My main concern about this paper is not with the model development per se, which is largely robust, but with the very extensive calibration. The authors calibrate their model very intensively using data from a very well instrumented site, which leaves me wondering if JULES CNP can be applied to any other site. In particular, the leaf C:P ratio, which the authors themselves show is a very sensitive parameter (Fig. 4) can be highly spatially variable and I do not see how the model can be run at sites where this data is not available, not to say anything about globally. Ideally, I would like to see the model validated at a site at which it has not been calibrated, but I understand this can be very difficult as it would involve obtaining more data. Alternatively, the authors could perform a more comprehensive parameter sensitivity analysis and include a discussion of the generality of their model.**

*We agree with the reviewer that our study would benefit from testing model at other sites with available data. However, as the reviewer mentioned, obtaining detailed measurements (similar to study under AFEX project) is challenging. We created a model suitable for global application, and it is appropriate to first test it at a site-level where data are available. At site-level it is standard practise to parameterise and calibrate the model where possible, to be able to then evaluate model performance for key variables. In order to run the JULES CNP at the global scale we need to define PFT-specific: plant stoichiometries (that can be obtained from Zechmeister-Boltenstern et al. (2015)), soil stoichiometries (can be obtained from  Zechmeister-Boltenstern et al. (2015) and Tipping et al. (2016)), sorption and weathering ratios (can be obtained based on lithological class specific from the GliM lithological map (Hartmann and Moosdorf, 2012) and soil shielding from Hartmann et al., (2014)). Following preparation and scaling this information to read in JULES CNP, we can run at the global scale. There are also approaches to extend model application to the regional scale, e.g. relating soil properties with leaf level traits. However, this is not currently in the scope of this study. Nevertheless, our ongoing project aims to apply JULES CNP to the Amazon region using existing soil (Quesada et al., 2011) and foliar structural and nutrient (Fyllas et al., 2009) data for the above-mentioned parameterisation. This is added in lines 827-833 in the discussion section 4:*
"Our new developments include major P processes in both plant and soil pools and can be applied to the Amazon region using existing soil (Quesada *et al.*, 2011) and foliar structural and nutrient (Fyllas *et al.*, 2009) data for parameterisation. Moreover, JULES CNP can be applied at the global scale and for future projections using global soil P data (Sun *et al.*, 2021) for model initialization and PFT-specific plant stoichiometries (Zechmeister-Boltenstern et al. 2015), soil stoichiometries (Zechmeister-Boltenstern et al. 2015; Tipping et al. (2016), sorption and weathering ratios (based on lithological class specific from the GliM lithological map (Hartmann and Moosdorf, 2012) and soil shielding from Hartmann et al., (2014))."

*As suggested by the reviewer we performed an extended parameter sensitivity analysis on the P-related parameters as outlined below. The extended model sensitivity was performed on C:P stoichiometry, P uptake scaling factor, organic and inorganic P adsorption coefficients, and maximum organic and inorganic sorbed P. Since the occluded and weathered P pools are prescribed in the simulations of this study, the occluded and weather P coefficients (other two P-related model parameters) were not included in these sensitivity tests. The R scripts and output files are uploaded on the Zenodo as well.*

*Modifications in methods section lines 595-604:*

"To test the sensitivity of the P and C related processes to the model P parameters, six sets of simulations were conducted with modified plant C:P stoichiometry (Plant C:P: *SENS1*), P uptake scaling factor ($K_P$) (Kp: *SENS2*), inorganic (KP_sorb_in: *SENS3*) and organic (KP_sorb_or: *SENS4*) P adsorption coefficients ($K_{sorp-or}, K_{sorp-in}$), and maximum inorganic (KP_sorb_in_max: *SENS5*) and organic (KP_sorb_or_max: *SENS6*) sorbed P ($K_{or-max}, K_{in-max}$). These values were prescribed to vary between ±50% of the observed values and their effect on C pools (plant and soil C) and fluxes (NPP and excess C), and P pools (plant, soil, and soil sorbed P) was assessed. As the occluded and weathered P pools are prescribed for this model application, the occluded and weather P coefficients (other two P-related model parameters) were not part of sensitivity tests."

*Modifications in results section 3.1.4 on model sensitivity lines 670-690 and added Figure S5 to supporting document:*

"3.1.4 Model sensitivity

The results indicate that among all the corresponding C and P pools and fluxes, the excess C flux – which demonstrates P limitation to growth – shows the highest sensitivity to changes in C:P ratios, $K_P$ and $K_{or-max}, K_{in-max}$. A decrease in plant C:P results in a large increase in excess C. This is due to the higher plant P demand as a result of lower plant C:P ratios. An increase in the uptake factor and maximum sorbed organic and inorganic P also results in an increase in excess C. This is due to the higher uptake demand through higher uptake capacity (due to higher $K_P$) and lower available P for uptake due to higher organic and inorganic sorbed P (due to higher $K_{or-max}, K_{in-max}$). Since the total P in the system is lower than the plant demand, the uptake capacity and sorbed P, higher P limitation is placed on growth (decreasing BP) which results in an increase in excess C and decrease in plant C, but also soil C which is a result of lower litter input (Figure 4). Total soil P shows low sensitivity to changes in plant C:P and uptake factor but high sensitivity to maximum inorganic sorbed P. Moreover, sorbed P shows middle to high sensitivity to maximum organic and inorganic sorbed P respectively (Figure. S5). Nevertheless, organic and inorganic P adsorption coefficients ($K_{sorp-or}, K_{sorp-in}$) show no sensitivity to C and P pools and fluxes. This is due to limiting the organic and inorganic P sorption terms controlled only by maximum sorption, hence no effect applied by organic and inorganic adsorption coefficients.

[Figure]

**Figure. 4-** Model sensitivity test results and corresponding C and P pools and fluxes under ambient $CO_2$.

[Figure]

Fig. S5- Model parameters absolute sensitivity values"

**Additionally, I think the paper would benefit from a more extensive discussion of the implementation of exudates and the knock on effect on CUE. The question of what to do with excess carbon under nutrient limitation is one that all models face and there have been a variety of solutions: respire it (what JULES CNP does too), down-regulate photosynthesis, decrease tissue nutrient content etc. I am not saying that the choice made here is necessarily wrong, but it does have implications for the model results. In particular, changes in carbon use efficiency are a direct result of this modeling choice and do not necessarily have an interpretable meaning.**

*As suggested by reviewer we revised the text and added the following part to the discussion on CUE in lines 812-826:*

"As soil P availability is low in the majority of Amazonia (Quesada *et al.*, 2012), the competition for nutrients by both plant and soil communities is high (Lloyd *et al.*, 2001). The responses of these communities to eCO₂ under P limited conditions remains uncertain (Fleischer *et al.*, 2019). These responses in P enabled models are represented in different ways regarding the excess C which is not used for plant growth due to P limitation. Either growth is directly downregulated taking the minimum labile plant C,N and P (Goll *et al.*, 2017), or photosynthesis is downregulated via $V_{cmax}$ and $J_{max}$ (Comins and McMurtrie, 1993; Yang *et al.*, 2014; Zhu *et al.*, 2016) and finally models like JULES CNP downregulate NPP via respiration of excess carbon that cannot be used for growth due to plant nutrient constraints (Haverd *et al.*, 2018). The estimated CUE depends on the modelling approach. Models that down regulate the photosynthetic capacity and GPP consequently (Comins and McMurtrie, 1993; Yang *et al.*, 2014; Zhu *et al.*, 2016), simulate a positive CUE response to CO₂ fertilization while models that down regulate the NPP and respire the excess C (Haverd *et al.*, 2018) simulate a negative CUE response (Fleischer *et al.*, 2019) which is in line with the studies showing lower CUE when nutrient availability declines (Vicca *et al.*, 2012). However, this remains a major uncertainty in understanding the implication of P limitation on terrestrial biogeochemical cycles."

**Detailed comments:**

**L 241 As far as I understand from the description here, what the authors term 'exudates' is just excess C that is respired by the plants. However, the term normally refers to carbohydrates released by plants into the soil for a potential benefit in additional nutrients. This is a complex process and extremely difficult to include in models, so I am not suggesting the authors include it here, but maybe a different term for this flux can be used here. I am also not clear what 'spread' refers to here.**

*To avoid this confusion, we have replaced "exudates" term with "excess C" throughout the whole text, tables and figures, as also suggested by RC1.*
*The spread term refers to the increase in vegetation cover due to reproduction and recruitment. We further clarify it in the revised version as follows:*
"NPP in JULES is calculated as the difference between GPP and autotrophic respiration. In JULES-CNP, potential NPP represent the amount of C, available for tissue growth (C density increase) on a unit area, and spreading (vegetation cover increase as a result of reproduction and recruitment), ie to increase the area covered by the vegetation type, assuming no nutrient limitation."

**L 367 Table 2 Check the notations here - doe eta_CP refer to litter or soil CP?**
*This refers to soil C:P. As suggested by reviewer #3, we replaced eta_CP with $C:P_{soil}$ in the revised version in line 358-359:*
"C:P ratios of plant (i) (DPM/RPM) ($C:P_{plant}$) and soil (HUM/BIO) ($C:P_{soil}$)"

**L 460 by 1000 times, do you mean 1000 years?**

*We modified the spin-up description in the revised version in lines 553-558 as follows*
"The JULES CNP simulations were initialized following the same methodology as in Fleischer et al., (2019), by the spin-up from1850 recycling climatology to reach equilibrium state (Figure S1) and spin up was performed separately for three versions of JULES (C/CN/CNP) following the same procedure. Furthermore, the transient run was performed for the period 1851-1998 using time-varying $CO_2$ and N deposition fields. Finally, for the extended simulation period (1999-2019) two runs were performed, the first with ambient the second elevated $CO_2$ concentrations"

**L 564 Figure 3 Would it be possible to label the panels more clearly? I have to assume that the one on the left is fluxes and the one on the right, pools?**

*The left plot includes fluxes (BP and Litter C), CUE and pools (leaf and root C (0.2 – 0.4 kg C m$^{-2}$)) (wood, veg and soil C (13-24 kg C m$^{-2}$)). Due to the scale size of C pools (leaf and root will not be visible if they are next to other C pools). However, in order to make it clearer, we modified it and separated the units on two Y axis as follows:*
"

[Figure]

**Figure. 3-** JULES C, CN, CNP modelled vs measured C pools (Leaf, root, wood, Veg and Soil C) (in kg C m$^{-2}$) and fluxes (BP and Litter C) (in kg C m$^{-2}$ yr$^{-1}$) and CUE under ambient $CO_2$. Note that CUE is unitless."

**L 581 Figure 4 Could you discuss why a change in biomass production does not result in a change in plant C?**

*As mentioned above, this figure and discussion is now replaced.*

**RC3:**

**The authors present an attempt at describing the implementation of a phosphorus cycle into a major land surface model and its evaluation. They compare simulated carbon variables with few observations and provide another model-based quantification of P effects on NPP, C stocks under 2 years of elevated CO2 which is compared to existing model predictions. The inclusion of P cycles in ESM is certainly a timely and important endeavor given the growing evidence of the importance of phosphorus cycling on land surface conditions and greenhouse gas fluxes.**

*We thank the reviewer for their constructive comments and for the thorough review of the manuscript. We have addressed the comments as described below.*

**However, my main concern is that this study adds little to existing studies: (1) the evaluation of the model falls short thereby no new insights could be gained;**

*As RC3 states the inclusion of a phosphorous cycle in the land surface model JULES which forms the land component of the UKESM is an important and new development. There are six land surface models with coupled C, N and P cycles however none was included as part of CMIP6. We also recognise that P limitation is likely important, but not limited to the tropical forest regions. This modelling endeavour has been the result of a concerted effort and coordination between modellers and empiricists to develop an appropriate P model for application in a global ESM, and to thoroughly investigate the performance of the model at a tropical forest site where targeted new P-cycle data has been measured for this purpose from project outset. Therefore, this provides new data to feed model development and evaluation for a critical global biome, and the sensitivity test identifies critical parameters.*

*Due to the complexity of ESMs, for incorporation of new processes it is preferable to use parameter-sparse algorithms and/or well known and tested schemes as a first step. We have specifically selected GMD as this is exactly the remit of this journal: 'discussion of the description, development, and evaluation of numerical models of the Earth system and its components'*

*Unique to this work is that calibration of P cycle in JULES CNP was done using actual and novel site measurements from soil and plants as a part of AFEX project. At site-level it is standard practise to parameterise and calibrate the model where possible, to be able to then evaluate model performance for key variables. This is further clarified in abstract section in lines 30-36:*

"We evaluate JULES-CNP using in situ data collected at a low fertility site in the Central Amazon, with a soil P content representative of 60% of soils across the Amazon basin, to parameterise, calibrate and evaluate JULES-CNP. Novel soil and plant P pool observations are used for parameterisation and calibration and the model is evaluated against soil P pools and C fluxes and stocks. We then apply the model under elevated $CO_2$ (600 ppm) at our study site to quantify the impact of P limitation on $CO_2$ fertilization. We compare our results against current state of the art CNP models using the same methodology that was used in the AmazonFACE model intercomparison study."

*And method sections in lines 119-125:*

"Here, we describe the development and implementation of the terrestrial P cycle in the Joint UK Land Environment Simulator (JULES) (Clark *et al.*, 2011), the land component of the UK Earth System Model (UKESM), developed following the structure of the current N cycle in JULES(Wiltshire *et al.*, 2021). The model (JULES-CNP) is parameterized and calibrated using novel in situ P soil and plant data from a well-studied forest site in Central Amazon near to Manaus, Brazil with soil P content representative of 60% of soils across the Amazon basin. We then evaluate the model against P pools (not used for calibration) and C stocks and fluxes from data sets from our study site and the nearby K34 field site.'

Also

*JULES_CNP model structure was designed to match soil P pools for which data are available for the whole Amazon region, with a global data set also available for global applications (Hartmann and Moosdorf, (2012); Hartmann et al. (2014); Zechmeister-Boltenstern et al. (2015); Tipping et al. (2016); Sun et al. (2021)). This is added in lines 827-833 in the discussion section 4:*
"Our new developments include major P processes in both plant and soil pools and can be applied to the Amazon region using existing soil (Quesada *et al.*, 2011) and foliar structural and nutrient (Fyllas *et al.*, 2009) data for parameterisation. Moreover, JULES CNP can be applied at the global scale and for future projections using global soil P data (Sun *et al.*, 2021) for model initialization and PFT-specific plant stoichiometries (Zechmeister-Boltenstern et al. 2015), soil stoichiometries (Zechmeister-Boltenstern et al. 2015; Tipping et al. (2016), sorption and weathering ratios (based on lithological class specific from the GliM lithological map (Hartmann and Moosdorf, 2012) and soil shielding from Hartmann et al., (2014))."

**(2) the eCO2 experiment is a repetition of intermodel comparison of Fleischer et al. (2019) and provides no new insights.**

*Our eCO$_2$ experiment purposely repeated the simulation protocol by Fleisher et al (2019) in order to compare JULES-CNP to the current state of the art P models. In our opinion there is a lot of value in knowing where our predictions (parameterised and constrained with Amazon forest P soil and plant data) lie compared to the current models (which were less data-informed with local site data than the work presented here), thus the well-known value of model intercomparison projects.*
*This is clarified in the text in lines 606-608 as follows:*

"However, in order to perform inter-models comparison with 15 models studied by Fleischer *et al.*, (2019) we also studied the response of GPP, NPP and BP to eCO2 for both initial (1999) and 15 years periods (between 1999-2013)

**Potential new contributions could have been (1) resolving root exudates. But (guessing from the incomplete model description) it seems a simply a relabelling of the 'excess NPP' (i .e NPP which cannot be allocated to new biomass growth (Thornton et al 2007, Goll et al 2012) as now 'root exudates'. On top of that, no attempt has been made at its evaluation**

*The reviewer is correct, the term "exudates" in our model refers to the excess C and this term has been replaced throughout the text. This first version of JULES-CNP does not have the representation of the root exudates which are very difficult to constrain due to challenges in directly measuring rates of exudation. We agree with the reviewer that root exudates need to be included on future developments when there is good data availability for constraining this flux, and information on the role exudates play in changing rates of below-ground nutrient cycling.*

**..and the authors seem to confuse observed BP as NPP (BP + root exudates)..**

*We apologise for the confusion. We now refer to excess C rather than exudates, and given we do not consider the latter, it means that for our purposes NPP = BP. We clarify this in the text in lines 255-260:*
"The reported NPP in the literature often includes other C fluxes related to the exudates, volatiles production and non-structural carbohydrates (Malhi *et al.*, 2009; Chapin *et al.*, 2011; Walker *et al.*, 2021) which are challenging to measure (Malhi, Doughty and Galbraith, 2011). Therefore, actual NPP is for our purposes equal to Biomass Production (BP), and is calculated as potential NPP minus excess C (lost to the plant through autotrophic respiration), with the latter the C that cannot be used to growth new plant tissue due to insufficient plant nutrient supply"

*And lines 270-272*
"Therefore, BP is calculated as the difference between potential NPP ($\Pi_c$) and total excess C:

$$BP = \Pi_c - \psi_t \qquad\qquad\qquad (eq.17)"$$

**Thus, I am not sure anything new has been learnt here about.**

*In summary, the novelty of our work lies on incorporating a P model into the land surface scheme of an ESM, in this case the UKESM. Our modelling framework was developed around novel existing data sets for our study site and well tested equations from other P schemes. As a result, our modelling framework can be applied in future studies for an Amazon basin-wide application making use of existing data sets and can also be applied globally.*

**(2) the use of response of biota to nutrient addition from the AFEX experiment**

*We agree with the reviewer that additional evaluation of JULES-CNP against treatment (fertilization) data from the Amazon nutrient fertilization experiment (AFEX) is of high value. AFEX is an ongoing project and only changes in fine root dynamics due to nutrient additions have been published so far. Therefore, it is not possible evaluate the model against the nutrient addition response yet but this is planned in a future model application study.*

*Given the fact that rightly two reviewers have asked about this, to avoid confusion, we have decided to simply refer to our 'study site' in Amazonia where we have access to key soil and plant data, rather than explicitly elaborate that this is the control site from AFEX.*

*We have modified lines 119-125 at the end of the introduction accordingly:*
"Here, we describe the development and implementation of the terrestrial P cycle in the Joint UK Land Environment Simulator (JULES) (Clark *et al.*, 2011), the land component of the UK Earth System Model (UKESM), following the structure of the prior N cycle development (Wiltshire *et al.*, 2021). The model (JULES-CNP) is parameterized and calibrated using novel in situ P soil and plant data from a well-studied forest site in Central Amazon near to Manaus, Brazil with soil P content representative of 60% of soils across the Amazon basin. We then evaluate the model against carbon stocks and fluxes from data sets from our study site and the nearby K34 field site"

*We also modified the following lines in the methods section 2.3; study sites in lines 460-471*
"This study uses data from two nearby sites in Central Amazon in Manaus, Brazil. The main site from here on termed *study site* (2°35´´21.08´´ S, 60°06´´53.63´´ W) (Lugli *et al.*, 2020) is for model development and evaluation. The second site is the Manaus K34 flux site (2°36´´32.67´´ S, 60°12´´33.48´´ W) which provides meteorological station data for running the model but also provides data for model evaluation.

We use detailed novel soil and plant P pool data from the *study site* (Lugli *et al.*, 2020, 2021) for model parameterisation and calibration and carbon stock data for model validation. The *study site* has a very similar forest, geomorphology, soil chemistry and species composition to the well-known and studied K34 eddy covariance flux site (Araújo *et al.*, 2002)."

**Overall, several shortcomings have been identified which are listed in the following.**

**1 The model description is incomplete**

**it is not clear how the different configurations of JULES differ (C,CN,CNP). In reality the cycles are closely intertwined, so you must have made some simplification to switch them on and off. As interactions between NP cycles are not explained one can only guess how NP affects C fluxes. This prevents the reader from understanding the implications of the model results as major underlying model assumptions are not given.**

*We omitted information on the interaction between N and P in our model description due to the lack of N limitation at our study site. However, in line with the comment from reviewer #1 we modified the manuscript and added details on the N and P interaction in JULES CNP in the methods section in lines 284-318 as follows:*

[revised manuscript text omitted]

**it is not clear which modelling approaches are novel and which are based on concepts from previous studies/models. The majority of process representation seem to be taken from earlier work (like early P work in JSBACH, CLM, ORCHIDEE, CABLE). The authors fail repeatedly to credit earlier works (most references given are related to JULES) and to justify their modelling choice.**

*We sincerely apologise for missing the original sources used on our modelling approach. We revised the text thoroughly and added the original references for each part in Method section 2.2.1- and 2.2.2- lines 285-432 as follows:*

"Description of the plant P pool ($P_p$) follows Zhu *et al.*, (2016) and is estimated as the difference between the input, plant uptake $F_P^{Up}$ (eq.26) and output of this pool, plant litter flux $F_P^{lit}$ (eq.28), with both fluxes expressed in kg P m$^{-2}$ yr$^{-1}$ as follows"

"Description of the inorganic sorbed P pool ($P_{inorg-sorp}$) follows Wang *et al.*, (2007) and is represented as the difference between the input flux of inorganic sorption ($F_{P_{in}}^{sorp}$) (eq. 37) and output fluxes of inorganic desorption ($F_{P_{in}}^{desorp}$) (eq. 38) and occluded P($F_P^{occ}$) (eq. 39), with all fluxes expressed in kg P m$^{-2}$ yr$^{-1}$ as follows"

"Descripting of the occluded ($P_{occ}$) P pool follows Wang *et al.*, (2007) and Hou *et al*., (2019) and is represented as the sum of input fluxes of occluded P from both organic ($F_P^{or-occ}$) (eq. 42) and inorganic P pools ($F_P^{occ}$) expressed in kg P m$^{-2}$ yr$^{-1}$, as follows"

"Descripting of the organic sorbed P pool ($P_{org-sorp}$) follows Wang *et al.*, (2007) and is represented as the difference between the input flux of organic sorption ($F_{P_{O_{S_n}}}^{sorp}$) and output fluxes of organic desorption ($F_{P_{O_{S_n}}}^{desorp}$) and occluded P($F_{P_n}^{occ}$), with all fluxes expressed in kg P m$^{-2}$ yr$^{-1}$ as follows"

"Descripting of P from parent material ($P_{pm}$) pool follows Wang *et al.*, (2007) and depends on the weathering flux ($F_P^w$) (eq. 43) in kg P m$^{-2}$ yr$^{-1}$ as follows"

"Description of the plant P and N demand follow Wang *et al.*, (2007) and are represented by the sum of demand ($\emptyset_t$) to sustain growth (P-related: ($\emptyset_{g_P}$), N-related: ($\emptyset_{g_N}$)) and to sustain vegetation spreading (to sustain PFT fractional coverage increment) (P-related: ($\emptyset_{S_P}$), N-related: ($\emptyset_{S_N}$)) and is expressed in (P-related in kg P m$^{-2}$ yr$^{-1}$; N-related in kg N m$^{-2}$ yr$^{-1}$). The total demand for growth ($\emptyset_g$) and spreading ($\emptyset_s$) is controlled by the dominant demand between P ($\emptyset_{g_P}$) and N ($\emptyset_{g_N}$) as follows"

"Description of the plant P uptake ($F_p{}^{up}$) varies spatially depending on the root uptake capacity ($u^{max}$) followed by Goll *et al.*, (2017). Therefore, in regions with limited P supply, the plant P uptake is limited to the $u^{max}$ and consequently impacts the excess C and BP."

"Description of the litter production of P ($F_{P_n}{}^{lit}$) (arrow b in Fig 1) follows JULES-CN as in Wiltshire *et al.*, (2021) and is calculated based on the litter flux of C (kg C m$^{-2}$ yr$^{-1}$) using leaf, root and wood turnovers (yr$^{-1}$), and through the vegetation dynamics due to large-scale disturbance and litter production density, as follows".

"where the description of $F_{P_n}$ for P pools ($F_{PP_n}$) follows Wang *et al.*,(2007) and is estimated based on the soil pool (BIO/HUM) mineralization ($minl_{P-BIO_n}$, $minl_{P-HUM_n}$) and immobilization ($immob_{P-BIO_n}$, $immob_{P-HUM_n}$) (in kg P m$^{-2}$ yr$^{-1}$), soil inorganic P ($P_{inorg_n}$) (in kg P m$^{-2}$), and litter pools (DPM/RPM) demand (in kg P m$^{-2}$ yr$^{-1}$) as follows"

"Description of the fluxes of adsorption ($F_{P_{in_n}}{}^{sorp}$) (arrow e in Fig 1) and desorption ($F_{P_{in_n}}{}^{desorp}$) (arrow f in Fig 1) of inorganic P in kg P m$^{-2}$ yr$^1$ follow Wang *et al.*, (2010) and are calculated based on soil inorganic ($P_{in_n}$) and sorbed inorganic ($P_{inorg-sorbed_n}$) P pools and inorganic adsorption ($K_{sorp-in}$), desorption ($K_{desorp-in}$) coefficients (kg P m$^{-2}$ yr$^{-1}$) and maximum sorbed inorganic ($P_{in-max}$) (kg P m$^{-2}$) as follows".

"Description of the occluded inorganic P flux ($F_{P_n}{}^{occ}$) (arrow g in Fig 1) follows Wang *et al.*, (2007) and Hou *et al.*, (2019) and is calculated based on sorbed inorganic P pool and P occlusion rate ($K_{occ}$) (kg P m$^{-2}$ yr$^{-1}$) as follows".

"Description of the fluxes of adsorption ($F_{PO_{S_n}}{}^{sorp}$) (arrow h in Fig 1) and desorption ($F_{PO_{S_n}}{}^{desorp}$) (arrow i in Fig 1) of organic P follow Wang *et al.*, (2010) are calculated based on soil organic and sorbed organic P pools and organic adsorption ($K_{sorp-or}$) (kg P m$^{-2}$ yr$^{-1}$), desorption ($K_{desorp-or}$) coefficients (kg P m$^{-2}$ yr$^{-1}$) and maximum sorbed organic ($P_{org-max}$) (which corresponds to the sorbed soil P saturation, thus modifying the sorption rate respectively) (kg P m$^{-2}$) as follows".

"Description of the occluded organic P flux ($F_{P_n}{}^{or-occ}$) (kg P m$^{-2}$ yr$^{-1}$) (arrow j in Fig 1) follows Wang *et al.*, (2007) and Hou *et al.*, (2019) is calculated based on sorbed organic P pool ($P_{org-sorbed_n}$) and P occlude rate ($K_{occ}$) (kg P m$^{-2}$ yr$^{-1}$) as follows"

"Description of the P flux from weathered parent material ($F_{P_n}{}^{w}$) (arrow k in Fig 1) follows Wang *et al.*, (2007) and is calculated based on amount of P in the parent material ($P_{pm}$) and P weathering rate ($K_w$) (kg P m$^{-2}$ yr$^{-1}$) as follows"

"Description of P diffusion between soil layers ($F_{D_n}$) expressed in (kg P m$^{-2}$ yr$^{-1}$) (arrow l in Fig 1) follows Goll *et al.*, (2017) and is calculated following Fick's second law and it is a function of the diffusion coefficient ($Dz$) in m$^2$ s$^{-1}$, the concentration of inorganic P at different soil depths ($P_{in}$) in kg P m$^{-2}$, the distance ($z$) between the midpoints of soil layers in metres and seconds to year unit conversion ($Yr$)"

**The presentation of model equations is poor making and many inaccuracies make it hard to follow (see minor points related to eq listed below).**

*We are sorry for some inaccuracies in the equations. We revised the equations where the reviewer commented and other equations in line with those modifications. Details below under "more specific comments".*

**Model input parameteres are not given**

*We revised the Table 2 and added all the model parameters as follows:*

**Table 2.** P Model parameters

| Parameter | Value | Unit | Eq. | Description | Source |
|---|---|---|---|---|---|
| **C and N related** | | | | | |
| $\alpha$ | 0.25 | - | 6 | Plant type material ratio | (Clark *et al.*, 2011) |
| $a_{wl}$ | 1.204 | kg C m$^{-2}$ | 50 | Allometric coefficient | calibrated |
| $\sigma_l$ | 0.0375 | kg C m$^{-2}$ per unit LAI | 48 | Specific leaf density | Clark *et al.*, 2011 |
| $b_{wl}$ | 1.667 | - | 50 | Allometric exponent. | Clark *et al.*, 2011 |
| $f_{dr}$ | 0.005 | - | 47 | Respiration scale factor | Calibrated |
| $resp\_frac$ | 0.25 | - | 32 | Respiration fraction | (Clark *et al.*, 2011) |
| $k_{leaf}$ | 0.5 | - | 28 | Leaf N re-translocation coefficient | (Zaehle and Friend, 2010) |
| $k_{root}$ | 0.2 | - | 28 | Root N re-translocation coefficient | (Zaehle and Friend, 2010) |
| $d_{root}$ | 3.0 | - | 27 | Root fraction in each soil layer | (Clark *et al.*, 2011) |
| $v_{int}$ | 7.21 | μmol CO$_2$ m$^{-2}$ s$^{-1}$ | 45 | Intercept in the linear regression between $V_{cmax}$ and $N_{area}$ | Calibrated (Clark *et al.*, 2011) |
| $v_{sl}$ | 19.22 | μmol CO$_2$ gN$^{-1}$ s-1 | 45 | Slope in the linear regression between $V_{cmax}$ and $N_{area}$ | Calibrated (Clark *et al.*, 2011) |
| LMA | 131.571852 | g m-2 | 45 | Observed Leaf Mass per Area | Study site |
| Leaf N | 1.79007596 | g g-1 | 45, 46 | Foliar N concentrations in area basis | Study site |
| **P related** | | | | | |
| $C{:}P_{soil}$ | 1299.6 | - | 32 | Soil C:P ratio | (Fleischer *et al.*, 2019) |
| $v_{max}$ | 0.0007 | kg P kg$^{-1}$ C yr$^{-1}$ | 27 | Maximum root uptake capacity | Calibrated (Goll *et al.*, 2017) |
| P | 0.7083062 | g kg$^{-1}$ | 46 | Foliar P concentrations | Study site |
| $c_f$ | 3.1×10$^{-5}$ | 1 kg P$^{-1}$ | 27 | Conversion factor | (Goll *et al.*, 2017) |
| $D_z$ | 0.001 | m$^2$ s$^{-1}$ | 44 | Diffusion coefficient | (Burke *et al*, 2017) |
| $K_{occ}$ | 1.2×10$^{-5}$ | yr$^{-1}$ | 39, 42 | P occlusion rate | (Yang *et al.*, 2014) |
| $K_p$ | 3.0 | kg P l$^{-1}$ | 27 | Scaling uptake ratio | Calibrated |
| $K_{sorp-in}$ | 0.0054 | kg P m$^{-2}$ yr$^{-1}$ | 37 | Inorganic P adsorption coefficient | Calibrated (Hou *et al.*, 2019) |
| $K_{sorp-or}$ | 0.00054 | kg P m$^{-2}$ yr$^{-1}$ | 40 | Organic P adsorption coefficient | Calibrated |
| $K_{in-max}$ | 0.0075 | kg P m$^{-2}$ yr$^{-1}$ | 37 | Maximum sorbed inorganic P | Study site |
| $K_{or-max}$ | 0.0042 | kg P m$^{-2}$ yr$^{-1}$ | 40 | Maximum sorbed organic P | Study site |
| $K_w$ | 3×10$^{-6}$ | kg P m$^{-2}$ yr$^{-1}$ | 43 | P weathering rate | (Wang *et al.*, 2010) |

**2 Some of the assumptions / choices are in contrast to current understanding and consensus while no explanation was given.**

**The assumption that CNP cycles are in steady-state with present day conditions (1999-2019) is not appropriate. There are multiple lines of arguments, that the historic increase in CO2 has led to a progressive limitation of nutrients (e,g, Luo et al 2004, Goll et al 2012, Penuelas et al 2013) and that present day land carbon cycle is not in equilibrium.  The non-steady state of the present day CNP cycles is accounted for in the majority of  modelling exercises (including Fleischer et al 2019. Trendy modelling protocol). The historic increase in CO2 is likely the more dominant factor affecting the**

**present day state of C cycle compared to (progressive) NP limitation. Thus this omission is a major shortcoming, in particular as model predictions which account for this exist (i.e. Fleischer et al 2019). You should at least test what the implications of omitting this on the results are / better redo the whole experiment.**

*The information on the initial spin-up process was missed in our manuscript hence the comment raised by the reviewer. Please note that we followed the same methodology as in* Fleischer *et al.,* (2019) *and follows the Trendy protocol. Thus, 1000 years spin-up recycling climatology was performed for the year 1850 to reach the equilibrium (Figure S1). This is followed by a transient run (1851-1998), using time-varying CO2 and N deposition. Finally, for the extended simulation period (1999-2019) two runs were performed, the first with ambient the second elevated CO2 concentrations. Note that the spin up was performed separately for three versions of JULES (C/CN/CNP) following the same procedure.  This information is added to method section 2.5 in lines 553-558 as follows:*

"The JULES CNP simulations were initialized following the same methodology as in Fleischer et al., (2019), by the spin-up from1850 recycling  climatology  to reach equilibrium state (Figure S1) and spin up was performed separately for three versions of JULES (C/CN/CNP) following the same procedure. Furthermore, the transient run was performed for the period 1851-1998 using time-varying $CO_2$ and N deposition fields. Finally, for the extended simulation period (1999-2019) two runs were performed, the first with ambient the second elevated $CO_2$ concentrations."

**There are several highly uncertain parameters in your model. It is not clear why you varied in the sensitivity test, only (a few) parameters which happen to be among the few observed ones and not choose more uncertain parameters? Besides, the impact of varying stoichiometry has been investigated in earlier models with a comparable plant P cycle (Goll et al 2012).**

*We agree with the reviewer that further sensitivity test on the P-related parameters beyond C:P ratios is useful. Considering this comment and a suggestion from reviewer #2, we expanded the parameter sensitivity analysis to include other P-related parameters such as the P uptake scaling factor, organic and inorganic P adsorption coefficients, and maximum organic and inorganic sorbed P. We did not include the occluded and weathered P coefficients as their respective P pools are prescribed in the current study. The R scripts and output files are uploaded on the Zenodo as well.*

*Modifications in methods section lines 595-604:*

"To test the sensitivity of the P and C related processes to the model P parameters, six sets of simulations were conducted with modified plant C:P stoichiometry (Plant C:P: *SENS1*), P uptake scaling factor ($K_P$) (Kp: *SENS2*), inorganic  (KP_sorb_in: *SENS3*) and organic (KP_sorb_or: *SENS4*) P adsorption coefficients ($K_{sorp-or}, K_{sorp-in}$), and maximum inorganic (KP_sorb_in_max: *SENS5*) and organic (KP_sorb_or_max: *SENS6*) sorbed P ($K_{or-max}, K_{in-max}$). These values were prescribed to vary between ±50% of the observed values and their effect on C pools (plant and soil C) and fluxes (NPP and excess C), and P pools (plant, soil, and soil sorbed P) was assessed. As the occluded and weathered P pools are prescribed for this model application, the occluded and weather P coefficients (other two P-related model parameters) were not part of sensitivity tests."

*Modifications in results section 3.1.4 on model sensitivity lines 670-690 and added Figure S5 to supporting document:*

"3.1.4 Model sensitivity

The results indicate that among all the corresponding C and P pools and fluxes, the excess C flux – which demonstrates P limitation to growth – shows the highest sensitivity to changes in C:P ratios, $K_P$ and $K_{or-max}, K_{in-max}$. A decrease in plant C:P results in a large increase in excess C. This is due to the higher plant P demand as a result of lower plant C:P ratios. An increase in the uptake factor and maximum sorbed organic and inorganic P also results in an increase in excess C. This is due to the higher uptake demand through higher

uptake capacity (due to higher $K_P$) and lower available P for uptake due to higher organic and inorganic sorbed P (due to higher $K_{or-max}$, $K_{in-max}$). Since the total P in the system is lower than the plant demand, the uptake capacity and sorbed P, higher P limitation is placed on growth (decreasing BP) which results in an increase in excess C and decrease in plant C, but also soil C which is a result of lower litter input (Figure 4). Total soil P shows low sensitivity to changes in plant C:P and uptake factor but high sensitivity to maximum inorganic sorbed P. Moreover, sorbed P shows middle to high sensitivity to maximum organic and inorganic sorbed P respectively (Figure. S5). Nevertheless, organic and inorganic P adsorption coefficients ($K_{sorp-or}$, $K_{sorp-in}$) show no sensitivity to C and P pools and fluxes. This is due to limiting the organic and inorganic P sorption terms controlled only by maximum sorption, hence no effect applied by organic and inorganic adsorption coefficients.

[Figure]

**Figure. 4-** Model sensitivity test results and corresponding C and P pools and fluxes under ambient $CO_2$.

[Figure]

Fig. S5- Model parameters absolute sensitivity values"

**Some processes usually included in model have been omitted without giving any rationale why. E.g. Why is phosphatase mediated mineralization being omitted? It seems all major P models account for it. I don't imply the author must account for it, but if they choose not to, an explanation should be given. However, Fig2 indicates modelled soil organic P is quite high compared to observation, which would point towards missing biochemical mineralisation (which reduces organic P by enhancing its turnover) is problematic;**

*We now clarify the rationale behind choice of P-related process formulation in JULES CNP and future required developments. However, the current version of JULES CNP only includes the total mineralization. Nevertheless, identifying the mineralization versus uptake is challenging. For instance, in the litter layer the fine roots can facilitate both litter decomposition and uptake of mineralized P* (Martins *et al.*, 2021). *Moreover, even the models which include the biochemical process show no significant difference between biochemical P mineralization to total P mineralization (Goll et. al., 2017). This is further clarified in the revised version in lines 896-899 as follows:*
"Moreover, biochemical mineralisation is not included in the current version of JULES CNP and it only accounts for total mineralization. However, even the models which includes this process, show no significant difference between total and biochemical mineralized P which can be due to complexity of identifying the inclination of mineralization versus uptake (Martins *et al.*, 2021)."

**Why is plant internal nutrient and carbon storage being omitted? Previous modelling studies showed the importance of accounting for such storage pools, see e.g. Yang, Xiaojuan, et al. "Global evaluation of ELM v1 and the role of the phosphorus cycle and non-structural carbon in the historical terrestrial carbon balance." AGU Fall Meeting Abstracts. Vol. 2020. 2020.**

*Please note that this is beyond the scope of the present study. A recent development of a Non-Structural Carbohydrate (NSC) model, SUGAR, for implementation in JULES has recently been developed* (Jones *et al.*, 2020) *and our intention in future work will be to explicitly consolidate JULES CNP and SUGAR into a single model framework. This is further clarified in the revised version in lines 900-902 as follows:*
"Lastly, in order to capture plant internal nutrient impact on the C storage, the future work should focus on implanting a recent developed Non-Structural Carbohydrate (NSC) model (SUGAR) (Jones *et al.*, 2020) in JULES-CNP."

**What about N losses like leaching or erosion, inputs from atmospheric deposition? Are they omitted? why?**

*JULES does not include a representation of erosion. However, JULES does include representation of N leaching, mineralized N gas emission, fixed N and atmospheric. Hence, in the revised version, we provide the relevant figure and text as follows:*

*In methods section 2.2 – lines 164-166:*
"However, despite JULES-CN that includes N leaching and deposition, P leaching and deposition are omitted in the current version of JULES-CNP."

*And*

*In results section 3.1.2 – lines 633-634:*
JULES CNP-CNP could reproduce the plant and soil C (Figure.2 and Table 5) and N pools and fluxes (Figure S6 and Table 6) pools and fluxes under ambient $CO_2$.

*And*

*In discussion section 4.1 – lines 888-895:*

"Moreover under low P availability, all available P is considered to be adsorbed or taken by plant and microbes for further consumption, with leaching considered to be minor within the time scales of our study period (Went and Stark, 1968; Bruijnzeel, 1991; Neff, Hobbie and Vitousek, 2000).

Due to the strong fixation of P in the soil (Aerts & Chapin, 2000; Goodale, Lajtha,Nadelhoffer, Boyer, & Jaworski, 2002), the P deposited is unlikely to be available to plants in the short term (de Vries et al., 2014), for this reason this version of JULES CNP did not include P deposition. However both P deposition and leaching are likely to have a very important role on sustaining the productivity of tropical forests in the Amazon over longer time scales (Van Langenhove *et al.*, 2020) and needs to be considered in future studies."

*And*

*In supporting document:*

"Our results show the highest N leaching in year 2017 at 0.34 g N m$^{-2}$ yr$^{-1}$ and averaged 0.025 g N m$^{-2}$ yr$^{-1}$ for the period 2017-2019. Input from N deposition comes from Fleischer et al (2019) and is fixed at a rate of 0.32 g N m$^{-2}$ yr$^{-1}$ and the averaged fixed N and mineralized gas emissions are set at 2.02 and 0.23 g N m$^{-2}$ yr$^{-1}$, respectively.

[Figure]

Fig. S6- N leaching, mineralized gas emission, fixed and deposition under ambient $CO_2$ condition

Table S2. N pools and fluxes under ambient $CO_2$ condition

| N pools and fluxes | |
|---|---|
| Organic N (kg N m$^{-2}$) | 0.71 |
| Inorganic N (kg N m$^{-2}$) | 0.004 |
| Litter N flux (kg N m$^{-2}$ yr$^{-1}$) | 0.006 |
| Leaf N (kg N m$^{-2}$) | 0.008 |
| Root N (kg N m$^{-2}$) | 0.0066 |
| Stem N (kg N m$^{-2}$) | 0.009 |

**3 The evaluation is insufficient and conducted poorly**

**The observed NPP is based on biomass increments (Fleischer et al 2019; SI table 1). It should be thus referred to BP and not NPP. Also the observed CUE is BP/GPP and NPP/GPP. There is little support that CNP performs better than the C version.**

*Our purposes NPP = BP, we clarify this in the previous comment (page 15 of this document) and also the following added text in lines 584-593:*

"Moreover, we also estimated the Carbon Use Efficiency (CUE) as an indicator of the required C for the growth (Bradford and Crowther, 2013) as follows:

$$CUE = BP/GPP \qquad\qquad (eq.54)$$

We use JULES-CNP to evaluate the extent of P limitation under ambient and $eCO_2$ at this rainforest site in Central Amazon. P limitation is represented by the amount of C that is not used to grow new plant tissue due to insufficient P in the system (excess C) (eq. 27). The excess C flux is highly dependent on the plant P and the overall P availability to satisfy demand. We also explore the distribution of the inorganic and organic soil P and their sorbed fraction within the soil layer and under ambient and $eCO_2$."

*We modified table 5 following your comment and reported the measured NPP as BP and previous mentioned NPP as NPP$_{pot}$ as follows:*

**Table 5.** Observed and simulated carbon pools and fluxes with JULES CNP (between period 2017-18)

| Carbon pools and fluxes | | | |
|---|---|---|---|
| | **Measured** | **Modelled Ambient CO$_2$** | **Modelled Elevated CO$_2$** |
| **GPP** (kg C m$^{-2}$ yr$^{-1}$) | 3.0 – 3.5 | 3.06 | 3.9 |
| **NPP$_{pot}$** (kg C m$^{-2}$ yr$^{-1}$) | - | 1.27 | 1.77 |
| **Plant respiration** (kg C m$^{-2}$ yr$^{-1}$) | 1.98 | 1.78 | 2.12 |
| **Excess C flux** (kg C m$^{-2}$ yr$^{-1}$) | - | 0.30 | 0.81 |
| **Biomass Production** (kg C m$^{-2}$ yr$^{-1}$) | 1.14±0.12 | 0.96 | 0.94 |
| **Litter C flux** (kg C m$^{-2}$ yr$^{-1}$) | 0.69±0.15 | 0.91 | 0.83 |
| **Leaf C** (kg C m$^{-2}$) | 0.37±0.2 | 0.38 | 0.40 |
| **Wood C** (kg C m$^{-2}$) | 22.01 | 22.4 | 24.71 |
| **Root C** (kg C m$^{-2}$) | 0.37±0.2 | 0.38 | 0.40 |
| **Vegetation C** (kg C m$^{-2}$) | 22.75±0.3 | 23.16 | 25.52 |
| **Soil C stock** (kg C m$^{-2}$) | 12.7 | 13.2 | 12.71 |
| **LAI** (m$^2$ m$^{-2}$) | 5.6±0.36 | 5.77 | 6.12 |

*Furthermore, please note that simulated BP and litter flux of C by JULES C/CN are higher than in JULES-CNP but also overestimate the observations (litter flux of JULES C/CN: 1.18, JULES CNP: 0.91 and obs 0.69 (kg C m$^{-2}$ yr$^{-1}$) and BP of JULES C/CN: 1.24, JULES CNP: 0.96 and obs1.14-1.31 (kg C m$^{-2}$ yr$^{-1}$), respectively). By including the P cycling in JULES an excess C flux of 0.3 (kg C m$^{-2}$ yr$^{-1}$) is simulated, indicating a 24% P limitation to BP at this site according to JULES CNP, which represents a 29% decrease in BP compared to JULES-C/CN. Consequently, the total vegetation C stock for models without P inclusion is higher than the CNP version (+3% difference) due to the lack of representation of P limitation. The simulated soil C stock in JULES C and JULES CN is also higher than in the CNP version (JULES C/CN: 13.93 vs. JULES CNP: 13.18 (kg C m$^{-2}$ yr$^{-1}$)) and higher than the observations. Moreover, CUE in JULES C/CN (eq.42) is higher than observations and JULES CNP version (JULES C/CN: 0.38 vs. JULES CNP: 0.31, obs: 0.34 ±0.1(dimensionless).*

**The short period of 2 years considered in the evaluation is inappropriate. Fluxes like NPP and GPP show large interannual variations. The fit of modelled fluxes with a long-term mean of fluxes (e.g GPP from Fleischer et l 2019) could be merely by chance. The justification for using only two years ( line 470 ) is not plausible as you don't use only the soil P measurement for evaluations (see first point). You should evaluate modelled variables over a longer time period.**

*We agree with reviewer that our study could benefit with the longer period for evaluation of fluxes such as GPP and NPP. However, we are not aware of any available measurements of such fluxes for such comparison. Nevertheless, when we compared our results with Fleischer et al., (2019) our simulations included both initial (1999) and 15 years periods (between 1999-2013), studying the relative effect of eCO2 on simulated for GPP and BP fluxes.*

**The soil P measurements were used to calibrate the model not to evaluate (misleading labelling in table 3)**

*Please note that, the plant P pools are calculated based on observations of C:P ratios, and soil P pools have been either prescribed (occluded and weathered) or calibrated via a single parameter (maximum sorption capacity) (sorbed organic and inorganic P) or evaluated (organic and inorganic P). Moreover, the model is able to reproduce observed fluxes and stock of C. We have now corrected Table 3 and clarified this in lines 32-34 as follows:*
"Novel soil and plant P pool observations are used for parameterisation and calibration and the model is evaluated against C fluxes and stocks, and for those soil P pools not used for parameterisation/calibration."

*And lines 515-517:*
"JULES-CNP has fixed stoichiometry and C:P ratios of leaf and root (measured), and wood (estimated from fresh coarse wood (Lugli, 2013)) which were taken from the *study site* and prescribed in JULES to simulate P dynamics in the plant"

*And lines 530-532:*
"Thus, in order to cap P sorption and uptake capacity, the maximum sorption capacities ($P_{in-max_n}$, $P_{or-max_n}$, eq.37 and 39) (adopted from (Wang, Houlton and Field, 2007)) were prescribed using maximum observed sorbed inorganic and organic P."

*And 533-535:*
"Moreover, as the magnitude of changes in the occluded and parent material pools are insignificant over a short-term (20 years) simulation period (Vitousek *et al.*, 1997), these two pools were prescribed using observations."

**There are several datasets available to evaluate nutrient cycles in ESM. Some of the remote sensing products have a spatial resolution sufficiently high to compare to site simulations.  e.g. Sun et al 2020, Hou et al 2020. Sun, Y., Goll, D. S., Chang, J., Ciais, P., Guenet, B., Helfenstein, J., Huang, Y., Lauerwald, R., Maignan, F., Naipal, V., Wang, Y., Yang, H., and Zhang, H.: Global evaluation of the nutrient-enabled version of the land surface model ORCHIDEE-CNP v1.2 (r5986), Geosci. Model Dev., 14, 1987–2010, https://doi.org/10.5194/gmd-14-1987-2021, 2021.**

*Thank you for your comment and suggested reference. Please note that Hou et al 2020: Includes 4 sites in Brazil (3 croplands and 1 perennial) hence is not comparable to our study site. Moreover, it includes P addition experiments which is not part of this study.*
*However, we used the other reference suggested by reviewer, and compared JULES CNP with ORCHIDEE CNP and added the following table and figure to the supporting document:*

Table S3. JULES CNP vs ORCHIDEE CNP P pools and fluxes

| | This study | ORCHIDEE CNP |
|---|---|---|
| Organic P (kg P m$^{-2}$) | 0.007 | 0.01 |
| Plant P (kg P m$^{-2}$) | 0.0046 | 0.0054 |
| Total sorbed P (g P m$^{-2}$) | 3.44 | 3.06 |
| P uptake (g P m$^{-2}$ day$^{-1}$) | 0.0003 | 0.0004 |

Figure. S7- P pools and fluxes provided by ORCHIDEE CNP and study site using JULES CNP

**There seems to be eddy covariance tower nearby. Why hasn't been any data other than long-term everage GPP being used to evaluate the model?**

*Indeed, we agree with the reviewer that our paper would benefit from such a comparison. But please note that this data is not available. Furthermore, Fleischer et al 2019. only received a single reported value for GPP from this site via personal communication. Thus, unfortunately we are unable to perform such a comparison.*

**It is misleading to refer to a control plot of AFEX as a fertilizer experiment. I assume there is no data available from the fertilizer experiment, otherwise you should take advantage of this data to evaluate the model (see main point). I would suggest dropping AFEX and referring to the plots as nearby plots of K34 tower.**

*We agree and to avoid confusion we have decided to simply refer to our 'study site' in Amazonia where we have access to key soil and plant data, rather than explicitly elaborate that this is the control site from AFEX.*

*We have modified lines 119-125 at the end of the introduction accordingly:*

"Here, we describe the development and implementation of the terrestrial P cycle in the Joint UK Land Environment Simulator (JULES) (Clark *et al.*, 2011), the land component of the UK Earth System Model (UKESM), following the structure of the prior N cycle development (Wiltshire *et al.*, 2021). The model

(JULES-CNP) is parameterized and calibrated using novel in situ P soil and plant data from a well-studied forest site in Central Amazon near to Manaus, Brazil with soil P content representative of 60% of soils across the Amazon basin. We then evaluate the model against carbon stocks and fluxes from data sets from our study site and the nearby K34 field site."

*We also modified the following lines in the methods section 2.3; study sites in lines 460-471:*

"This study uses data from two nearby sites in Central Amazon in Manaus, Brazil. The main site from here on termed *study site* (2°35′′21.08′′ S, 60°06′′53.63′′ W) (Lugli *et al.*, 2020) is for model development and evaluation. The second site is the Manaus K34 flux site (2°36′′32.67′′ S, 60°12′′33.48′′ W) which provides meteorological station data for running the model but also provides data for model evaluation.

We use detailed novel soil and plant P pool data from the *study site* (Lugli *et al.*, 2021) for model parameterisation and calibration and carbon stock data for model for validation. The *study site* has a very similar forest, geomorphology, soil chemistry and species composition to the well-known and studied K34 eddy covariance flux site (Araújo *et al.*, 2002)."

**more specific points:**

**Model description**

**Which processes / C Fluxes are affected by N and P limitation?  How are interactions among N P limitations accounted for? Does P affect N fluxes and vice versa?**

*Please find our answer and corrections in response to your general comments regarding the N:P limitations and affected fluxes (page 16-17 of this document).*

**Eq6,7,9,10: the flux decPi,n - you should specify the i here - otherwise it seems the same fluxes are substrate from two different pools / i.e double accounting of fluxes.**

*We modified these equations as follows:*

"The litter P pool ($P_{O_l}$) is estimated as a sum of $P_{DPM}$ and $P_{RPM}$ pools. Each pool is formed by the fluxes of plant litter input ($F_P^{lit}$) and the outgoing decomposed P ($dec_P^{lit}$)  both expressed in kg P m$^{-2}$ yr$^{-1}$ (eq.28-29). Furthermore, the plant litter input is modified based on the plant type material ratio $\alpha$ (in order to distribute the litter input based on the DPM/RPM fraction) as follows:

$$\frac{dP_{DPM}}{dt} = F_{P_n}^{lit} \times \alpha - dec_{P_{DPM,n}} \qquad (eq.6)$$

$$\frac{dP_{RPM}}{dt} = F_{P_n}^{lit} \times (1-\alpha) - dec_{P_{RPM,n}} \qquad (eq.7)$$

$$P_{O_l} = \sum_{n=1}^{N} P_{DPM_n} + \sum_{n=1}^{N} P_{RPM_n} \qquad (eq.8)$$

The soil organic pool ($P_{O_s}$) is represented as the sum of $P_{BIO}$ and $P_{HUM}$. These pools are estimated from the difference between P inputs from total immobilized ($F_{immob_P}$) distributed between BIO and HUM based on fixed fraction (0.46 for BIO, 0.54 for HUM) (Jenkinson *et al.*, 1990; Jenkinson and Coleman, 2008) and desorbed P $F_{P_{O_s}}^{desorp}$ and P outputs from mineralized ($F_{minl_P}$), and adsorbed P fluxes ($F_{P_{O_s}}^{sorp}$) (adsorption: eq. 40 and desorption: eq.41) with all fluxes expressed in kg P m$^{-2}$ yr$^{-1}$ as follows:

$$\frac{dP_{BIO}}{dt} = 0.46 \times F_{immob_{P_n}} + F_{P_{O_{S_{BIO,n}}}}^{desorp} - F_{minl_{P_{BIO,n}}} - F_{P_{O_{S_{BIO,n}}}}^{sorp} \qquad (eq.9)$$

$$\frac{dP_{HUM}}{dt} = 0.54 \times F_{immob_{P_n}} + F_{P_{O_{S_{HUM,n}}}}^{desorp} - F_{minl_{P_{BIO,n}}} - F_{P_{O_{S_{HUM,n}}}}^{sorp} \qquad (eq.10)$$

$$P_{O_s} = \sum_{n=1}^{N} P_{BIO_n} + \sum_{n=1}^{N} P_{HUM_n} \qquad (eq.11)"$$

**Eq9,10: What are the factors 0.46 and 0.54 - how are they derived? They should be parameters listed in Table2.**

*The modification regarding this is given above.*

**Eq11: It is not clear how the delta P_O_S equals the sum of all P subpools. Do you mean the change of all P subpools?**

*Yes, the reviewer is correct, apologies. P_O_l and P_O_S are sum of the subpools after changes. Hence, we modified the P_O_l and P_O_S as follows:*

"$P_{O_l} = \sum_{n=1}^{N} P_{DPM_n} + \sum_{n=1}^{N} P_{RPM_n}$ (eq.8)

$P_{O_S} = \sum_{n=1}^{N} P_{BIO_n} + \sum_{n=1}^{N} P_{HUM_n}$ (eq.11)"

**205-207: can you indicate the equation describing the fluxes (as done before e.g. line 195)**

*We modified the text as follows:*

"Description of the inorganic sorbed P pool ($P_{inorg-sorp}$) follows Wang *et al.*, (2007) and is represented as the difference between the input flux of inorganic sorption ($F_{P_{in}}{}^{sorp}$) (eq. 37) and output fluxes of inorganic desorption ($F_{P_{in}}{}^{desorp}$) (eq. 38) and occluded P($F_P{}^{occ}$) (eq. 39), with all fluxes expressed in kg P m$^{-2}$ yr$^{-1}$ as follows"

*Furthermore, in the revised version we also added the referenced equation throughout the method where it was originally missed to keep the consistency.*

**The calculation is missing for Total exudates, and the subcomponents related to growth and spread (eq16)**

*We agree with the reviewer that these calculations were missed in the text. We revised the method accordingly – lines 305-318 as follow:*

"Equations 20 and 22 are solved by first setting $\psi_g = 0.0$ to find the total plant P (eq. 20) and N demand (eq.22). If the P and N demand for growth are less than the available P and N and fractional coverage ($\lambda$) (NPP fraction used for fractional cover increment; for detail see Wiltshire *et al.,* (2021)) at the considered timestep $\Delta t$ then there is no limitation to growth ($i.e. \emptyset_{gP} < \frac{(1-\lambda)P_{avail}}{\Delta t}$; $\emptyset_{gN} < \frac{(1-\lambda)N_{avail}}{\Delta t}$),. Where there is limited P and/or N availability, the uptake equals the available P and N ($\emptyset_{gP} = \frac{(1-\lambda)P_{avail}}{\Delta t}$; $\emptyset_{gN} = \frac{(1-\lambda)N_{avail}}{\Delta t}$), and the plant growth which cannot be achieved due to nutrient constraints will be deducted from potential NPP, here termed excess C term ($\psi_g$), to give an actual NPP. Following Wiltshire et al., 2021, we assume excess C is respired by the plant.

Similarly, in order to estimate the P and N demand for spreading (eq. 21 and 23), initially the excess C from spreading is set to 0.0 ($\psi_s = 0.0$), i.e under the assumption that there is no nutrient limitation. If the P and N demand for spreading are lower than the available P and N and fractional coverage ($\lambda$) ($\emptyset_{SP} < \frac{(1-\lambda)P_{avail}}{\Delta t}$; $\emptyset_{SN} < \frac{(1-\lambda)N_{avail}}{\Delta t}$), then there is no limitation on spreading and in case of limited P and N availability, the uptake equals the available P and N ($\emptyset_{SP} = \frac{(1-\lambda)P_{avail}}{\Delta t}$; $\emptyset_{SN} = \frac{(1-\lambda)N_{avail}}{\Delta t}$), and the excess C for spread ($\psi_s$) is subtracted from potential NPP."

**Plant P:C ratio (eq20)**

*We added these estimations in the revised version as follows:*

"$\emptyset_{gP} = \frac{P_p}{C_V}\left(\Pi_c - \frac{dC_v}{dt} - \psi_g\right)$ (eq.20)

$\emptyset_{SP} = \frac{P_p}{C_V}\left(\Pi_c - \frac{dC_v}{dt} - \psi_s\right)$ (eq.21)

$\emptyset_{gN} = \frac{N_v}{C_V}\left(\Pi_c - \frac{dC_v}{dt} - \psi_g\right)$ (eq.22)

$\emptyset_{SN} = \frac{N_v}{C_V}\left(\Pi_c - \frac{dC_v}{dt} - \psi_s\right)$ (eq.23)"

**What is dC/dt in eq 19,20**

*The dC/dt represents the rate of change in a pool. We added the description and reference to Clark et al., (2011) for more information in the revised text as follows:*
"$\frac{dC_v}{dt}$ is rate of change in plant C (see Clark *et al.,* (2011) for more detail),"

**Eq19 how does this NPP differ from the one in eq17 - or is it the same?**

*It is corrected as follows:*
"Therefore, BP is calculated as the difference between potential NPP ($\Pi_c$) and total excess C:

$$BP = \Pi_c - \psi_t \qquad \text{(eq.17)"}$$

**Eq28: how does R_in related to R_n?**

*The R_in is the final estimated soil respiration from potential soil respiration and N/P limitation. We clarified it in the revised text as follows:*
"The soil respiration from each soil layer ($R_{i,n}$) is estimated from potential soil respiration ($R_{POT_{i,n}}$) for the DPM, RPM pools and the litter decomposition rate modifier ($F_{P_n}$) as follows:

$$R_{i,n} = R_{POT_{i,n}} \times F_{P_n} \qquad \text{(eq.33)"}$$

**Eq29: what is DEM_DPMn and DEM_RPMn? What is the rationale behind this formulation?**

*DEM_DPMn and DEM_RPMn correspond to the net demand associated with decomposition of litter pools. Thus, where FP (that is estimated based on DEM_DPMn and DEM_RPMn) is less than 1, the availability of N or P limits the decomposition of litter into soil organic matter. This limitation is because respiration is carried out by microbes that require sufficient N or P to convert the RPM and DPM pools into BIO and HUM pool. We further clarified this in the revised text on lines 378-383 as follows:*
"The net demand associated with decomposition of litter pools ($DEM_{k,n}$) represents the P required by microbes which convert DPM and RPM into BIO and HUM. The limitation due to insufficient P availability is estimated based on the potential mineralization ($minl_{p-pot}$) and immobilization ($immob_{p-pot}$) (in kg P m$^{-2}$ yr$^{-1}$) of pools (k) as follows:

$$DEM_{k,n} = immob_{p-pot,k} - minl_{p-pot,k} \qquad \text{(eq.35)"}$$

**Eq30: why is the plant demand a function of soil P availability and not plant P demand?**

*This represents the demand related with the decomposition of each litter pool not the plant P demand. We clarified this in the revised text in the previous comment.*

**Eq31-36: give the rationale behind the choice of equation.**

*The answer to this is given in the comment on line 438.*

**Table 1: misses variables e.g. desorption fluxes**

*The variables are added in the revised table.*

**Table 2: misses value for parameters which are PFT or depth dependent**

*The values are added in the revised table.*

**Eq39,40 is nleaf the same as N?**

*Indeed, both nleaf and N are foliar N concentrations in area basis. This is now clarified in the revised version as follows:*
"JULES vn5.5 (JULES CN in this study) estimates $V_{cmax}$ (µmol m$^{-2}$ s$^{-2}$) based on Kattge et al. (2009) using foliar N concentrations in area basis ($nleaf$), as follows:

$$V_{cmax} = v_{int} + v_{sl} * nleaf \qquad (eq.45)$$

where $v_{int}$ is the estimated intercept and $v_{sl}$ is the slope of the linear regression derived for the $V_{cmax}$ estimation. We incorporated an additional P dependency on the estimation of $V_{cmax}$ following Walker et al. (2014) as follows:

$$\ln(V_{cmax}) = 3.946 + 0.921\ln(N) + 0.121\ln(P) + 0.282\ln(N)\ln(P) \qquad (eq.46)$$

Where N and P are foliar concentrations in area basis."

**Line 248: : Zaehle & Friend has no P cycle, the references seems inappropriate**

*Please note that the reference is appropriate. We derived these values from Clark et al 2011 and Zaehle & Friend and applied the C:P to estimate the k for each pool. This is information was missed in our submitted manuscript and now it is clarified in the revised text in lines 335-344 as follows:*
"Description of the litter production of P ($F_{P_n}^{lit}$) (arrow b in Fig 1) follows JULES-CN as in Wiltshire *et al.*, (2021) and is calculated based on the litter flux of C (kg C m$^{-2}$ yr$^{-1}$) using leaf, root and wood turnovers (yr$^{-1}$), and through the vegetation dynamics due to large-scale disturbance and litter production density, as follows:

$$F_{P_n}^{lit} = (1 - k_{leaf})\gamma_{leaf}C_{leaf} \times C{:}P_{leaf} + (1 - k_{root})\gamma_{root}C_{root} \times C{:}P_{root} + \gamma_{wood}C_{wood} \times C{:}P_{stem}$$
$$(eq.28)$$

where $\lambda$ is the leaf, root and stem re-translocation (at daily timestep) coefficient (Zaehle and Friend, 2010; Clark *et al.*, 2011) and the related $C{:}P$ ratios for P fraction and $\gamma$ is a temperature dependent turnover rate representing the phenological state (Clark *et al.*, 2011)."

**Line 438: why do you need to cap these fluxes? Why can these max be assigned to observed stocks? You should explain.**

*These fluxes need to be capped in order to keep the equilibrium between free and sorbed organic and inorganic soil P. Otherwise, there will be a constant adsorption from soil P pools which disturbs the mass balance and equilibrium between these pools. Following Wang, Law and Pak, (2010), we defined the same term as "maximum sorption capacity". However, while there are available measurements for this parameter from some global data (e.g. Sun et al., 2021), as we have site measurement for this parameter, we used the maximum observed organic and inorganic P for this term. We clarified it in the revised text as follows:*
"The organic and inorganic soil P assumed to be always at equilibrium with the relative sorbed pools (Wang, Law and Pak, 2010). Thus, in order to cap P sorption and uptake capacity, the maximum sorption capacities ($P_{in-max_n}$, $P_{or-max_n}$, eq.37 and 39) (adopted from (Wang, Houlton and Field, 2007)) were prescribed using maximum observed sorbed inorganic and organic P. Hence, the maximum sorption capacity defines the equilibrium state of sorbed and free-soil P."

**Line 458: why do you prescribe LAI? Isn't that computed prognostically by JULES?**

*Indeed, LAI is computed prognostically. The "initial value" however, should be defined in the model. We clarified it in the text as follows:*
"Furthermore, the averaged measured LAI from study site was used to initialise the vegetation phenology module, but was allowed to vary in subsequent prognostic calculations."

**The choice of various different ways to label fluxes make it hard to read the equations. I would suggest adopting a more homogenous way. Here are some examples how inhomogeneous labelling: compare fluxes in eq9**

*We modified it as follows:*

$$\frac{dP_{BIO}}{dt} = 0.46 \times F_{immob\,P_n} + F_{P_{OS\,BIO,n}}{}^{desorp} - F_{minl\,P\,BIO,n} - F_{P_{OS\,BIO,n}}{}^{sorp} \qquad (eq.9)$$

$$\frac{dP_{HUM}}{dt} = 0.54 \times F_{immob\,P_n} + F_{P_{OS\,HUM,n}}{}^{desorp} - F_{minl\,P\,BIO,n} - F_{P_{OS\,HUM,n}}{}^{sorp} \qquad (eq.10)$$

**Eq10 and eq12 use two different spellings to refer to desorption. Same goes for occlusion in eq 12 and 14.**

*We modified it as follows:*

$$\frac{dP_{HUM}}{dt} = 0.54 \times F_{immob\,P_n} + F_{P_{OS\,HUM,n}}{}^{desorp} - F_{minl\,P\,BIO,n} - F_{P_{OS\,HUM,n}}{}^{sorp} \qquad (eq.10)$$

**For carbon fluxes (eq16) you again use also a mix of labels (e.g. greek letters for exudate, NPP. etc).**

*The rationale behind using the Greek letters for C fluxes is to keep the similarity with the C model description (Clark et al., 2011). However, we modified equations and unified the labels where possible. More information above.*

**See labelling of CP ratio in eq1-3 vs eq19ff**

*We modified equation 19 as mentioned above.*

**You use P subscript to refer to potential and phosphorus.**

*We replaced the P with POT for potential in the revised version.*

**Eq 42-44 vs equations before: concerning leaf, root, wood c mass**

*We corrected these equations as follows:*
$$C_{leaf} = \sigma_l\, L_b \qquad\qquad (eq.48)$$
$$C_{root} = C_{leaf} \qquad\qquad (eq.49)$$
$$C_{stem} = a_{wl}\, L_b{}^{b_{wl}} \qquad\qquad (eq.50)$$

**I didn't list all the inconsistencies here, there are more but it shouldn't be the job of the referee ...**

*Thank you for your detailed comments on equations and method. We further modified equations which are not listed here according to your previous comments and corrections. Please see the changes in the revised version with tracked changes.*

**others**

**Line 23: N -> nitrogen**

*Corrected.*

**Line 24: not only tropical systems (Huo et al 2020 and references therein): Hou, E., Luo, Y., Kuang, Y. et al. Global meta-analysis shows pervasive phosphorus limitation of aboveground plant production in natural terrestrial ecosystems. Nat Commun 11, 637 (2020). https://doi.org/10.1038/s41467-020-14492-w**

*We modified the abstract lines 24-25 as follows:*
"In topical ecosystems, this is likely to be important as N tends to be abundant but the availability of rock-derived elements, such as P, can be very low."

*Also modified the introduction lines 79-82:*
"P-limitation is pervasive in natural ecosystems (Hou *et al.*, 2020) and the lack of large P inputs into ecosystems, especially those growing on highly weathered soil, may make P limitation a stronger constraint on ecosystem response to elevated $CO_2$ (e$CO_2$) than N (Gentile et al., 2012; Sardans, Rivas-Ubach and Peñuelas, 2012)."

**Line 30: you don't use any data from the fertilizer experiment, only from the control plot. This is misleading.**

*Thank you for your comment. In order to prevent the misleading, we modified the fertilization experiment to "study site" throughout the text and tables. More information above.*

**Line 31-35: Given this study introduces a new model, I would expect more information on the model evaluation and performance, less on model prediction.**

*We agree. Therefore, we performed extended model sensitivity tests, including all P-related parameters and discuss results in further detail. More information above.*

**Line 54: nutrient -> nitrogen**

*Corrected.*

**Line 58: you need to add references which indicates that much progress regarding has been made**

*We added the references accordingly:*
"Seven years later, for the update in CMIP5 (Anav *et al.*, 2013), three models out of eighteen with N dynamics were included (Bentsen *et al.*, 2013; Long *et al.*, 2013; Ji *et al.*, 2014)."

**Line 64/65: Reference is missing. E.g. Huo et al 2020 and references therein**

*We added the references as follows:*
"Soil P is hypothesized to be among the key limiting nutrients to plant growth in tropical forests (Vitousek *et al.*, 1997, 2010; Hou *et al.*, 2020), unlike temperate forest where N is hypothesised to be the main constraint (Aerts and Chapin, 1999; Luo *et al.*, 2004)."

**Line 68-73: This formulation is unclear and the statement is not very nuanced.**

*We modified these lines as follows:*
"Although N limitation can impact the terrestrial C sink response to increasing atmospheric $CO_2$ by changing plant C fixation capacity (Luo et al., 2004), this can be partially ameliorated over time by input of N into the biosphere via the continuous inputs of N into ecosystems from atmospheric deposition and biological N fixation (Vitousek et al., 2010). P-limitation is pervasive in natural ecosystems (Hou *et al.*, 2020) and the lack of large P

inputs into ecosystems, especially those growing on highly weathered soil, may make P limitation a stronger constraint on ecosystem response to elevated $CO_2$ ($eCO_2$) than N (Gentile et al., 2012; Sardans, Rivas-Ubach and Peñuelas, 2012)."

**Line 73:  Most model studies don't show that P is stronger than N with respect to limiting the ecosystem response to eCO2 (e.g Goll et al 2012, Yang et al 2014)  Goll, D. S., Brovkin, V., Parida, B. R., Reick, C. H., Kattge, J., Reich, P. B., van Bodegom, P. M., and Niinemets, Ü.: Nutrient limitation reduces land carbon uptake in simulations with a model of combined carbon, nitrogen and phosphorus cycling, Biogeosciences, 9, 3547–3569, https://doi.org/10.5194/bg-9-3547-2012, 2012.**

*We modified these lines as explained in the previous comment.*

**Line 73: what is separate knowledge?**

*We modified these lines as follows:*

"P-limitation is pervasive in natural ecosystems (Hou *et al.*, 2020) and the lack of large P inputs into ecosystems, especially those growing on highly weathered soil, may make P limitation a stronger constraint on ecosystem response to elevated $CO_2$ ($eCO_2$) than N (Gentile et al., 2012; Sardans, Rivas-Ubach and Peñuelas, 2012). This causes considerable uncertainty in predicting the future of the Amazon forest C sink (Yang *et al.*, 2014)."

**Line 86-88: reference does not support this claim**

*We modified these lines and references as follows:*
"However, modelling studies are unable to reproduce observed spatial patterns of NPP and biomass in the Amazon due to missing information on nutrient availability and soil fertility impact on productivity (Wang, Law and Pak, 2010; Vicca *et al.*, 2012; Yang *et al.*, 2014) and due to the lack of inclusion of soil P constraints on plant productivity and function."

**Line 88: 2016 not so recent**

*We modified these lines as follows:*
"Nevertheless, some modelling works have focused on improving process and parameter representation using the observational data of spatial variation in woody biomass residence time (Johnson *et al.*, 2016), soil texture and soil P to parameterise the maximum RuBiCo carboxylation capacity ($V_{cmax}$) (Castanho *et al.*, 2013)."

**Line 133: this is not clear. I would assume light competition affects plant cover via photosynthesis and growth, and not directly.  You should be more clear how canopy processes are resolved.**

*In JULES we assume a process-based leaf-level photosynthesis scaled up to the canopy. Therefore, in JULES CNP in order to keep consistency with JULES C-CN, we also assume a multi-level canopy, with N and P concentration in leaves decreasing exponentially through the canopy (CanRadMod 6) (Clark et al., 2011). This information is added to the revised text in lines 142-145 as follows:*
"In JULES we assume a process-based leaf-level photosynthesis scaled up to the canopy. Therefore, in JULES CNP in order to keep consistency with JULES C-CN, we also assume a multi-level canopy, and leaf N and P in exponentially decreases through the canopy (CanRadMod 6) (Clark *et al.*, 2011)."

**Line140: it is not clear how N limitation on SOC can applied to vegetation. What thats' the evidence/rationale for that?**

*This limitation is because respiration is carried out by microbes that require sufficient N or P to convert the RPM and DPM pools into BIO and HUM pool. We further clarified it in the revised text in the method section as explained above.*

**Figure 1: This figure caption is too short and does not reflect the text. E.g where is the depth information here? It would ne nice to indicate in the figure which equation explains which flux.**

*We modified this figure following your suggestions and added the related equations to each flux as follows:*

[Figure]

**Figure. 1-** JULES CNP model scheme

**Line 159: you list only 7 pools, not 8.**

*It is now corrected as follows:*
"JULES represents eight P pools comprising organic and inorganic P: in plant P ($P_p$) and soil pools (in each soil layer (n)): litter P ($P_{O_l}$), soil organic P ($P_{O_s}$), soil inorganic P ($P_{in}$), organic sorbed ($P_{org-sorp}$), inorganic sorbed ($P_{inorg-sorp}$), parent material ($P_{pm}$) and occluded ($P_{occ}$) P comprised of both organic and inorganic P. All pools are in units of kg P m$^{-2}$ (Fig 1, Tables 1 and 2)."

**Line 165: are they rigid ratios?**

*We clarified it in the revised text as follows:*
"Plant P pool is composed of leaf ($P_{leaf}$), fine root ($P_{root}$) and stem together with coarse root ($P_{stem}$), which are related to their associated C pools ($C_{leaf}, C_{root}, C_{stem}$) in (kg C m$^{-2}$) and fixed C to P ratios ($C:P_{leaf}, C:P_{root} C:P_{stem}$) as follows"

**L226: this section also give C fluxes, should be relabelled**

*We corrected the title as follows:*
"2.2.2.   C and P fluxes"

**Line 228: I guess this should read: 'In JULES-CNP, NPP represent the potential amount'**

*We needed to modified it follows:*
"NPP in JULES is calculated as the difference between GPP and autotrophic respiration. In JULES-CNP, potential NPP represent the amount of C, available for tissue growth (C density increase) on a unit area, and spreading (vegetation cover increase as a result of reproduction and recruitment), ie to increase the area covered by the vegetation type, assuming no nutrient limitation."

**Line 228-235: this is hard to follow. It seems you list definitions of potential NPP, actual NPP and BP as a justification for your modelling choice. I struggle to make the connection.**

*Yes, we agree, and have modified these lines as follows:*
"NPP in JULES is calculated as the difference between GPP and autotrophic respiration. In JULES-CNP, potential NPP represent the amount of C, available for tissue growth (C density increase) on a unit area, and spreading (vegetation cover increase as a result of reproduction and recruitment), ie to increase the area covered by the vegetation type, assuming no nutrient limitation. The reported NPP in the literature often includes other C fluxes related to the exudates, volatiles production and non-structural carbohydrates (Malhi *et al.*, 2009; Chapin *et al.*, 2011; Walker *et al.*, 2021) which are challenging to measure (Malhi, Doughty and Galbraith, 2011). Therefore, actual NPP is for our purposes equal to Biomass Production (BP), and is calculated as potential NPP minus excess C (lost to the plant through autotrophic respiration), with the latter the C that cannot be used to growth new plant tissue due to insufficient plant nutrient supply. Hence, if the system is limited by the availability of N and/or P, NPP will be adjusted to match the growth that can be supported with the limited N or P supply, with any excess carbohydrate lost through excess C as autotrophic respiration."

**Line 244-246: maybe first follow the BP idea before. Also I think it needs some more explanation to model litter production as a function of NPP and not plant pools.**

*We modified this part as you suggested as follows:*
"Therefore, BP is calculated as the difference between potential NPP ($\Pi_c$) and total excess C:

$$BP = \Pi_c - \psi_t \tag{eq.17}$$

The litter production in JULES before limitation is estimated based on the as follows:

$$F_{C_n}^{lit} = \gamma_{leaf} C_{leaf} + \gamma_{root} C_{root} + \gamma_{wood} C_{wood} \tag{eq.18}$$

where $\lambda$ is the leaf, root and stem re-translocation (at daily timestep) coefficient (Clark *et. al.,* 2011) and $\gamma$ is a temperature dependent turnover rate representing the phenological state (Clark *et al.*, 2011)."
**Line 284: Zaehle & Friend has no P cycle.**

*Please find the answer in the comment on line 248.*

**Line 300-306, I can't follow the equation because of the confusing way of labelling to C:P ratios (see point above).**

*In order to prevent confusion, we relabelled these paragraphs and equations as follows:*
"The decomposition of litter ($dec^{lit}$) (arrow c in Fig 1) depends on soil respiration ($R$) (kg C m$^{-2}$ yr$^{-1}$), the litter C:P ratio ($C:P_{lit}$) at each soil layer (n) as follows:

$$dec_P^{lit} = \frac{\sum_{n=1}^N R_n}{C:P_{lit}} \tag{eq.29}$$

where the $C:P_{lit}$ is calculated based on litter C pool (DPM and RPM) ($lit^C$) (kg C m$^{-2}$ yr$^{-1}$) and litter P pool ($P_{O_l}$) as follows:

$$C:P_{lit} = \frac{\sum_{n=1}^N lit_n^C}{P_{O_{l_n}}} \tag{eq.30}$$

The mineralized ($F_{minl_P}$) (arrow d in Fig 1) and immobilized ($F_{immob_P}$) (arrow e in Fig 1) P fluxes are calculated based on C mineralization and immobilization, C:P ratios of plant (i) (DPM/RPM) ($C:P_{plant}$) and soil (HUM/BIO) ($C:P_{soil}$), soil pool potential respiration ($R_{POT_i}$) (kg C m$^{-2}$ yr$^{-1}$) and the respiration partitioning fraction (resp_frac) as follows:

$$F_{minl_{P_n}} = \frac{\sum_{n=1}^{N} R_{POT_{i,n}}}{\varepsilon_{cp_i}}$$ (eq.31)

$$F_{immob_{P_n}} = \frac{\sum_{n=1}^{N} R_{i,n} \times resp\_frac}{C:P_{soil}}$$ (eq.32)"

**Line 308: remove 'However'**

*Corrected.*

**Line 381-382: this repeats info given before (line 376 ..)**

*We removed this line in the revised version.*

**Line 391-394: Please give information how data from plots were aggregated. You also indicate vegetation information from AFEX in Tab3.. How were vegetation C stock derived.**

*We added this information in the revised version lines 520-527 as follows:*
"The measurements were collected between 2017 and 2018 in control plots. All measurements were conducted at four soil layers (0-5 ,5-10, 10-20, 20-30 cm). However, to be consistent with the JULES model soil layer discretization scheme, we defined 4 soil layers (0-10 cm, 10-30 cm, 30-100 cm and 100-300 cm) and we used the average between 0 and 30 cm to compare against the measurement from the same depth for model evaluation.
Vegetation C stocks were derived based on tree diameter measurements at breast height, that are linked to allometric equations and wood density databases to estimate the C stored in each individual tree, and then scaled to the plot (Chave *et al.*, 2014)."

**Line 398: you need to explain how photosynthesis is implemented in JULES. Is it a big leaf approach?**

*Please find the answer to this on your comment on line 133.*

**Line 428-436: you should explain which observable pools were assigned to which modelled pools.**

*We modified these lines as suggested in the revised version as follows:*
"The following belowground data were used to represent various soil P pools: Resin and bicarbonate inorganic P (inorganic P:$P_{in}$), organic bicarbonate P (organic P: $P_{O_S}$), NaOH organic P (sorbed organic P: $P_{org-sorp}$), NaOH inorganic P (sorbed inorganic P : $P_{inorg-sorp}$), residual P (occluded P: $P_{occ}$) and HCL P (parent material P: $P_{pm}$) (Table 3)."

**Line: 459 : what's the equilibrium criteria, e.g. do you apply the trendy protocol criteria? From the equations occluded and parental material pools cannot equilibrate. I guess you considered them outside the boundary of your model.**

*Please find the answer to this on your comment:* "**2 Some of the assumptions / choices are in contrast to current understanding and consensus while no explanation was given.**"

**Line 460: 1000 times what? Years , timesteps, …**

*We modified the spin-up description in the revised version in lines 553-558 as follows*
"The JULES CNP simulations were initialized following the same methodology as in Fleischer et al., (2019), by the spin-up from1850 recycling climatology to reach equilibrium state (Figure S1) and spin up was performed separately for three versions of JULES (C/CN/CNP) following the same procedure. Furthermore, the transient run was performed for the period 1851-1998 using time-varying $CO_2$ and N deposition fields. Finally, for the extended simulation period (1999-2019) two runs were performed, the first with ambient the second elevated $CO_2$ concentrations"

**Line 470: why did you limit it to two years? Your evaluation is based on C variables, so your argumentation based on soil P measurement seems invalid.**

*Please note that our evaluation of P pools (soil organic and inorganic and plant organic pools) was based on the measurements from the same period. However, the additional C evaluation was also performed based on the measurements that period. More information can be found in the answer to general comment section (page 26 of this document).*

**Line 483: Be clear about the timestep use here.**

*We added this information as follows:*
We studied the Water Use Efficiency (WUE) (eq. 53) at half-hourly timestep, then aggregated per month, as one of the main indicators of GPP changes (Xiao *et al.*, 2013), and soil moisture content (SMCL), as one of the main controllers of maximum uptake capacity (eq. 27), in order to better understanding the changes in GPP, P demand and uptake as well as excess C fluxes.

$WUE = GPP/Transpiration$ (eq.53)"

**Line 500-503: how was this done exactly? E.g. Did you vary the ratio one by one? The relative differences between C:P ratios is critical for the availability of P. Did you test for this effect (as has been done in earlier studies (e.g. Goll et al 2012)). The N:P ratios are usually much more constrained than the C:P ratios ? Did you take this information into account when varying C:P ratios?**

*We conducted the sensitivity tests by changing only one parameter at the time and testing once with +50% and once with -50% of the measured value and keeping other parameters unchanged. Since the derived model parameters from measurements already have their own level of uncertainty, we took 50% of change to test these parameters at reasonable degree. Moreover, our model is essentially linear (see main model's equations in the manuscript, most fluxes follow first order reaction rate), so we are not too worried about larger perturbations. We still believe that changing the parameter values by +/-50% is a valid choice to clearly show the insensitivity of the model to some parameters, while still exploring possible ranges for these parameters. This information is now added in lines 595-604 as follows:*

"To test the sensitivity of the P and C related processes to the model P parameters, six sets of simulations were conducted with modified plant C:P stoichiometry (Plant C:P: *SENS1*), P uptake scaling factor ($K_P$) (Kp: *SENS2*), inorganic (KP_sorb_in: *SENS3*) and organic (KP_sorb_or: *SENS4*) P adsorption coefficients ($K_{sorp-or}, K_{sorp-in}$), and maximum inorganic (KP_sorb_in_max: *SENS5*) and organic (KP_sorb_or_max: *SENS6*) sorbed P ($K_{or-max}, K_{in-max}$). These values were prescribed to vary between ±50% of the observed values and their effect on C pools (plant and soil C) and fluxes (NPP and excess C), and P pools (plant, soil, and soil sorbed P) was assessed. As the derived model parameters from measurements haver their own level of uncertainty, we took the 50% of the change to test these parameters at reasonable degree. However, the occluded and weathered P pools are prescribed for this model application, the occluded and weather P coefficients (other two P-related model parameters) were not part of sensitivity tests."

**Line 514: which pools exactly? How did you calibrate them?**

*The answer is given in general comment section (page 26 of this document),*

**Table 3: P pools were used to calibrate the model ( stated on line 514) not to evaluate them.**

*The answer is given in general comment section (page 26 of this document),*

**Table 5: The 'observed NPP' from Fleischer et al 2019 is based on biomass increments, thus you should compare it to modelled BP. This also affects CUE**.

*Please find the answer to this on your general comments section (page 25 of this document).*

**Figure 3: you should add some measure of uncertainties of the obs, e.g. variation among plots**

*We revised the figure and added the observational uncertainties:*

[Figure]

**Figure. 3-** JULES C, CN, CNP modelled vs measured C pools (Leaf, root, wood, Veg and Soil C) (in kg C m$^{-2}$) and fluxes (BP and Litter C) (in kg C m$^{-2}$ yr$^{-1}$) and CUE under ambient $CO_2$. Note that CUE is unitless.

**Line 555: how do you compute P limitation? I couldn't find an equation.**

*This information is now added to the revised manuscript following your general comments sections (page 16-18 of this document). More information above.*

**Line 655: the units of GPP are misleading, i.e. monthly GPP as yr-1.**

*We corrected the units in the revised version section 3.3 lines 761-776 as follows:*
"Under ambient $CO_2$ condition the highest GPP is estimated at 3.5±0.19 kg C m$^{-2}$ month$^{-1}$ in July and the lowest at 2.06±0.61kg C m$^{-2}$ month$^{-1}$ in October (Figure. 6-a). The estimated WUE and SMCL in October is among the lowest estimated monthly values at 2.3±0.51 kg $CO_2$/kg $H_2O$ and 526.2±31 kg m$^{-2}$ respectively (Figure. 6-c). The highest P demand is estimated at 0.4±0.02 g P m$^{-2}$ month$^{-1}$ in July and the lowest demand at 0.2±0.08 g P m$^{-2}$ month$^{-1}$ in October. Consequently, the highest and lowest uptake (0.32±0.01 and 0.19±0.07 g P m$^{-2}$ month$^{-1}$, respectively). The excess C for the highest and lowest GPP and demand periods are estimated at 0.4±15 and 0.04±0.07 kg C m$^{-2}$ month$^{-1}$, respectively.

However, similar to ambient $CO_2$, under $eCO_2$ condition the highest estimated GPP is in July at 4.36±0.21 kg C m$^{-2}$ month$^{-1}$ and lowest for October 3.02±0.75 kg C m$^{-2}$ month$^{-1}$ (Figure. 6-b). The estimated WUE and soil moisture content (SMCL) for the lowest GPP period is among the lowest monthly estimated values at 3.5±0.74 kg $CO_2$/kg $H_2O$ and 552±33 kg m$^{-2}$ for October respectively (Figure. 6-d). The highest P demand is estimated for July at 0.51±0.02 g P m$^{-2}$ month$^{-1}$ with the uptake flux of 0.31±0.02 g P m$^{-2}$ month$^{-1}$ and the lowest demand is estimated for October at 0.32±0.1 g P m$^{-2}$ month$^{-1}$ with the estimated uptake flux of 0.26±0.06 g P m$^{-2}$ month$^{-1}$. The highest excess C flux is also for July at 1.01±0.17 kg C m$^{-2}$ month$^{-1}$ and lowest for October 0.27±0.29 kg C m$^{-2}$ month$^{-1}$, respectively."

**Line 674: why is it the higher soil water content which enhances uptake and not something? This needs to be demonstrated or explained in more detail**.

*We agree with the reviewer that further explanation on uptake flux was required. In JULES both vertical discretisation and mineralisation terms are controlled by soil moisture and temperature. Thus, the increase SMCL results in higher P concentration and uptake consequently. We modified these lines in the revised version as follows:*

"However, despite the P limitation in both $eCO_2$ and ambient $CO_2$ conditions, the P uptake flux under $eCO_2$ is higher than the ambient $CO_2$ condition. This is due to the higher WUE and increased SMCL (controlling uptake capacity (eq. 27)) under $eCO_2$ condition, hence more water availability during the dry season to maintain productivity and critically transport P to the plant (see eq. 27), compared to ambient $CO_2$ condition (Figure. 6-c and d). Additionally, in JULES both the vertical discretisation (Burke, Chadburn and Ekici, 2017) and mineralisation terms (Wiltshire *et al.*, 2021) depend on the soil moisture and temperature. Thus, higher P concentration and uptake under $eCO_2$ condition."

**L690: this is grassland study. You need to explain why this can be compared to a tropical forest.**

*The reference was not appropriate and necessary in this line. We removed it in the revised version.*

**L709: you should make the link between high competition for P and unclear role of P for plant CO2 response.**

*We modified the discussion and added this information in lines 812-826 as follows:*

"As soil P availability is low in the majority of Amazonia (Quesada *et al.*, 2012), the competition for nutrients by both plant and soil communities is high (Lloyd *et al.*, 2001). The responses of these communities to $eCO_2$ under P limited conditions remains uncertain (Fleischer *et al.*, 2019). These responses in P enabled models are represented in different ways regarding the excess C which is not used for plant growth due to P limitation. Either growth is directly downregulated taking the minimum labile plant C,N and P (Goll *et al.*, 2017), or photosynthesis is downregulated via $V_{cmax}$ and $J_{max}$ (Comins and McMurtrie, 1993; Yang *et al.*, 2014; Zhu *et al.*, 2016) and finally models like JULES CNP downregulate NPP via respiration of excess carbon that cannot be used for growth due to plant nutrient constraints (Haverd *et al.*, 2018). The estimated CUE depends on the modelling approach. Models that down regulate the photosynthetic capacity and GPP consequently (Comins and McMurtrie, 1993; Yang *et al.*, 2014; Zhu *et al.*, 2016), simulate a positive CUE response to $CO_2$ fertilization while models that down regulate the NPP and respire the excess C (Haverd *et al.*, 2018) simulate a negative CUE response (Fleischer *et al.*, 2019) which is in line with the studies showing lower CUE when nutrient availability declines (Vicca *et al.*, 2012). However, this remains a major uncertainty in understanding the implication of P limitation on terrestrial biogeochemical cycles."

**L711: it's odd to refer to the site as well document. From your work it seems there is hardly any data available for model evaluation available.**

*Please note these lines are now modified as explained in the previous comment.*

**Line 712: you included all major processes in detail ? Then the role of P should be clear now or?**

*Please note these lines are now modified as explained in the previous comment.*

**Line 720-725: unclear formulations. Weren't the soil P pools optimized in JULES?**

*Information on the optimized pools is given in general comment section (page 26 of this document). We also modified the discussion section 4.1 lines 837-841 as follows:*

"JULES-CNP could reproduce the magnitude of soil organic and inorganic P pools and fluxes. The relative distribution of total organic P, total inorganic P and residue P fractions of total P in soils under Brazilian Eucalyptus plantations (Costa *et al.*, 2016) shows inorganic P fraction of 28% from total soil P which is close to our estimation of 24% and organic P fraction of 30% from total soil P which is higher than our estimated fraction of 18%."

**Line 7 27: inappropriate reference**

*We corrected the reference as follows:*
"Our estimated maximum P uptake, which represents the actual available P for plant uptake (Goll *et al.*, 2017), for both ambient and e$CO_2$ conditions, is highly correlated with the plant P demand ($R^2$ = 0.96 and 0.52 respectively)."

**Line 829: 'a cornerstone' - a more humble forumation could be use here**

*We modified this line as follows:*

[revised manuscript text omitted]

**CEC1:**

**Please provide the reasons, why registration is required and how an interested person can get access to code, i.e. how to register.**

*We modified the code availability section and clarified the link for the registration and software license form, lines 962-970 as follows:*

"*Code availability*

The modified version of JULES vn5_5 and the P extension developed for this paper are freely available on Met Office Science Repository Service:

https://code.metoffice.gov.uk/svn/jules/main/branches/dev/mahdinakhavali/vn5.5_JULES_PM_NAKHAVALI/ after registration (http://jules-lsm.github.io/access_req/JULES_access.html) and completion of software license form. Codes for compiling model available at: (https://doi.org/10.5281/zenodo.5711160). Simulations were conducted using two sets of model configurations (namelists): ambient $CO_2$ condition (https://doi.org/10.5281/zenodo.5711144) and elevated $CO_2$ condition (https://doi.org/10.5281/zenodo.5711150)"

---

## Author Response (AR2)

Dear Authors,

3 reviewers have assessed your revised manuscript. As you will see, not all reviewers were satisfied with the revision.

Two main points that stand our from the review reports: 1. the novelty of the work, 2. the evaluation of the presented model.

About the novelty of the work we can have a long debate (also the reviewers have different opinions here), but in my opinion in principle the addition of the P cycle in a major land surface mode (like JULES), is worth to be published as a model description paper in a journal like GMD. However, I advise the authors to be very clear and honest about novel and non-novel aspects in the text. If the P cycle is implemented in the model based on existing concepts, or other models this should be explicitly stated in the text.

The real weakness of the current manuscript for me is still the evaluation of the model. If this manuscript aims to be the reference study presenting the CNP version of JULES, then a more thorough model validation needs to be added to the analysis. I can agree that this would be a site-scale evaluation (i.e. not including a large scale regional evaluation). But it should at least be a multi-site evaluation, and preferably include an evaluation for an experiment (one of the reviewers is suggestion to use the Gigante nutrient addition experiments for example). Only with such an evaluation the manuscript can act as a reference publication for other studies that use the validated JULES-CNP model for applications and to address actual research questions.

A revised version of the manuscript should thus account for these two point and should address all other remarks and suggestions raised by the reviewers.

best regards,
Hans Verbeeck

*Dear Hans Verbeeck,*

*We thank you for your comment on our submitted manuscript. Following are our responses and the modifications we did based on your two main points:*

*1- Novelty of model:*

*For the development of JULES-CNP which will be eventually incorporated in the UK earth system model, instead of coming up with brand new equations or processes that have not been yet incorporated in any global model which will need a lot of testing, we opted for implementing existing and already tested equations from global land surface/vegetation P enabled models. The current version of JULES-CNP forms the basis of future developments. As requested by reviewer 3 after the first round of reviews, we included citations to all equations taken from the literature.*

*This is now clearly stated in two parts in the manuscript, in the introduction in lines 119-123:*
"Here, we describe the development and implementation of the terrestrial P cycle in the Joint UK Land Environment Simulator (JULES) (Clark *et al.*, 2011), the land component of the UK Earth System Model (UKESM), following the structure of the prior N cycle development (Wiltshire *et al.*, 2021) and utilising state of the art already tested and implemented descriptions of P cycling in other land surface models (Wang, Houlton and Field, 2007; Zhu *et al.*, 2016; Goll *et al.*, 2017)."

*And also, at the start of section 2.2 on JULES-CNP description in lines 168-170:*
"JULES-CNP includes the representation of the P cycle in JULES version (vn5.5) and it is built on existing and well tested representations of P cycling in other global land surface models (Wang, Houlton and Field, 2007; Yang *et al.*, 2014; Goll *et al.*, 2017; Sun *et al.*, 2021)."

*However, a unique feature of our extended P component in JULES is the estimation of the soil organic and inorganic P sorption based on the saturation status of the relative adsorbed P pools. This is now clarified in the manuscript in the introduction (line 125 -129) as follows:*

"The model (JULES-CNP) is parameterized and calibrated using novel in situ P soil and plant data from a well-studied forest site in Central Amazon near to Manaus, Brazil with soil P content representative of 60% of soils across the Amazon basin. The new developed P component estimates the sorption of the soil organic and inorganic P based on the saturation status of the adsorbed P pools, which is unique compared to the other existing P models and enable more realistic estimation of P ad/desorption processes."

*2- Evaluation:*

*We have performed additional site-level evaluation to show model performance at other sites. The extended test sites are located in the Amazon (AGP-01, SA03 and CAX) which include a gradient of fertility from west to east Amazon, and two manipulation experiments one in the Gigante Peninsula in Panama and one at the Hawaii chronosequence (Hawaii Kokee). These site level simulations which were parameterised with site level tissue and soil C:P ratios and maximum sorbed P capacities using site specific parameters, showed a significant improvement of JULES-CNP over the C and CN only versions. Specifically, simulated C pools and fluxes with JULES-CNP were closest to the measurements as opposed to JULES C and CN which overestimated all observations at all test sites. Additional text in all sections (Abstract, introduction, methods and results) is included in track changes in the manuscript. Below we include tables with site selected (Table 3 and 5 in the text) and figures with results obtained (figures 4 and S8 in the text)*

**Table 3.** Test sites name, location and climate characterises.

| Site | Name | Location | | Climate | |
|------|------|----------|--|---------|--|
| | | Lat. | Lon. | Rainfall (mm yr$^{-1}$) | Temperature(˚C) |
| *Study site* | AFEX project | -2.58 | -60.11 | 2431 | 26 |
| **AGP-01** | Agua pudre plot E | -3.72 | -70.3 | 2723 | 25.5 |
| **CAX** | Caxiuanã flux tower site | -1.72 | -51.5 | 2314 | 26.9 |
| **SA3** | Tapajós flux tower site | -2.5 | -55 | 1968 | 26.1 |
| **Gig. Pen.** | Gigante peninsula (control data) | -9.1 | -79.84 | 2600 | 26 |
| **Hawaii K.** | Hawaii Kokee (control data) | 22.13 | -159.62 | 2500 | 16 |

**Table 5.** Additional test sites data used for model parameterisation

| | AGP-01[a,b] | CAX[a,b] | SA3[a,b] | Gig. Pen.[c] | Hawaii K.[b,d] |
|--|-------------|----------|----------|--------------|----------------|
| *Leaf$_{C:P}$* | 600 | 600 | 600 | 700 | 691.5 |
| *Root$_{C:P}$* | 1000 | 1000 | 1000 | 1750 | 1100 |
| *Wood$_{C:P}$* | 3000 | 3000 | 3000 | 5500 | 5937.5 |
| *Soil$_{C:P}$* | 2000 | 2000 | 2000 | 800 | 2000 |
| $K_{or-max}$ | 0.001 | 0.001 | 0.001 | 0.0033 | 0.001 |
| $K_{in-max}$ | 0.001 | 0.001 | 0.001 | 0.0185 | 0.001 |

[a]C:P ratios from Wang, Law and Pak, 2010 and [b]maximum sorbed P capacities from Yang *et al.*, 2014. [c]Mirabello *et al.*, 2013 [d] C:P ratios from Vitousek, 2004

[Figure]

**Figure. 4-** Observed and simulated (JULES C, CN, CNP) C fluxes and pools (averaged measurements: red points, sd: red arrows) and available observed P (dark red points and lines (reported in ppm)) at test sites across the Amazon (AGP, SA03, CAX), Gigante Peninsula (Gig. Pen.) and Hawaii Kokee (Hawaii K.).

[Figure]

**Figure. S8-** Solar radiation at the extended test sites

**RC1:**

I thank the authors for the revision, but I fear the manuscript still suffers from two key flaws: the lack of novelty and a lack of model evaluation. Both flaws have been mentioned by more than one reviewer, who gave suggestions on how to address both points.

The authors argue that documenting the inclusion of a P cycle in LSM which hadn't had one before is novel enough. I have doubts that sufficient to qualify as 'substantial new concepts, ideas, or methods': The authors do not provide any new modelling concepts but have adapted published formulation, the single model application is a repetition of a multi-model exercise, the evaluation is insufficient to show the method is working reliably. Please see in the following more details about aspects.

*We thank the reviewer for their feedback on the revised version. Please see below the answers to the main issues raised by the reviewer. Note, reviewer's comment in grey highlight and our responses in blue italic format, followed by the modified text in black colour.*

**Novelty:**
The author's argumentation regarding novelty in the replies to Dr. Jiang and reviewer #3 are largely based on a misperception of the current state of science in this field:
There are several globally applicable CNP models (including land surface models) which emerged more than 10 years ago (e.g. CABLE(Wang et al 2010), JSBACH (Goll et al 2012), ELM/CLM (Yang et al 2014), ORCHIDEE (Goll et al. 2017)) and more in the pipeline (e.g. QUINCY (Thum et al 2019). JULES is merely another LSM which adds a P cycle. This is not a novelty of this paper.

*As mentioned in the response to the editor, we have added few lines in the manuscript to explicitly say upfront that we are 'utilising state of the art already tested and implemented descriptions of P cycling in other land surface models (Wang, Houlton and Field, 2007; Zhu et al., 2016; Goll et al., 2017.' and that "JULES-CNP includes the representation of the P cycle in JULES version (vn5.5) and it is built on existing and well tested representations of P cycling in other global land surface models (Wang, Houlton and Field, 2007; Yang et al., 2014; Goll et al., 2017; Sun et al., 2021)."*

There are several dedicated 'model-data nutrient cycling studies specifically for the Amazon forest with poor soils and limited P availability' in contrast to what the authors claim on page 3, namely: Goll et al 2018, Yang et al 2014, 2016, Fleischer et al 2019, Sun et al 2020 to name a few. This is not a novelty of this paper.
*Perhaps overlooked by the reviewer, some of the data sets from our study listed in Table 4 used for either model evaluation/initialisation/parameterisation/evaluation or calibration are unpublished, from the Manaus region but still have not been used before in any modelling context.*

The presentation of P processes is arguably not more detailed than in the first CNP models (Wang et al 2010, Goll et al 2012), and less detailed than in later models (processes missing here are: biochemical mineralisation, microbial dynamics, stoichiometric flexibility, atmospheric nutrient deposition, to name a few). There is no novel concept or process in this study.
*Agreed and now explicitly upfront in the text as explained above.*

Other studies also used site specific parameterization of their model when applying at site (e.g. Yang et al 2014, Fleischer et al 2016). This is not a novelty of this paper. Besides, the authors do on stringently use site specific data. E.g. They use weathering rates from Wang et al. 2010 which distinguish soils globally into three weathering classes based on very few data points. This led to unrealistic weathering rates (e.g. compare with Hartmann et al. 2014). Thus, most studies use either site-specific (optimised) rates (e.g. Yang et al 2014, Goll et al 2017) or use data-constrained global gridded weathering rates (Sun et al 2021) to resolve the large variation in weathering inputs (several orders of magnitude).
Other models have soil P structures which are based on measurable soil P pools, too (Yang et al 2014). This is not a novelty of this paper.
A side note: Wang et al 2018 is not a land surface model, but a biome-scale model designed to assimilate a wealth of observational data. The comparison is inappropriate.

*We do not agree with the reviewer on this point. It was already mentioned in the revised version of the manuscript that despite the representation of the weathering processes in model, due to the simulation period,*

*we deactivated this process and instead prescribed a constant weathering release rate (similar to (Goll et al., 2017)), thus the argument the reviewer regarding the weathering process in JULES and the unrealistic estimation is not valid. This is defined in the line 548-549 as follows:*

"Moreover, despite the initial representation of the parent material pool in JULES and its depletion through weathering (eq. 43), as the magnitude of changes in the occluded and parent material pools are insignificant over a short-term (20 years) simulation period (Vitousek et al., 1997), these two pools were prescribed using observations."

*Please find the reaction to the novelty of model in the replies to editor (page 1 of this document).*

Evaluation:
The heavy model calibration vs hardly any evaluation was criticised by all reviewers. The evaluation in the revised manuscript is still insufficient ( i.e. a few C stocks and annual fluxes , and two soil P pools). The uncertainties in the observation does not allow us to distinguish if the C, CN, or CNP model is more realistic (Figure3). Given the large number of model parameters, it is not surprising the model is able to capture the two P pool stocks. The stocks are commonly used as target pools to optimise model parameters (e.g. Wang et al 2010).
No further attempt in model evaluation has been made. Dr Jiang and myself suggested using data from nutrient addition experiment(s) like e.g. from AFEX as the authors emphasised this experiment in their manuscript. But AFEX is not the only nutrient addition experiment (e.g the Gigante experiment in Panama) and other experiments have been used to evaluate models (e.g. Yang et al 2014, Goll et al 2017). Besides, there is other information which can be used to evaluate models (e.g. Sun et al 2021). The availability of data for model evaluation has to be considered when selecting the study site and in the model design, poor judgement regarding site selection is not an argument for insufficient evaluation. Land surface models are commonly criticised for being overparameterized (e.g. Prentice et al 2015); evaluation is key to avoid repeating past mistakes.

*We have addressed this now with five extra sites, 3 from the Amazon across the west to east fertility gradient, one from west, one form central and one from easter amazon, one site from the Gigante Peninsula experiment and one from the Hawaii chronosequence.*

Dr Jiang and myself expressed concerns that there is no data from the eCO2 experiment to evaluate the model, and it has been already simulated by a set of models. The authors claim 'there is a lot of value in knowing where our predictions lie compared to other models' with respect to the eCO2 experiment. However, they do not provide any further explanation of what the value would be or any example. In my opinion, there is little to none. I understand it might be interesting to a model developer to see how his models compare to others, but what is the scientific value?

*Please find the response to the evaluation of model in the replies to editor (page 2-5 of this document).*

**RC2:**

This manuscript is the revised version of a previous discussion paper. In my opinion, the authors have successfully addressed the comments of both myself and the other reviewers. The paper now includes a more complete and detailed model description and a comprehensive parameter analysis. It is now easier to follow and the results are more robust.

I would like to take a moment to address the comments of reviewer 3 regarding novelty. This is a model description study and its main purpose is to provide the reader with a very detailed description of the model development, be it novel or not. It is of great value to the community as it allows others to understand and reproduce the work done in particular models. It is also of great use to the authors themselves, as they can then publish the novel science without the need to describe the new model and the evaluation in such detail. Additionally, and perhaps most importantly, model development is a painstaking and time-consuming task that often gets very little formal recognition, especially for junior scientists and model description papers such as this are in part a formal recognition of developers' work.

*We thank the reviewer for their positive feedback on the revised version, the suggested corrections and for defending the novelty and importance of this study, which was raised by reviewer 3. We have addressed the comments as described below. Note, reviewer's comment in grey highlight and our responses in blue italic format, followed by the modified text in black colour*

A handful of minor comments:

L 144 Do the C:P and N:P ratios remain the same throughout the canopy?

*Indeed, although the leaf N and P exponentially decreases through the canopy, the C:P and N:P ratios remain fixed throughout the canopy. We clarified this in the revised text as follows:*

"Therefore, in JULES CNP in order to keep consistency with JULES C-CN, we also assume a multi-level canopy, and leaf N and P in exponentially decreases through the canopy (CanRadMod 6) (Clark *et al.*, 2011) while the C:P and N:P ratios remain the same."

l 176 Is the parent material pool a pool that can be depleted?

*The parent material can be depleted using weathering rate over the parent material (eq. 43), however this operates over a much longer time scale than our study period (20 years) leading to insignificant changes in the pools. Therefore, these two pools are prescribed in model without consideration of the weathering process. This is further clarified in the second revision as follows (line 534-535):*

"Moreover, despite the initial representation of the parent material pool in JULES and its depletion through weathering (eq. 43), as the magnitude of changes in the occluded and parent material pools are insignificant over a short-term (20 years) simulation period (Vitousek *et al.*, 1997), these two pools were prescribed using observations."

L 325 Check the edit here - the description varies spatially?

*We corrected these lines as follows:*

"Plant P uptake ($F_p^{up}$) varies spatially depending on the root uptake capacity ($u^{max}$) followed by Goll *et al.*, (2017). Therefore, in regions with limited P supply, the plant P uptake is limited to the $u^{max}$ and consequently impacts the excess C and BP."

L 595 Each parameter was varied independently, yes?

*Indeed, each parameter was tested independently. This is further clarified now as follows:*

"To test the sensitivity of the P and C related processes to the model P parameters, six sets of simulations were conducted independently with modified plant C:P stoichiometry (Plant C:P: *SENS1*), P uptake scaling factor ($K_P$) (Kp: *SENS2*), inorganic (KP_sorb_in: *SENS3*) and organic (KP_sorb_or: *SENS4*) P adsorption coefficients ($K_{sorb-or}$, $K_{sorb-in}$), and maximum inorganic (KP_sorb_in_max: *SENS5*) and organic (KP_sorb_or_max: *SENS6*) sorbed P ($K_{or-max}$, $K_{in-max}$)."

L 640 'The excess C flux is highly dependent on the plant P and the overall P availability to satisfy demand' aren't these the only things that excess C is dependent on?

*Indeed, the reviewer is right. The excess depends only on the Plant P and inorganic P availability. We corrected these lines as follows:*

"The excess C flux depends on the plant P and the overall P availability to satisfy demand (Table 5)."

L 891 Perhaps also worth mentioning here some papers that think p deposition will play a role, rather than just the one that doesn't , for example Gross, 2021https://doi.org/10.1111/nph.17344 or Van Langenhove 2020) https://doi.org/10.1007/s10533-020-00673-8 . (Please do not feel like I'm asking to cite these particular two, just any study that shows P deposition will contribute to plant and microbe available P)

*Thank you for the suggested addition and references. We modified these lines as follows:*

"Moreover, despite studies that show the possibility of P fixation as a source of available P for plants (Van Langenhove *et al.*, 2020; Gross *et al.*, 2021), due to the strong fixation of P in the soil (Aerts & Chapin, 2000; Goodale, Lajtha,Nadelhoffer, Boyer, & Jaworski, 2002), the P deposited is unlikely to be available to plants in the short term (de Vries et al., 2014), for this reason this version of JULES CNP did not include P deposition"

RC3:

I thank the authors for addressing my comments. I'm generally happy with most of their responses/revisions. One point that I would like the authors to further strengthen on is its novelty. Again, as I indicated in my previous review, I appreciate the extensive efforts went into this work; this is a great achievement to add P cycle into JULES. However, I'm not sure how to justify its novelty. Most of the P cycle processes included in this work are based on previously published literature, hence I would argue that their inclusion into a model is not the novelty per se. Maybe the authors should further strengthen why adding P into JULES is needed. Moreover, I can see that site-specific evaluation is valuable, and the authors have spent efforts to differentiate variables that are parameterized and those that are simulated, which addressed the concerns over over-parameterization. But

for their predicted CO2 responses, what can we learn given that there is no actual data to evaluate the predictions? The authors often refer to Fleischer et al. (2019), among other papers, to compare their simulation results. I understand that the simulated CO2 responses of JULES fall into the broad spectrum of responses shown in Fleischer et al., which is great. But the value of Fleischer et al. is that they identified different model-based mechanisms to explain the predicted CO2 responses. I would suggest the authors to make comparison in terms of the underlying mechanisms. The way the Discussion is written in its current form reads more like a comparison of numbers. It would be more useful for future model development's purpose to identify why the model predicts these numbers.

*We thank reviewer for their positive comment.*

*Please find the answer to the novelty of the model and evaluation at further test sites in the replies to editor (page -1:3 of this document)*